# Distributionally Robust Markov Games with Average Reward

**Zachary Roch** [1]  **Yue Wang** [1][2]

## Abstract

We propose and study distributionally robust Markov games (DR-MGs) with the average-reward criterion as a crucial framework for multi-agent decision-making under model mismatches and over extended horizons. Under a standard irreducible assumption, we first derive a correspondence between the optimal policies and the solutions of the robust Bellman equation, based on which we further show the existence of a stationary Nash Equilibrium (NE) of the game. We further study DR-MGs under a more general weakly communicating setting. We construct a set-valued map based on the constant-gain optimal robust Bellman operator and show that its value is a subset of the best-response policies. We further prove that this map admits a fixed point, which implies the existence of NE. We then design two algorithms, Robust Nash-Iteration and robust TD Descent, with provably convergent guarantees. Finally, we show that the NE under average-reward can be approximated by the ones for the discounted DR-MGs as the discount factor approaches one. Our studies provide a comprehensive theoretical and algorithmic foundation for decision-making in complex, uncertain, and long-running multi-player environments.

## 1. Introduction

A Markov Game (MG) or stochastic game provides a powerful mathematical framework for modeling sequential decision-making in competitive multi-agent environments (Shapley, 1953; Filar & Vrieze, 1996; Littman, 1994; Fink, 1964). Solving an MG involves finding an equilibrium policy profile for all players, where each player aims to maximize their own cumulative reward. However, a critical challenge in practice is the potential for a mismatch between the assumed MG model and the true underlying environment. This discrepancy, commonly called the Sim-to-Real gap (McMahan et al., 2024), can arise from various sources, including environmental non-stationarity, inherent modeling errors, exogenous disturbances, or even adversarial attacks (Bukharin et al., 2023; Ma et al., 2023). Consequently, policies derived from a misspecified model can exhibit significantly degraded performance when deployed.

To address this vulnerability, the framework of distributionally robust Markov Games (DR-MGs) has been developed (Zhang et al., 2020; Kardeş et al., 2011), extending the principles of distributionally robust Markov Decision Processes (MDPs) (Bagnell et al., 2001; Nilim & El Ghaoui, 2003; Iyengar, 2005) to the multi-agent setting. Instead of relying on a single, fixed MG model, the robust approach seeks to find an equilibrium that optimizes the *worst-case* performance over a predefined uncertainty set of plausible game models under potential model mismatches. The resulting robust equilibrium provides a performance guarantee across all models within this set, thereby ensuring each agent's resilience against model mismatch.

The nature of a DR-MG problem is fundamentally shaped by the optimality criterion used. Much of the prior work has concentrated on the finite-horizon (Ma et al., 2023; Shi et al., 2025; 2024; Jiao & Li, 2024; Li et al., 2025; Blanchet et al., 2023) or discounted-reward setting (Zhang et al., 2020; Kardeş et al., 2011), where the objective is to maximize the expected reward over a finite number of steps or a discounted sum of rewards.

In these formulations, the influence of future rewards diminishes exponentially, which often simplifies analysis as the results are less dependent on the long-term chain structure of the underlying game. However, despite the analytical elegance of these settings, policies learned under them can be suboptimal for multi-agent systems designed to operate over extended or indefinite time horizons, such as in warehouse robotics (Krnjaic et al., 2024), communication networks (Qu et al., 2020), autonomous vehicle coordination (Peake et al., 2020), financial markets (Vadori et al., 2024), or even peer-to-peer energy trading (Qiu et al., 2021). In these scenarios, the long-term average reward is a more natural and

---

[1]Department of Electrical and Computer Engineering, University of Central Florida, Orlando, Florida, USA. [2]Department of Computer Science, University of Central Florida, Orlando, Florida, USA. Correspondence to: Zachary Roch <zachary.roch@ucf.edu>, Yue Wang <yue.wang@ucf.edu>.

*Proceedings of the 43rd International Conference on Machine Learning*, Seoul, South Korea. PMLR 306, 2026. Copyright 2026 by the author(s).

meaningful performance measure. However, the study of DR-MGs under the average-reward criterion is nascent and fraught with complexities not present in the discounted or finite-horizon cases. The analysis of average-reward games hinges on the limiting behavior of the underlying stochastic process, making it markedly more intricate (Sobel, 1971). For instance, a general non-robust vanilla Markov game with average reward may not have a stationary Nash Equilibrium (Filar & Vrieze, 1996); the direct correspondence between stationary policies and the limit points of state-action frequencies, a cornerstone of discounted analysis, breaks down in the average-reward setting even for single-agent MDPs, except under restrictive structural assumptions (Puterman, 2014; Atia et al., 2021). This difficulty is compounded in the multi-agent context, where strategic interactions are intertwined with the uncertain, complex dynamics.

Moreover, none of the methods for DR-MGs under the discounted reward or finite horizon can be extended to the average-reward setting (also see our detailed discussion in Section 3). The existence of (non-stationary) Nash Equilibrium under finite horizon is derived through backward induction (Blanchet et al., 2023), which does not apply to infinite-horizon settings. The existence of a stationary Nash Equilibrium under the discounted reward criterion is derived in (Kardeş et al., 2011), which, however, heavily relies on the uniqueness of the solution to the discounted robust Bellman equation (Iyengar, 2005) and hence cannot be applied to the average reward setting. To date, the literature on average-reward DR-MGs remains sparse, leaving fundamental questions about the structure of robust equilibria and the design of convergent solution algorithms largely unanswered.

In this paper, we develop a comprehensive study of DR-MGs under the average-reward formulation. Our contributions are summarized as follows.

**Existence of Robust Nash Equilibrium:** We first study the solvability of average reward DR-MGs, i.e., the existence of robust NE. We first note that, under the irreducible setting, finding the best response policy of an agent is equivalent to finding the solution to the robust Bellman equation under average reward. Based on such an equivalence, we show that its solution set is convex and semi-continuous, which further implies the fixed point of the best-response mapping and the existence of a stationary Nash Equilibrium. We further study the more general weakly communicating DR-MGs, where such correspondence fails. We instead construct a proxy mapping based on the Bellman equation and show that it is a subset of the best-response mapping. After characterizing the convexity and continuity of the mapping, we prove the existence of NE. These results address the fundamental solvability question under these complex, robust, and long-term horizon competitive scenarios.

**Convergent Algorithms:** We then design concrete algorithms to find the robust NE in average reward DR-MGs. Inspired by the standard Nash value iteration approach, we first propose a robust Nash iteration algorithm. Under some additional assumptions on the game structure and a norm-form game NE computing oracle, we prove the convergence to a robust NE of our algorithm. Moreover, to bypass these assumptions and the computation oracle, we further develop an equivalent characterization of the NE in terms of temporal difference error, whose minimizer is shown to be a robust NE. We then design a two-time-scale algorithm to minimize the TD error. The algorithm does not require any NE oracle, and can be executed efficiently. We further develop its stationary convergence guarantee. Our studies hence provide the first comprehensive and algorithmic understanding of solving average-reward DR-MGs.

**Connection to Discounted Equilibria:** We further study the connection between the DR-MGs under average and discounted reward. We show that the robust NE under the average reward can be effectively approximated by the robust NE under the discounted reward with a discount factor large enough. Such a connection enables us to solve the complicated and challenging average reward DR-MGs through the discounted ones, utilizing their element mathematical properties and extensively developed algorithms.

## 2. Preliminaries and Problem Formulation

**Markov Decision Processes.** A Markov decision process (MDP) $(\mathcal{S}, \mathcal{A}, \mathsf{P}, r)$ is specified by: a state space $\mathcal{S}$, an action space $\mathcal{A}$, a transition kernel $\mathsf{P} = \{p_s^a \in \Delta(\mathcal{S}), a \in \mathcal{A}, s \in \mathcal{S}\}$[1], where $p_s^a$ is the distribution of the next state over $\mathcal{S}$ upon taking action $a$ in state $s$ (with $p_{s,s'}^a$ denoting the probability of transitioning to $s'$), and a reward function $r : \mathcal{S} \times \mathcal{A} \to [0, 1]$. At each time step $t$, the agent at state $s_t$ takes an action $a_t$, at which point the environment transitions to the next state $s_{t+1}$ according to $p_{s_t}^{a_t}$, and produces a reward signal $r(s_t, a_t) \in [0, 1]$ to the agent. We write $r_t = r(s_t, a_t)$ for convenience.

A stationary policy $\pi : \mathcal{S} \to \Delta(\mathcal{A})$ is a distribution over $\mathcal{A}$ for any given state $s$, meaning the agent takes action $a$ at state $s$ with probability $\pi(a|s)$. The goal of an MDP is to maximize the cumulative reward. The more extensively studied criterion is the discounted reward, whose objective function is the discounted value function: $V_\mathsf{P}^\pi(s) \triangleq \mathbb{E}_{\pi, \mathsf{P}} \left[ \sum_{t=0}^\infty \gamma^t r_t | S_0 = s \right]$, where $\gamma \in [0, 1)$ is some discount factor. When long-term performance is of interest, the average reward is generally adopted as the criterion of choice. The average reward induced by following policy $\pi$ starting from state $s \in \mathcal{S}$ is defined as $g_\mathsf{P}^\pi(s) \triangleq \liminf_{n \to \infty} \mathbb{E}_{\pi, \mathsf{P}} \left[ \frac{1}{n} \sum_{t=0}^{n-1} r_t | S_0 = s \right]$.

---

[1]$\Delta(\mathcal{S})$: the $(|\mathcal{S}| - 1)$-dimensional probability simplex on $\mathcal{S}$.

**Markov Games.** Markov games (MGs) are the extension of MDPs under the multi-agent setting (Littman, 1994), specified by $(\mathcal{N}, \mathcal{S}, \mathcal{A} = \times_{i \in \mathcal{N}} \mathcal{A}_i, \{r_i\}_{i \in \mathcal{N}}, \mathsf{P})$, where $\mathcal{N} = \{1, 2, \dots, N\}$ is the set of all agents or players, $\mathcal{S}$ is the state space that all agents operate within, $\mathcal{A}_i$ denotes agent $i$'s action space. In an MG, both rewards and state transitions are determined by the joint actions of all agents. At some state $s$, all agents act simultaneously by taking a joint action $a = (a_1, \dots, a_N) \in \mathcal{A}$. Agent $i$ receives a reward of $r_i(s, a)$ for all $i \in \mathcal{N}$, and the next state is generated following the transition kernel $\mathsf{P}_s^a$.

In MGs, each agent $i$ has its own policy $\pi_i$, resulting in a joint (product) policy $\pi = (\pi_1, \dots, \pi_N)$. Each agent shares different goals as they have different reward functions to optimize their own value function/average reward based on their own reward and the joint policy: $V_{\mathsf{P},i}^\pi$ or $g_{\mathsf{P},i}^\pi$. Due to possible conflicting goals between agents, an MG aims to find some equilibria among all agents. In this work, we focus our study on the stationary Nash Equilibrium (NE). More specifically, a stationary NE is a product policy $\pi^\star = (\pi_1^\star, \dots, \pi_N^\star)$, such that no single agent can improve its objective function by deviating from it, while the others stick to $\pi^\star$:

$$V_{\mathsf{P},i}^{\pi^\star}(\rho) \geq V_{\mathsf{P},i}^{(\pi_{-i}^\star, \pi_i)}(\rho), \forall i \in \mathcal{N}, \forall \pi_i \in \Pi_i, \quad (1)$$

for some initial distribution $\rho$ and $V_{\mathsf{P},i}^\pi(\rho) \triangleq \mathbb{E}_{s \sim \rho}[V_{\mathsf{P},i}^\pi(s)]$, and $\Pi_i$ denotes the set of stationary policies for agent $i$, $\pi_{-i}^\star$ denotes the joint policy of *all* other agents *except* $i$ from $\pi^\star$, and $(\pi_{-i}^\star, \pi_i)$ is the joint policy. The existence of a stationary NE is derived for MGs with discounted reward (Fink, 1964) and average-reward MGs with additional structural assumptions (Filar & Vrieze, 1996).

**Distributionally Robust Markov Games.** Different from MGs, distributionally robust Markov games (DR-MGs) do not have a fixed transition kernel, but rather an uncertainty set $\mathcal{P}$ of transition kernels. We consider the standard $(s, a)$-rectangular uncertainty set $\mathcal{P} = \times_{(s,a) \in \mathcal{S} \times \mathcal{A}} \mathcal{P}_s^a$ (Iyengar, 2005; Nilim & El Ghaoui, 2003; Zhang et al., 2020; Blanchet et al., 2023; Shi et al., 2024), where the uncertainty set is independently defined over all $(s, a)$ pairs.

After all agents take their joint action $a$ at state $s$, the next state is determined by an arbitrary kernel $\mathsf{P}_s^a \in \mathcal{P}_s^a$. In DR-MGs, agents adopt the principle of pessimism and optimize for the worst-case performance, which is defined as the robust value function $V_{\mathcal{P},i}^\pi(\rho) \triangleq \min_{\mathsf{P} \in \mathcal{P}} V_{\mathsf{P},i}^\pi(\rho)$, and robust average reward $g_{\mathcal{P},i}^\pi(\rho) \triangleq \min_{\mathsf{P} \in \mathcal{P}} g_{\mathsf{P},i}^\pi(\rho)$. The notion of Nash Equilibrium is similarly defined in terms of worst-case performance. In this paper, we focus on DR-MGs with average reward whose Nash Equilibrium is some product policy $\pi^\star$ such that for some initial state distribution $\rho$,

$$g_{\mathcal{P},i}^{\pi^\star}(\rho) \geq g_{\mathcal{P},i}^{(\pi_{-i}^\star, \pi_i)}(\rho), \forall i \in \mathcal{N}, \forall \pi_i. \quad (2)$$

## 3. Hardness in Average-Reward DR-MGs

DR-MGs under finite-horizon or discounted reward are extensively studied recently (Zhang et al., 2020; Jiao & Li, 2024; Blanchet et al., 2023; Shi et al., 2025; Ma et al., 2023; Kardeş et al., 2011; McMahan et al., 2024), yet extending them to the average-reward setting is non-trivial. At a high level, two issues make the average-reward regime qualitatively more challenging: (1). the presence of a *player-specific environment* that selects worst-case transition kernels, which breaks the usual min–max duality even in discounted zero-sum games; and (2). The intricate structure of (robust) average reward, which depends sensitively on the chain decomposition and is far less tractable.

**Player-specific environment and failure of min–max duality.** In a standard (non-robust) Markov game with a fixed transition kernel $P$, the existence of equilibria in multi-player general-sum MGs was established by (Fink, 1964). In two-player zero-sum games with discounted reward, the analysis is built on a clean saddle-point structure. Let $r_1 = -r_2 = r$ and denote by $V_{P,r}^{(\pi_1, \pi_2)}(\rho)$ the $\gamma$-discounted value of player 1 under joint policy $(\pi_1, \pi_2)$ and initial distribution $\rho$. Then the classical result of (Shapley, 1953) shows that

$$\max_{\pi_1} \min_{\pi_2} V_{P,r}^{(\pi_1, \pi_2)}(\rho) = \min_{\pi_2} \max_{\pi_1} V_{P,r}^{(\pi_1, \pi_2)}(\rho), \quad (3)$$

and any maximizer/minimizer pair in (3) is a stationary discounted NE. Similar arguments, combined with the limit $\gamma \to 1$, underlie many existence results for non-robust average-reward Markov games; see, e.g., (Filar & Vrieze, 1996; Sennott, 1994; Rieder, 1995).

In the distributionally robust setting, the situation changes qualitatively. The model uncertainty is encoded by an ambiguity set $\mathcal{P}$ of transition kernels. Conceptually, this introduces a "player-specific environment" (rather than a monolithic adversarial third player) that aims to minimize the specific agent's payoff, potentially propelling the overall system into different recurrent classes depending on the agents' joint policy. Crucially, because different agents have different reward functions, in general their associated worst-case kernels need not coincide. Consequently, even in a two-player zero-sum game with strictly opposing rewards $(r_1 = -r_2 = r)$, the robust value functions are not strictly opposing $(V_{\mathcal{P},r_1} \neq -V_{\mathcal{P},r_2})$, forcing this to be treated as a general-sum game.

See that the robust counterpart of (3) splits into two distinct problems. From the perspective of player 1 (maximizer), the worst-case is governed by $\max_{\pi_1} \min_{\pi_2} \min_{P \in \mathcal{P}} V_{P,r}^{(\pi_1, \pi_2)}(\rho)$, while from the perspective of player 2 (minimizer), whose reward is $-r$, the worst case is governed by $\min_{\pi_2} \max_{\pi_1} \max_{P \in \mathcal{P}} V_{P,r}^{(\pi_1, \pi_2)}(\rho)$. In general, these

two values need not coincide:

$$\max_{\pi_1} \min_{\pi_2} \min_{P \in \mathcal{P}} V_{P,r}^{(\pi_1,\pi_2)}(\rho) \ \neq \ \min_{\pi_2} \max_{\pi_1} \max_{P \in \mathcal{P}} V_{P,r}^{(\pi_1,\pi_2)}(\rho),$$

meaning that in general-sum games, the min–max duality which guarantees a saddle-point in non-robust studies breaks down for DR-MGs. In fact, there is no saddle-point structure to exploit for discounted and zero-sum settings, and this problem is only exacerbated in general-sum or multi-player settings. Therefore, establishing the existence of an equilibrium is significantly harder as neither Shapley's fixed-point argument or standard discounted saddle-point theory can be directly reused to establish NE existence in robust games.

One might hope to recover average-reward NE by letting the discount factor $\gamma \to 1$ in robust discounted games, using uniform convergence of robust value functions to interchange the order of max, min, and lim (Wang et al., 2023b). However, the failure of min–max duality already at the discounted level implies that this limit approach still does not yield NE existence for average-reward DR-MGs, and it does not apply at all to multi-player general-sum games.

**Complicated structure of robust average reward and an impossibility example.** The second difficulty comes from the structure of the (robust) average reward itself. Even without robustness, Markov games under the average-reward criterion need not admit a stationary NE in general (Filar & Vrieze, 1996; Wikecek & Altman, 2015; Lozovanu, 2016; Thuijsman, 1997). The usual link between stationary policies and stationary/occupancy distributions breaks down except under additional structural assumptions such as irreducibility or unichain conditions (Puterman, 2014; Atia et al., 2021). This makes it substantially harder to reason about long-run behavior and to construct equilibria.

Robustness further complicates the picture. The robust average reward must account for the worst-case choice of transition kernels; the resulting robust Bellman equations are nonlinear, may admit multiple solutions, and their solvability and uniqueness are largely unresolved in prior work (Wang et al., 2023b;c; Wang & Si, 2025). In particular, existing algorithms for robust average-reward reinforcement learning either rely on incorrect Bellman solvability claims or bypass the average-reward Bellman equation altogether via discounted reductions. As a consequence, neither the techniques for discounted DR-MGs (Kardeş et al., 2011; Zhang et al., 2020) nor those for non-robust average-reward MGs (Guo & Yang, 2008; Hernández-Lerma & Lasserre, 2000; Guo & Hernández-Lerma, 2003; Zheng & Guo, 2024; Nowak, 1999; IWASE et al., 1976; Küenle & Schurath, 2003; Tanaka & Wakuta, 1977; Jaśkiewicz & Nowak, 2001; Wikecek & Altman, 2015) can be directly transplanted to distributionally robust average-reward games.

To illustrate that these issues are not merely technical, we show that, without any structural assumptions, average-reward DR-MGs may fail to admit *any* stationary NE.

**Lemma 3.1.** *There exists a finite-state, two-player zero-sum DR-MG under the average-reward criterion that admits no stationary robust Nash equilibrium.*

The construction uses a simple three-state, two-action zero-sum DR-MG. The underlying Markov chains induced by the ambiguity set are multi-chain, where the adversary can discontinuously alter the chain structure to force the agent into different recurrent classes based on changes in its policy.

Lemma 3.1 demonstrates that average reward DR-MGs are ill-posed without some structural assumptions. In the remainder of the paper, we therefore focus on two standard classes of average-reward problems where such a structure is present: in Section 4 we study *irreducible* DR-MGs, where every induced Markov chain under any stationary joint policy and any admissible kernel belongs to a single recurrent class; and in Section 5 we relax this to *weakly communicating* DR-MGs, which allow multiple recurrent classes but still admit a well-defined optimal robust average reward.

## 4. NE of Irreducible DR-MGs

As discussed above, the existence of a NE cannot be guaranteed without any structural assumptions. In this section, we first study a relatively easier setting, where the underlying DR-MG is assumed to be irreducible.

**Assumption 4.1.** For any deterministic joint policy $\pi$ and any transition kernel $\mathsf{P} \in \mathcal{P}$, the induced Markov chain $(\mathcal{S}, \mathcal{A}, \mathsf{P}^\pi)$ is irreducible, i.e., all states belong to a single recurrent class. Moreover, $\mathcal{P}$ is compact and convex.

*Remark* 4.2. Irreducibility is a standard assumption (slightly stronger than the unichain assumption (Wang et al., 2023b;c; 2024; Roch et al., 2025b;a)) in robust average reward studies (Xu et al., 2025b;a). Under this assumption, the robust average reward $g_{\mathcal{P}}^\pi(\rho) = g_{\mathcal{P}}^\pi(s)$ is always a constant over all initial states, ensuring the tractability of robust average reward. Hence in this section, we omit the initial state and interchangeably use $g_{\mathcal{P}}^\pi$ $g_{\mathcal{P}}^\pi(\rho)$, and $g_{\mathcal{P}}^\pi(s)$.

In non-robust average reward MG studies, the NE can be proved by studying the induced stationary/occupancy distribution (Wikecek & Altman, 2015). However, when considering robustness, the worst-case stationary distribution set becomes much more complicated. For example, it is non-convex (Wang et al., 2022a; Zhang et al., 2024b; Ma et al., 2025), whereas the convexity is necessary in non-robust proofs and hence cannot be adapted. We thus propose a direct approach by studying the best response mapping.

**Definition 4.3** (Best Response mapping). In an average-reward DR-MG, fix agent $i$ and the joint policy $\pi_{-i}$ of all other agents. The best response mapping of agent $i$ w.r.t.

$\pi_{-i}$ is a set-valued mapping defined as

$$BR_i(\pi_{-i}) \triangleq \left\{ \pi_i : g_{\mathcal{P},i}^{(\pi_{-i},\pi_i)} = \max_{\mu_i} g_{\mathcal{P},i}^{(\pi_{-i},\mu_i)} \right\}. \quad (4)$$

That is, the best response policy is agent $i$'s optimal policy when other agents fix their policies. A standard result in game theory literature (Osborne, 2003; Fudenberg & Tirole, 1991) is that, $\pi^\star$ is a Nash Equilibrium if and only if $\pi_i^\star \in BR_i(\pi_{-i}^\star), \forall i \in \mathcal{N}$, i.e., $\pi^\star$ is a fixed point of the BR mapping: $\pi^\star \in BR(\pi^\star) = (\times_i BR_i(\pi_{-i}))$. Hence, proving the existence of NE is equivalent to showing the BR mapping has a fixed point, which is generally obtained through the Kakutani fixed point theorem (Lemma D.4). Our studies will similarly follow this direction. We first show that it is sufficient to reduce to single-agent cases.

**Lemma 4.4.** *If for any $i \in \mathcal{N}$ and any product policy $\pi_{-i}$, $BR_i(\pi_{-i})$ is nonempty, convex, and $BR_i(\cdot)$ is upper semicontinuous (see Lemma D.4 for definitions), then the BR mapping has a fixed point, and there exists a robust NE.*

### 4.1. Induced Distributionally Robust MDPs

We then develop a framework of induced distributionally robust MDPs to provide a framework for multi-agent studies.

**Definition 4.5** (Induced Distributionally Robust MDP). Given a distributionally robust Markov game $\mathcal{MG} = (\mathcal{N}, \mathcal{S}, \mathcal{A}, \mathcal{P}, r)$ and an agent $i \in \mathcal{N}$. The induced distributionally robust MDP of the $\mathcal{MG}$ by a joint policy $\pi_{-i}$ is defined as $M_i(\pi_{-i}) \triangleq (\mathcal{S}, \mathcal{A}_i, \mathcal{P}^{\pi_{-i}}, r^{\pi_{-i}})$, where

$$(\mathcal{P}^{\pi_{-i}})_s^{a_i} \triangleq \left\{ \sum_{a_{-i} \in \mathcal{A}_{-i}} \pi_{-i}(a_{-i}|s)\mathsf{P} \,\middle|\, \mathsf{P} \in \mathcal{P}_s^{(a_i,a_{-i})} \right\},$$

$$r^{\pi_{-i}}(s, a_i) \triangleq \sum_{a_{-i} \in \mathcal{A}_{-i}} \pi_{-i}(a_{-i}|s)r_i(s, (a_i, a_{-i})). \quad (5)$$

Clearly, $M_i(\pi_{-i})$ is a single-agent distributionally robust MDP for agent $i$. Due to rectangularity of $\mathcal{P}$, $\mathcal{P}^{\pi_{-i}}$ is $(s, a_i)$-rectangular, and $\sum_{a_{-i}} \pi_{-i}(a_{-i}|s)\sigma_{\mathcal{P}_s^{(a_i,a_{-i})}}(V) = \sigma_{(\mathcal{P}^{\pi_{-i}})_s^{a_i}}(V)$, where $\sigma_{\mathcal{P}_s^a}(V) \triangleq \min_{\mathsf{P} \in \mathcal{P}_s^a} \mathsf{P}V$ is the support function for a specific state-action pair. Moreover, per Assumption 4.1, $M_i(\pi_{-i})$ is also compact and irreducible, which reduces the DR-MG to a single-agent problem.

However, even the single-agent average reward robust RL is not fully studied. Specifically, the most essential tool, its robust Bellman equation, is not well studied: its solvability or characterizations of solutions are not yet justified (an incorrect proof is given in (Wang et al., 2023b) for the single-agent setting), therefore making the connection between optimal policies and the robust Bellman equation unclear. Although a huge body of existing work develops convergent algorithms for single-agent robust average-reward MDPs,

their convergence guarantees either rely on the incorrect result in (Wang et al., 2023b), e.g., (Wang et al., 2023c; Roch et al., 2025a), or bypass its solvability through discounted proxy (Xu et al., 2025b;a; Chatterjee et al., 2024; Meggendorfer et al., 2025). Moreover, there are no prior studies on the uniqueness of these solutions. We hence derive the following foundational results under Assumption 4.1 (proofs are deferred to Section D).

**Theorem 4.6.** *Under Assumption 4.1. (1). Denote any worst-case transition kernel w.r.t. $g_\mathsf{P}^\pi$ by $\mathsf{P}_w \triangleq \arg\min_{\mathsf{P} \in \mathcal{P}} g_\mathsf{P}^\pi$. Then the gain/bias of $(\pi, \mathsf{P}_w) : (g_{\mathsf{P}_w}^\pi, h_{\mathsf{P}_w}^\pi)$ is the unique solution to the equation:*

$$V(s) = \sum_a \pi(a|s)(r(s, a) - g + \sigma_{\mathcal{P}_s^a}(V)). \quad (6)$$

*The uniqueness is in the sense that, for any solution $(g, V)$ to (6), it holds that $g = g_\mathcal{P}^\pi, V = h_{\mathsf{P}_w}^\pi + c\boldsymbol{e}$, for some constant $c$ and $\boldsymbol{e} = (1, ..., 1)$.[2]*

*(2). The robust Bellman optimality equation*

$$V(s) = \max_a \{r(s, a) - g + \sigma_{\mathcal{P}_s^a}(V)\} \quad (7)$$

*has a unique solution (up to some vector $c\boldsymbol{e}$).*

Our results therefore prove the solvability and uniqueness of the robust Bellman equation, especially (6), solidifying the connection between the robust average reward and its solutions. It allows us to evaluate any policy by solving a (nonlinear) Bellman equation, while simultaneously certifying the existence of a worst-case kernel realizing the robust average reward. Then, the robust Bellman optimality equation provides a complete characterization of optimality in irreducible DR-MDPs.

We further highlight that our results are fundamental to characterizing the complicated structure of the average reward. For instance, it is discussed in (Wang et al., 2023c) that (6) may have non-unique solutions even up to constant vectors, which we rule out under Assumption 4.1. Our proof techniques are also different from the non-robust ones, which rely on the Laurent series of the non-robust average reward (Puterman, 2014), whereas we directly show that the constructed pair $(g_{\mathsf{P}_w}^\pi, h_{\mathsf{P}_w}^\pi)$ is a solution.

### 4.2. Existence of a Nash Equilibrium

With the results derived for single-agent robust Bellman equations, we can then show the following results.

**Theorem 4.7.** *(1). For any distributionally robust MDP under Assumption 4.1, the optimal policy set $\arg\max_\pi g_\mathcal{P}^\pi$ is convex. (2). For any convergent policy sequences*

---

[2]Even if there exist multiple worst-case kernels, their gain/bias are all identical up to $c\boldsymbol{e}$. We denote this unique solution by $h_\mathcal{P}^\pi$.

$\{\pi_{-i}^n\} \to \pi_{-i}$ and $\{\mu_i^n\} \to \mu_i$, if $\mu_i^n$ is an optimal robust policy of the induced distributionally robust MDP $(\mathcal{S}, \mathcal{A}_i, \mathcal{P}^{\pi_{-i}^n}, r^{\pi_{-i}^n})$ for agent $i \in \mathcal{N}$, then $\mu_i$ is also optimal for $(\mathcal{S}, \mathcal{A}_i, \mathcal{P}^{\pi_{-i}}, r^{\pi_{-i}})$.

As $\mathrm{BR}_i(\pi_{-i})$ is the set of optimal robust policies of the (induced) distributionally robust MDP $(\mathcal{S}, \mathcal{A}_i, \mathcal{P}^{\pi_{-i}}, r^{\pi_{-i}})$, this result hence directly implies the convexity of $\mathrm{BR}_i(\pi_{-i})$. And for (2), the inducing policy $\pi_{-i}^n$ varies, resulting in a distributionally robust MDP sequence with different dynamics. Hence, we need to reveal the equicontinuity and uniform convergence of induced distributionally robust MDPs. Similarly, this proof relies on Theorem 4.6.

We highlight that the proof is **not** a direct extension from its counterpart under the discounted reward, as we need to tackle the non-contraction of the Bellman equation, whereas the discounted Bellman equation is a contraction and admits a unique solution. Also, compared to non-robust studies, we need to additionally tackle the non-linearity and uncertainty transition kernel introduced by the worst-case consideration. We also highlight that our proof heavily relies on the solvability and uniqueness (up to constant vectors) of (6), which enables us to fully characterize the optimal policies through the robust Bellman optimality equation. For instance, for convexity, when interpolating between two optimal policies $\pi_1$ and $\pi_2$, the corresponding worst-case kernel $P^w$ could shift and differ from the ones of $\pi_1$ and $\pi_2$. Our results, however, ensure that the intersection of the optimality half-spaces remains convex, despite the kernel shift and the concave nature of the robust operator.

Combining all of the above results, we can finally show the existence of a Nash Equilibrium, which is the most fundamental property of the game.

**Theorem 4.8** (Existence of a Nash Equilibrium). *Under Assumption 4.1, a robust Nash equilibrium exists.*

## 5. NE of Weakly Communicating DR-MGs

In this section, we extend our existence result from irreducible DR-MGs to a more general class of *weakly communicating* DR-MGs. The weakly communicating assumption is known to be the weakest standard structural assumption under which efficient algorithms and sharp structural results are available for average-reward control; see, e.g., (Wikecek & Altman, 2015; Bertsekas, 2007; Wan et al., 2021).

**Assumption 5.1** (Weakly communicating DR-MG). The uncertainty set $\mathcal{P}$ is convex and compact. Moreover, for any $i$ and $\pi_{-i}$ and any transition kernel $P \in \mathcal{P}$, the state space can be decomposed as $\mathcal{S} = \mathcal{S}_0^i \cup \mathcal{S}_1^i$, with $\mathcal{S}_0^i \cap \mathcal{S}_1^i = \emptyset$, such that: (1). Every state in $\mathcal{S}_0^i$ is transient under *every* stationary joint policy; (2). For any two states $s, s' \in \mathcal{S}_1^i$, there exist a stationary policy $\pi_i$ such that $\mathbb{P}(S_N = s' \mid S_0 = s, (\pi_i, \pi_{-i}), P) > 0$ for some $N$. Additionally, for

any $\pi_{-i}$, the corresponding optimal robust Bellman equation (7) has a unique solution (up to constant shifts).

*Remark* 5.2. The uniqueness of the optimal bias function (up to a constant shift) is a standard requirement in average-reward settings (Wan et al., 2021). We emphasize that this condition *only* applies to the *optimal* robust Bellman equation, not the Bellman equation for every policy $\pi$; thus it is **strictly weaker** than the unichain assumption. A weakly communicating robust MDP can fail to be unichain while still having a unique optimal bias function. This uniqueness is critical as it ensures our proxy map $G_i(\pi_{-i})$ (defined subsequently) is well-defined and upper hemicontinuous (proved in the Appendix), preventing discontinuous jumps in the maximizer sets $M_s(\pi_{-i})$ that would occur if multiple distinct bias solutions existed.

As in the non-robust case, weakly communicating is strictly weaker than irreducibility: it only requires accessibility under *some* policy, rather than under all joint policies. Under Assumption 5.1, only the *optimal* robust average reward $g_{\mathcal{P}}^\star(s)$ is constant across initial states, whereas the robust average reward $g_{\mathcal{P}}^\pi(s)$ of a general stationary policy $\pi$ may depend on $s$ (Puterman, 2014; Atia et al., 2021; Wang & Si, 2025; Feinberg & Shwartz, 2012). The solvability guarantees for weakly communicating robust Bellman equations are weaker (Wan et al., 2021; Wan & Sutton, 2022; Wan et al., 2024). There is no understanding on the Bellman equation for a fixed policy in (6), and the solvability of the *robust Bellman optimality equation* was recently established by (Wang & Si, 2025) as follows.

**Lemma 5.3.** *(Wang & Si, 2025) Under Assumption 5.1, the optimal robust Bellman equation*

$$V(s) = \max_{a \in \mathcal{A}} \left\{ r(s,a) - g + \sigma_{\mathcal{P}_s^a}(V) \right\}, \qquad s \in \mathcal{S}, \ (8)$$

*admits a solution $(g^\star, h^\star)$, with $g^\star = g_{\mathcal{P}}^\star(\rho)$ equal to the optimal robust average reward for any initial distribution $\rho$.*

Compared with the irreducible case (Theorem 4.6), these results are strictly weaker. Theorem 4.6 gives solvability of the robust Bellman equation for *every* stationary policy $\pi$, as well as a tight correspondence between optimal policies and solutions of the optimal Bellman equation. Under Assumption 5.1, Lemma 5.3 guarantees solvability only for the *optimal* equation (8); The equation associated with a fixed policy $\pi$ may fail to have a solution, and one can no longer characterize optimal policies solely via the Bellman equations. As a consequence, the proof strategy of irreducible DR-MGs cannot be directly reused.

*Remark* 5.4. The uniqueness part in Assumption 5.1 is also assumed in non-robust settings, e.g., (Wan et al., 2021), which is used to ensure convergence stability. Notably, the commonly used unichain assumption (Wang et al., 2023b;c) satisfies this uniqueness (see Lemma E.1).

To study the NE of weakly communicating DR-MGs, we construct a proxy set-valued map, derived from solutions of the optimal robust Bellman equation, and show that this map is contained in the true best-response map. Fix a player $i$ and a stationary joint policy $\pi_{-i}$ of all other players, and consider the induced single-agent DR-MDP $\mathsf{M}_i(\pi_{-i}) = (\mathcal{S}, \mathcal{A}_i, \mathcal{P}^{\pi_{-i}}, r_i^{\pi_{-i}})$. Applying Lemma 5.3 to $\mathsf{M}_i(\pi_{-i})$ yields a unique solution $(g_{\pi_{-i}}^\star, h_{\pi_{-i}}^\star)$ of the optimal Bellman equation, with $g_{\pi_{-i}}^\star$ equal to the optimal robust average reward.

For each $s \in \mathcal{S}$ and any $\nu \in \Delta(\mathcal{A}_i)$ of player $i$, we define the one-step Bellman functional $F_s^{\pi_{-i}}(\nu)$ as

$$\sum_{a_i \in \mathcal{A}_i} \nu(a_i)\Big(r_i^{\pi_{-i}}(s, a_i) - g_{\pi_{-i}}^\star + \sigma_{(\mathcal{P}_{s,a_i}^{\pi_{-i}})}(h_{\pi_{-i}}^\star)\Big), \quad (9)$$

where $r_i^{\pi_{-i}}(s, a_i)$ and $(\mathcal{P}_{s,a_i}^{\pi_{-i}})$ is the induced uncertainty slice at $(s, a_i)$. Intuitively, $F_s^{\pi_{-i}}(\nu)$ is the robust one-step Bellman return at state $s$ when player $i$ plays $\nu(\cdot \mid s)$ and the others follow $\pi_{-i}$. We then define the *Bellman-greedy* response set of player $i$ w.r.t. $\pi_{-i}$ by

$$G_i(\pi_{-i}) \triangleq \Big\{\nu \in \Pi_i \mid \nu(s) = \arg \max_{\mu \in \Delta(\mathcal{A}_i)} F_s^{\pi_{-i}}(\mu), \forall s\Big\}.$$

Thus, $G_i(\pi_{-i})$ consists of stationary policies that are point-wise maximizers of the optimal Bellman right-hand side for the induced DR-MDP $\mathsf{M}_i(\pi_{-i})$.

Different from the irreducible case, we show a slightly weaker result $G_i(\pi_{-i}) \subseteq \mathrm{BR}_i(\pi_{-i})$, i.e., $G_i(\pi_{-i})$ is a *subset* of the best-response set, which is sufficient to establish existence of Nash equilibria.

Finally, define the joint map $G(\pi) \triangleq \times_{i \in \mathcal{N}} G_i(\pi_{-i})$ on the product policy space. If each $G_i$ satisfies the conditions of Kakutani's fixed-point theorem (Lemma D.4), then $G$ admits a fixed point $\pi^\star \in G(\pi^\star)$, and the inclusion $G_i(\pi_{-i}) \subseteq \mathrm{BR}_i(\pi_{-i})$ guarantees that this $\pi^\star$ is also a fixed point of the best-response map BR and hence a robust NE.

We summarize our main results in the next theorem, whose proofs are deferred to Appendix E.

**Theorem 5.5** (NE of weakly communicating DR-MGs).
*Suppose the average-reward DR-MG satisfies Assumption 5.1, then it has a stationary robust Nash equilibrium.*

Our results hence imply that DR-MGs with an average reward are also solvable under the weakly communicating settings. Different from irreducible cases, our proof techniques are fundamentally different and rely on the strong duality and span bound developed in (Wang & Si, 2025), and are deferred to Appendix E.

# 6. Algorithmic Solutions

In this section, we develop algorithmic solutions for irreducible average-reward DR-MGs under Assumption 4.1.

Notably, finding a stationary NE is significantly challenging in general-sum games, even under non-robust settings (Hu & Wellman, 2003; Li et al., 2007; Li, 2003; Zinkevich et al., 2005; Deng et al., 2023; Daskalakis et al., 2023).

## 6.1. Nash Iteration

Value-iteration has been extensively studied in algorithmic Markov game studies (Hu & Wellman, 2003; Li et al., 2007; Li, 2003; Liu et al., 2021; Shi et al., 2024; Farhat et al., 2026). We hence first develop a robust Nash iteration algorithm (Algorithm 2) for average reward DR-MGs. Our algorithm generally extends the standard Nash iteration algorithm to the robust average reward setting: it maintains an estimation, $Q_i$, for all agents, and in each step, the algorithm finds a Nash Equilibrium for the matrix-form game $\{Q_i(s, a, h_i^0)\}, i \in \mathcal{N}$ through an oracle **NE**. Notably, due to the non-uniqueness of Nash Equilibria and the complicated structure of general-sum MGs, the algorithm is not guaranteed to converge if an arbitrary oracle **NE** is used, even in non-robust MGs (Li, 2003; Hu & Wellman, 2003; Littman, 2001; Bowling & Veloso, 2001; Bowling, 2000). Notably, these assumptions hold for a variety of games, including potential Markov games (Leonardos et al., 2022) or cooperative DR-MGs (Wang et al., 2022b). We thus adopt additional assumptions and derive the guarantee.

**Theorem 6.1.** *Under Assumption 4.1 and Assumption G.1 (in Appendix), Algorithm 2 converges to a robust NE.*

## 6.2. Robust TD Descent Algorithm

Despite the convergence guarantee we obtained in Theorem 6.1, the algorithm suffers from two disadvantages. Firstly, the convergence is only guaranteed with additional assumptions, which are standard but may not hold generally (Filar & Vrieze, 1996); Moreover, in each step, Algorithm 2 solves for a NE of a matrix game, which can be PPAD-complete in the worst case (Daskalakis et al., 2009; Chen et al., 2009). Although assuming such an oracle is also standard in Markov game studies (Liu et al., 2021; Hu & Wellman, 2003; Shi et al., 2024), we are motivated to find other algorithms with better computational complexity.

In this section, we propose another algorithm to address these two disadvantages. Our algorithm is based on an equivalent characterization of the robust NE in terms of the temporal difference (TD) errors, inspired by non-robust studies of MGs (Sahabandu et al., 2024; Sobel, 1971). Specifically, given a joint policy $\pi$ and any $(g_k, V_k) \in \mathbb{R} \times \mathbb{R}^{\mathcal{S}}$, define the $k$-th player's TD error as

$$\Omega_k^R(s, a_k; \pi_{-k} \mid g_k, V_k) \triangleq g_k + V_k(s) \quad (10)$$
$$- \sum_{a_{-k}} \pi_{-k}(a_{-k} \mid s)\Big(r_k(s, a_k, a_{-k}) + \sigma_{\mathcal{P}_s^{(a_k, a_{-k})}}(V_k)\Big);$$

And for $(\pi, g = (g_1, ..., g_N), V = (V_1, ..., V_N))$, define the

global robust TD gap $\Delta_{\mathrm{R}}(\pi, g, V)$ as

$$
\begin{aligned}
&\Delta_{\mathrm{R}}(\pi, g, V) \\
&\triangleq \sum_{k \in \mathcal{N}, s \in \mathcal{S}, a_k} \pi_k(a_k \mid s)\, \Omega_k^R(s, a_k; \pi_{-k} \mid g_k, V_k), \quad (11)
\end{aligned}
$$

We then develop a TD-based characterization of robust NE as follows.

**Theorem 6.2.** *Denote* $\mathcal{L}(\pi) \triangleq \Delta_{\mathrm{R}}\big(\pi, g^\star(\pi), V^\star(\pi)\big)$, *where* $g^\star(\pi)_k$ *and* $V^\star(\pi)_k$ *is the solution to the optimal Bellman equation of the induced RMDP* $\mathsf{M}_k(\pi_{-k})$. *Then, for every* $\pi \in \Pi$, $\mathcal{L}(\pi) \geq 0$. *Moreover,* $\mathcal{L}(\pi) = 0$ *if and only if* $\pi$ *is a stationary robust Nash equilibrium.*

This result shows that the global robust TD gap, $\mathcal{L}$, plays the role of a potential function for robust Nash equilibria: its global minimizer is a NE. This viewpoint motivates using $\mathcal{L}$ as the objective of our TD-based algorithm: each iteration tries to reduce the global TD gap, and convergence to $\mathcal{L} = 0$ corresponds exactly to convergence to a robust Nash equilibrium. However, due to the worst-case consideration in robustness, the function $\mathcal{L}$ is generally non-smooth, whose analysis can be extremely challenging. Toward this, we propose a smoothed proxy $\mathcal{L}_\lambda$ of it by considering its Moreau envelop (Moreau, 1965; Wang et al., 2023a; 2025) (discussions are deferred to Section H.3). We note that when $\lambda \to 0$, the NE of the smoothed proxy approximates the robust NE of the original DR-MG, hence minimizing $\mathcal{L}_\lambda$ with a small $\lambda$ approximately finds a robust NE. Toward this, we develop a two-time-scale algorithm as follows.

---

**Algorithm 1** Robust TD Descent

---

1: **Input:** step sizes $\{\eta_n\}, \{\alpha_n\}$, smoothing $\lambda > 0$
2: Initialize $\pi^0 \in \Pi$, $(g^0, V^0) \in \mathcal{G} \times \mathcal{V}$
3: **for** $n = 0, 1, 2, \ldots$ **do**
4:  Fast scale: $(g^{n+\frac{1}{2}}, V^{n+\frac{1}{2}}) \leftarrow$ $\Gamma_{\mathcal{G} \times \mathcal{V}}\Big((g^n, V^n) - \alpha_n \nabla_{(g,V)} \mathcal{L}_\lambda(\pi^n, g^n, V^n)\Big)$
5:  Slow scale: $\pi^{n+1} \leftarrow \Gamma_\Pi\Big(\pi^n - \eta_n \nabla_\pi \mathcal{L}_\lambda(\pi^n, g^{n+\frac{1}{2}}, V^{n+\frac{1}{2}})\Big)$
6:  $(g^{n+1}, V^{n+1}) \leftarrow (g^{n+\frac{1}{2}}, V^{n+\frac{1}{2}})$
7: **end for**

---

In the algorithm, $\Gamma$ denotes the projection operator in Euclidean norm. A more detailed discussion of the algorithm is in Section H. Compared to Algorithm 2, which relies on a span-seminorm contraction analysis and requires a PPAD-complete NE computing subroutine, Algorithm 1 bypasses the NE oracle requirement entirely. While the joint action space computation grows exponentially in $N$, our framework can efficiently exploit factored action spaces or local interactions, since the effective space can be much

smaller. Furthermore, the inner-loop minimization over the ambiguity set ($\sigma_{\mathcal{P}_s^a}(V)$) is highly tractable for standard divergence-based uncertainty sets (e.g., $\mathcal{O}(|S| \log |S|)$ for KL-divergence via water-filling algorithm, or a closed-form solution in $\mathcal{O}(|S|)$ for Total-variation (Nilim & El Ghaoui, 2003)). Thus, the per-iteration complexity remains polynomial in the size of the DR-MG parameters (see Section H), making Algorithm 1 significantly more practical and feasible.

We then present our convergence results.

**Theorem 6.3.** *Assume the DR-MG satisfies Assumption 4.1 and step sizes satisfy the Robbins-Monro condition. Then* $\{\pi^n\}$ *weakly converges to a stationary point of* $\mathcal{L}_\lambda$.

Our results imply the convergence of Algorithm 1 to a stationary point of $\mathcal{L}_\lambda$ (note that any NE is a stationary point). Such a result is weaker than the convergence guarantee of Algorithm 2, yet it does not require any additional assumptions on the game structure or NE-computing oracle, hence standing as the first practical algorithm for average reward DR-MGs with provable convergence guarantees.

*Remark* 6.4. We highlight that, finding the exact NE even in standard general-sum Markov games is significantly challenging without additional structural assumptions (Zhang et al., 2024a; Deng et al., 2023; Jin et al., 2023). Unlike standard non-robust Markov games, the set of stationary points for DR-MGs does not necessarily coincide with the set of robust Nash Equilibria due to the inherent non-smoothness of the worst-case objective. Because the robust value function involves a $\min$ over the uncertainty set, gradient domination does not hold in general for DR-MGs.

However, any global minimizer of $\mathcal{L}_\lambda$ (with value 0) is an exact NE of the $\lambda$-smoothed robust game. As $\lambda \to 0$ in the original game, the smoothed TD errors converge to the original errors. Thus, for a sufficiently small $\lambda$, an approximate NE of the smoothed game translates to an $\mathcal{O}(\lambda + \epsilon)$-approximate NE of the original DR-MG. Our robust TD algorithm converges to a stationary point of $\mathcal{L}_\lambda$, which serves as a necessary first step toward finding a robust NE in robust average-reward environments. We leave explorations of local gradient domination for the smoothed game under strict equilibrium assumptions for stronger convergence guarantees as future work.

## 7. Connections with Discounted DR-NE

In this section, we study the connection between distributionally robust games with discounted and average rewards. Our motivation is to utilize the extensively studied and mathematical elegance of the discounted setting to solve the more challenging average-reward setting. Recall that a policy $\pi_\gamma^\star$

is a Nash Equilibrium of the $\gamma$-discounted DR-MG if

$$V_{\mathcal{P},i}^{\pi^\star}(\rho) \geq V_{\mathcal{P},i}^{(\pi_{-i}^\star, \pi_i)}(\rho), \forall i \in \mathcal{N}, \forall \pi_i, \quad (12)$$

where $V_{\mathcal{P},i}^{\pi^\star}(\rho)$ is the $\gamma$-discounted robust value function. A policy $\pi^\star$ is an $\epsilon$-Nash Equilibrium, if

$$g_{\mathcal{P},i}^{\pi^\star}(\rho) + \epsilon \geq g_{\mathcal{P},i}^{(\pi_{-i}^\star, \pi_i)}(\rho), \forall i \in \mathcal{N}, \forall \pi_i, \quad (13)$$

We then derive the following results under Assumption 4.1.

**Theorem 7.1.** *(1). For any discount factor $\gamma \in [0,1)$, denote a Nash Equilibrium of the $\gamma$-discounted DR-MG by $\pi_\gamma$. Then any cluster point $\pi$ of $\{\pi_\gamma\}_{\gamma \in [0,1)}$ is a Nash Equilibrium of the average reward DR-MG.*

*(2). For any $\epsilon$, there exists some discount factor $\gamma$, such that any $\epsilon$-Nash Equilibrium of the $\gamma$-discounted DR-MG is also an $\mathcal{O}(\epsilon)$-Nash Equilibrium under the average reward.*

This theorem establishes a crucial bridge between the discounted and average reward criteria in DR-MGs. Specifically, we showed that the worst-case long-run average behavior of a multi-agent system can be approximated by its discounted behavior when the future is valued almost as much as the present (i.e., as $\gamma \to 1$). Moreover, we provide a concrete choice of such a factor $\gamma$ in Section 7 to validate the approximation approach. It also provides significant computational implications. The analysis of $\gamma$-discounted games is often more tractable (Kardeş et al., 2011), benefiting from properties like the Banach fixed-point theorem, which guarantees the existence and uniqueness of value functions. Many reinforcement learning and game theory algorithms, such as Q-learning and policy gradient, can be applied for the discounted setting (Zhang et al., 2020). This result therefore assures us that for any desired level of accuracy $\epsilon$, we can simply select a single discount factor $\gamma$ sufficiently close to 1 and solve for this $\gamma$-discounted game.

## 8. Conclusion

In this paper, we developed a comprehensive framework for distributionally robust Markov games under the average-reward criterion, a setting critical for multi-agent systems requiring long-term reliability in the face of severe model uncertainty. We proved the existence of a stationary Nash Equilibrium under both irreducible and weakly communicating settings, and proposed two practical algorithms as the first provably convergent methods for average reward DR-MGs. We also connected average-reward equilibria to their discounted counterparts, enabling their tractable approximation via robust equilibria with large discount factors.

**Limitations and Future Works.** Our complete theoretical framework and algorithmic blueprint opens several interesting directions for future study. First, exploring local

gradient domination under a strict equilibrium assumption could yield local finite-time convergence guarantees for the smoothed objective. Second, extending this framework beyond $(s, a)$-rectangularity would allow us to handle state-coupled ambiguity. Finally, while our formulation models uncertainty as a player-specific environment, investigating fully shared worst-case kernels (e.g., robust cooperative games or robust Stackelberg games, where nature acts as adversarial leader) presents a fascinating challenge for future research on multi-agent robust optimization.

## Acknowledgments

This work was supported by DARPA under Agreement No. HR0011-24-9-0427, and an Amazon Research Award, Fall 2025. Any opinions, findings, and conclusions or recommendations expressed in this material are those of the author(s) and do not reflect the views of DARPA or Amazon.

## Impact Statement

This paper presents work whose goal is to advance the field of Machine Learning, specifically robust reinforcement learning. There are many potential societal consequences of our work; however, since we provide a general framework, we do not feel that there is a need to highlight these.

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

## A. Related Works

We discuss related works in this section.

**Non-robust Markov Games.** Markov Games (Littman, 1994) provide a foundational mathematical framework for multi-agent sequential decision-making. The majority of early work focused on two-player zero-sum MGs, especially under the discounted-reward setting. Under these settings, the existence of a stationary Nash Equilibrium was established in (Shapley, 1953) by formulating the problem as a Max-Min problem. It is later extended to multi-player general-sum MGs in (Fink, 1964). The average-reward criterion, while more suitable for systems with long operational horizons, presents greater analytical challenges. Unlike the discounted case, a stationary NE is not generally guaranteed to exist, even in two-player zero-sum games (Filar & Vrieze, 1996). Instead, NE existence has only been proven under some additional assumptions on the structure of the game, including irreducibility, zero-sum (Guo & Yang, 2008; Hernández-Lerma & Lasserre, 2000; Guo & Hernández-Lerma, 2003; Zheng & Guo, 2024; Nowak, 1999; IWASE et al., 1976; Küenle & Schurath, 2003; Tanaka & Wakuta, 1977; Jaśkiewicz & Nowak, 2001; Filar & Vrieze, 1996), or single recurrent class under any policy (Sobel, 1971; Sahabandu et al., 2024). However, these non-robust methods cannot be applied to our robust setting.

**Distributionally Robust MDPs.** To address the performance degradation that occurs when a model is mis-specified, (single-agent) distributionally robust MDPs are first developed in (Iyengar, 2005; Nilim & El Ghaoui, 2003) for both finite horizon and infinite horizon discounted reward settings. Studies under the average reward setting are limited until recently. In (Wang et al., 2023b;c), fundamental understandings of robust average reward are developed under the unichain assumption, and is later extended to more general settings like weakly communicating ones (Grand-Clement et al., 2023; Wang & Si, 2025). A huge body of robust reinforcement learning for average reward is also developed, mainly focusing on the sample complexity (Xu et al., 2025b;a; Roch et al., 2025b;a; Chen et al., 2025; Grand-Clément & Petrik, 2023; Chatterjee et al., 2024; Wang et al., 2024). However, single-agent robust MDPs with average reward are still not fully studied (e.g., uniqueness or solvability of robust Bellman equation are not clear). These works are all developed for single-agent settings, and do not address the challenges we faced in multi-agent DR-MGs.

**Distributionally Robust Markov Games.** Extending distributional robustness to the multi-agent setting is a recent and active area of research. However, the literature has overwhelmingly concentrated on the finite-horizon (Ma et al., 2023; Shi et al., 2025; 2024; Jiao & Li, 2024; Li et al., 2025; Blanchet et al., 2023; Farhat et al., 2026) and discounted-reward criteria (Zhang et al., 2020; Kardeş et al., 2011). These settings are often more analytically tractable because the influence of future rewards diminishes, simplifying the analysis. However, none of these methods can be extended to the average-reward setting, as we discussed earlier.

## B. Preliminaries

In this section, we briefly review previous studies of single-agent distributionally robust MDPs.

### B.1. Discounted Setting

Under the robust setting, the worst-case performance over the uncertainty set of MDPs is defined through the discounted reward. More specifically, the robust discounted value function of a policy $\pi$ for a discounted MDP is defined as

$$V_{\mathcal{P},\gamma}^{\pi}(s) \triangleq \min_{\kappa \in \times_{t \geq 0} \mathcal{P}} \mathbb{E}_{\pi,\kappa} \left[ \sum_{t=0}^{\infty} \gamma^t r_t | S_0 = s \right], \tag{14}$$

where $\kappa = (\mathsf{P}_0, \mathsf{P}_1 ...) \in \times_{t \geq 0} \mathcal{P}$.

For robust discounted MDPs, it has been shown that the robust discounted value function is the unique fixed-point of the robust discounted Bellman operator (Nilim & El Ghaoui, 2003; Iyengar, 2005; Puterman, 2014):

$$\mathbf{T}_{\pi} V(s) \triangleq \sum_{a \in \mathcal{A}} \pi(a|s) \left( r(s,a) + \gamma \sigma_{\mathcal{P}_s^a}(V) \right), \tag{15}$$

where $\sigma_{\mathcal{P}_s^a}(V) \triangleq \min_{p \in \mathcal{P}_s^a} p^\top V$ is the support function of $V$ on $\mathcal{P}_s^a$. Based on the contraction of $\mathbf{T}_{\pi}$, robust dynamic programming approaches, e.g., robust value iteration, can be designed (Nilim & El Ghaoui, 2003; Iyengar, 2005).

The goal is to find the optimal robust policy:

$$\pi^\star = \arg\max_\pi V_{\mathcal{P},\gamma}^\pi(s), \tag{16}$$

and it is shown (Iyengar, 2005) that the optimal robust value function $V_{\mathcal{P},\gamma}^{\pi^\star}(s)$ satisfies the robust Bellman optimality equation

$$V(s) \triangleq \max_a \left\{ \pi(a|s)\left(r(s,a) + \gamma \sigma_{\mathcal{P}_s^a}(V)\right) \right\}. \tag{17}$$

### B.2. Average Reward Setting

For a fixed kernel $\mathsf{P}_0$, the average reward (or gain) and the bias are defined as

$$g_{\mathsf{P}_0}^\pi(s) \triangleq \liminf_{n\to\infty} \mathbb{E}_{\pi,\mathsf{P}_0}\left[\frac{1}{n}\sum_{t=0}^{n-1} r_t | S_0 = s\right] \tag{18}$$

and

$$h_{\mathsf{P}_0}^\pi(s) \triangleq \mathbb{E}_{\pi,\mathsf{P}_0}\left[\sum_{t=0}^{\infty}(r_t - g_{\mathsf{P}_0}^\pi)|s_0 = s\right]. \tag{19}$$

We can view equation (19) as the cumulative difference over time $t$ as $t \to \infty$. Defining $H_{\mathsf{P}_0}^\pi \triangleq (I - \mathsf{P}_0^\pi + \mathsf{P}_*^\pi)^{-1}(I - \mathsf{P}_*^\pi)$ as the deviation matrix of $\mathsf{P}_0^\pi$, then $h_{\mathsf{P}_0}^\pi = H_{\mathsf{P}_0}^\pi r_\pi$ by (Puterman, 2014). When the MDP is irreducible, $(g_{\mathsf{P}_0}^\pi, h_{\mathsf{P}_0}^\pi)$ is a solution to the following non-robust Bellman equation (Feinberg & Shwartz, 2012):

$$h_{\mathsf{P}_0}^\pi(s) = \mathbb{E}_\pi\left[r(s,a) - g_{\mathsf{P}_0}^\pi(s) + \sum_{s'\in\mathcal{S}} p_{s,s'}^a h_{\mathsf{P}_0}^\pi(s')\right]. \tag{20}$$

In the robust setting, we consider the robust average reward, defined as

$$g_{\mathcal{P}}^\pi(s) \triangleq \min_{P\in\mathcal{P}} \liminf_{n\to\infty} \mathbb{E}_{\pi,P}\left[\frac{1}{n}\sum_{t=0}^{n-1} r_t | S_0 = s\right]. \tag{21}$$

It is shown that under the unichain assumption, the robust average reward is independent from the initial state: $g_{\mathcal{P}}^\pi(s_1) = g_{\mathcal{P}}^\pi(s_2)$, for any $s_1, s_2$. The robust Bellman equation (for a policy $\pi$) and the robust optimal Bellman equation for the average reward setting are derived in (Wang et al., 2023b):

$$V(s) + g = \sum_a \pi(a|s)\left(r(s,a) + \sigma_{\mathcal{P}_s^a}(V)\right), \tag{22}$$

$$V(s) = \max_a \left\{r(s,a) - g + \sigma_{\mathcal{P}_s^a}(V)\right\}, \quad \forall s \in \mathcal{S}. \tag{23}$$

It is shown (Wang et al., 2023c) that if the equation (22) ((23)) has a solution $(g, V)$, then the solution $g$ is the robust average reward $g_{\mathcal{P}}^\pi$ (the optimal robust average reward $g_{\mathcal{P}}^* = \max_\pi g_{\mathcal{P}}^\pi$). However, whether these two equations are solvable is not studied but directly assumed.

## C. Proof of Lemma 3.1

We derive the result by constructing an example.

Fix $\varepsilon \in (0,1)$. Consider a two-player zero-sum distributionally robust Markov game

$$(\mathcal{S}, \mathcal{A}_1, \mathcal{A}_2, \mathcal{P}, r_1, r_2),$$

with state space $\mathcal{S} = \{s,\, G,\, B\}$, action sets $\mathcal{A}_1 = \{1, 2\}$ and $\mathcal{A}_2 = \{1, 2\}$, and rewards

$$r_1(G, a_1, a_2) = 1, \qquad r_1(B, a_1, a_2) = 0, \qquad r_1(s, a_1, a_2) = 0, \qquad r_2 = -r_1.$$

The ambiguity set consists of two kernels $p^{(1)}$ and $p^{(2)}$:

$$p^{(1)}(\,\cdot\,|s, (1, 1)) = \delta_G, \qquad\qquad\qquad p^{(1)}(\,\cdot\,|s, (2, 2)) = \varepsilon\,\delta_G + (1 - \varepsilon)\,\delta_B,$$
$$p^{(2)}(\,\cdot\,|s, (1, 1)) = \varepsilon\,\delta_G + (1 - \varepsilon)\,\delta_B, \qquad\qquad p^{(2)}(\,\cdot\,|s, (2, 2)) = \delta_G,$$
$$p^{(1)}(\,\cdot\,|s, (1, 2)) = \delta_B, \qquad\qquad\qquad p^{(1)}(\,\cdot\,|s, (2, 1)) = \delta_B,$$
$$p^{(2)}(\,\cdot\,|s, (1, 2)) = \delta_B, \qquad\qquad\qquad p^{(2)}(\,\cdot\,|s, (2, 1)) = \delta_B,$$

and, for $x \in \{G, B\}$, $p^{(1)}(\,\cdot\,|x, (a_1, a_2)) = p^{(2)}(\,\cdot\,|x, (a_1, a_2)) = \delta_x$ (absorbing). Thus, under any $p \in \{p^{(1)}, p^{(2)}\}$ the chain has two closed recurrent classes $\{G\}$ and $\{B\}$ (i.e., it is multi-chain).

Let the initial state be $S_0 = s$. For stationary policies $\pi_1 \in \Delta(\mathcal{A}_1)$, $\pi_2 \in \Delta(\mathcal{A}_2)$, we specify them as

$$\pi_1(1) = p \in [0, 1], \qquad \pi_2(1) = q \in [0, 1].$$

Because $G, B$ are absorbing and $r_1 \in \{0, 1\}$ on $\{G, B\}$, the average reward of player 1 equals the probability (under the chosen kernel) of transitioning from $s$ to $G$ in one step. Thus we have that

$$U(p, q) \triangleq \min_{k \in \{1, 2\}} \mathbb{P}_{p^{(k)}}\big(S_1 = G \,\big|\, S_0 = s,\, \pi_1, \pi_2\big)$$
$$= \min\big\{\, pq + \varepsilon(1 - p)(1 - q),\ \varepsilon\, pq + (1 - p)(1 - q) \,\big\}.$$

Then the robust average reward for player 1 is $U(p, q)$ and for player 2 is $-U(p, q)$.

**Lemma C.1.** *For $U$ defined above and any $\varepsilon \in (0, 1)$,*

$$\sup_{p \in [0,1]}\ \inf_{q \in [0,1]}\ U(p, q) = \frac{\varepsilon}{2}, \qquad \inf_{q \in [0,1]}\ \sup_{p \in [0,1]}\ U(p, q) = \frac{\varepsilon}{1 + \varepsilon}.$$

*Proof.* Fix $p \in [0, 1]$. For $q \in [0, 1]$, $U(p, q)$ is the pointwise minimum of two affine functions of $q$, hence concave in $q$. Therefore the infimum over $q$ is attained at an endpoint:

$$\inf_{q \in [0,1]} U(p, q) = \min\{\, U(p, 0),\, U(p, 1)\,\} = \min\{\varepsilon(1 - p),\ \varepsilon p\} = \varepsilon \min\{p, 1 - p\}.$$

Maximizing over $p$ gives

$$\sup_{p \in [0,1]}\ \inf_{q \in [0,1]}\ U(p, q) = \sup_{p \in [0,1]} \varepsilon \min\{p, 1 - p\} = \varepsilon \cdot \frac{1}{2} = \frac{\varepsilon}{2}.$$

Fix $q \in [0, 1]$. As a function of $p$, $U(p, q)$ is again the pointwise minimum of two affine functions, hence concave and piecewise linear with a single kink at the solution of equality:

$$pq + \varepsilon(1 - p)(1 - q) = \varepsilon p q + (1 - p)(1 - q) \iff p = 1 - q.$$

Thus

$$\sup_{p \in [0,1]} U(p, q) = \max\Big\{\, U(0, q),\, U(1, q),\, U(1 - q, q)\,\Big\} = \max\Big\{\, \varepsilon(1 - q),\, \varepsilon q,\, (1 + \varepsilon)q(1 - q)\,\Big\}.$$

Hence

$$\inf_{q \in [0,1]}\ \sup_{p \in [0,1]}\ U(p, q) = \min_{q \in [0,1]}\ \max\Big\{\, \varepsilon(1 - q),\, \varepsilon q,\, (1 + \varepsilon)q(1 - q)\,\Big\}.$$

Denote $f(q) \triangleq \max\Big\{\, \varepsilon(1 - q),\, \varepsilon q,\, (1 + \varepsilon)q(1 - q)\,\Big\}$. Then since $f(q) = f(1 - q)$, it suffices to only consider $q \in [0, 0.5]$. Clearly, the minimum of $f$ is achieved at $q^* = \frac{\varepsilon}{1 + \varepsilon}$, with $f(q^*) = \frac{\varepsilon}{1 + \varepsilon}$. $\qquad\square$

**Theorem C.2** (Nonexistence of a Nash equilibrium). *For the robust Markov game above, no (stationary stochastic) Nash equilibrium exists.*

*Proof.* In a two-player zero-sum game, a stationary Nash equilibrium $(p^\star, q^\star)$ is a saddle point:

$$U(p^\star, q) \geq U(p^\star, q^\star) \geq U(p, q^\star) \qquad \forall p, q \in [0,1],$$

which implies

$$\inf_q U(p^\star, q) = U(p^\star, q^\star) = \sup_p U(p, q^\star).$$

Consequently,

$$\sup_p \inf_q U(p, q) \geq \inf_q U(p^\star, q) = U(p^\star, q^\star) = \sup_p U(p, q^\star) \geq \inf_q \sup_p U(p, q),$$

so a saddle requires $\sup_p \inf_q U = \inf_q \sup_p U$. By Lemma C.1, $\sup_p \inf_q U = \varepsilon/2$ while $\inf_q \sup_p U = \frac{\varepsilon}{1+\varepsilon} > \frac{\varepsilon}{2}$, leading to a contradiction. Hence no saddle point exists and no Nash equilibrium exists. $\qquad\square$

# D. Irreducible DR-MGs

In this appendix we work in the single-agent setting and fix a stationary policy $\pi$. The state space $\mathcal{S}$ and action space $\mathcal{A}$ are finite. For each $(s, a)$, the local ambiguity set $\mathcal{P}_s^a \subset \Delta(\mathcal{S})$ is nonempty, compact and convex, and the overall ambiguity set is the rectangular product

$$\mathcal{P} = \times_{(s,a) \in \mathcal{S} \times \mathcal{A}} \mathcal{P}_s^a.$$

For each $P \in \mathcal{P}$, we denote by $P^\pi$ the induced Markov kernel on $\mathcal{S}$:

$$P^\pi(s' \mid s) = \sum_{a \in \mathcal{A}} \pi(a \mid s) P(s' \mid s, a).$$

Under Assumption 4.1, $P^\pi$ is irreducible for every stationary $\pi$ and for every $P \in \mathcal{P}$.

For fixed $P$ and $\pi$, the average reward (gain) and bias are defined as

$$g_P^\pi(s) = \liminf_{n \to \infty} \mathbb{E}^{\pi, P}\left[\frac{1}{n} \sum_{t=0}^{n-1} r_t \,\middle|\, S_0 = s\right], \qquad h_P^\pi(s) = \mathbb{E}^{\pi, P}\left[\sum_{t=0}^{\infty} \left(r_t - g_P^\pi\right) \,\middle|\, S_0 = s\right].$$

By irreducibility, $g_P^\pi(s)$ does not depend on $s$, so we write simply $g_P^\pi$; moreover $(g_P^\pi, h_P^\pi)$ is a solution to the (non-robust) Poisson/Bellman equation (Puterman, 2014):

$$g_P^\pi + h_P^\pi(s) = r^\pi(s) + \sum_{s' \in \mathcal{S}} P^\pi(s' \mid s) h_P^\pi(s'), \quad s \in \mathcal{S}, \tag{24}$$

unique up to an additive constant on the bias.

We define the robust average reward of $\pi$ as

$$g_{\mathcal{P}}^\pi \triangleq \min_{P \in \mathcal{P}} g_P^\pi.$$

Compactness of $\mathcal{P}$ and continuity of $P \mapsto g_P^\pi$ (see, e.g., (Puterman, 2014), Prop. 8.4.6) imply that the minimum is attained: there exists at least one *worst-case kernel*

$$P^w \in \arg\min_{P \in \mathcal{P}} g_P^\pi.$$

For each state $s$ and action $a$, define the local robust one-step operator

$$\sigma_s^a(v) \triangleq \min_{p \in \mathcal{P}_s^a} p^\top v, \qquad v \in \mathbb{R}^{\mathcal{S}},$$

and the $\pi$-averaged robust Bellman operator

$$\sigma_s^\pi(v) \triangleq \min_{P \in \mathcal{P}} \sum_{s' \in \mathcal{S}} P^\pi(s' \mid s) v(s') = \min_{p \in \mathcal{P}_s^\pi} p^\top v,$$

where $\mathcal{P}_s^\pi \triangleq \{\sum_a \pi(a \mid s) p^a : p^a \in \mathcal{P}_s^a\}$. We also use $r^\pi, \sigma^\pi(v)$ to denote $S$-dimensional vectors, with entry $r^\pi(s) = \sum_a \pi(a|s) r(s, a)$ and $\sigma^\pi(v)(s) = \sum_a \pi(a|s) \sigma_{\mathcal{P}_s^a}(v)$.

**Theorem D.1.** *Consider an uncertainty set $\mathcal{P}$ satisfying Assumption 4.1 and fix a stationary policy $\pi$. Let $P^w \in \arg\min_{P \in \mathcal{P}} g_P^\pi$ be any worst-case kernel, and let $(g_{Pw}^\pi, h_{Pw}^\pi)$ be the corresponding (non-robust) gain and bias, i.e. a solution to (24) with $P = P^w$. Then the pair $(g_{\mathcal{P}}^\pi, h_{Pw}^\pi)$ solves the robust Bellman equation*

$$g_{\mathcal{P}}^\pi + h(s) \;=\; r^\pi(s) + \sigma_s^\pi(h), \qquad s \in \mathcal{S}, \tag{25}$$

*and is unique up to addition of a constant to $h$. Equivalently, writing $V = h$ and rearranging, $V$ satisfies*

$$V(s) \;=\; \sum_{a \in \mathcal{A}} \pi(a \mid s)\big(r(s,a) - g_{\mathcal{P}}^\pi + \sigma_s^a(V)\big), \quad s \in \mathcal{S}.$$

*Proof.* Fix $\pi$ and let $P^w$ be any worst-case kernel for the robust average reward, so $g_{\mathcal{P}}^\pi = g_{Pw}^\pi$. By the irreducible average-reward MDP theory (Puterman, 2014), there exists a bias $h_{Pw}^\pi$ such that

$$g_{Pw}^\pi + h_{Pw}^\pi(s) \;=\; r^\pi(s) + \sum_{s'} P^{w\pi}(s' \mid s) h_{Pw}^\pi(s'), \qquad s \in \mathcal{S}. \tag{26}$$

Now fix any vector $v \in \mathbb{R}^{\mathcal{S}}$ and define a kernel $P^v \in \mathcal{P}$ by taking, independently for each state $s$,

$$P^{v,\pi}(\cdot \mid s) \in \arg\min_{p \in \mathcal{P}_s^\pi} p^\top v,$$

which exists by compactness of $\mathcal{P}_s^\pi$. In particular, for $v = h_{Pw}^\pi$ we define $P^V \triangleq P^{h_{Pw}^\pi}$ so that

$$\sigma_s^\pi(h_{Pw}^\pi) \;=\; \sum_{s'} P^{V\pi}(s' \mid s)\, h_{Pw}^\pi(s'), \qquad s \in \mathcal{S}.$$

By Assumption 4.1, $P^{V\pi}$ is irreducible.

For the kernel $P^V$, denote by $(g_{PV}^\pi, h_{PV}^\pi)$ the corresponding non-robust gain/bias pair. It satisfies

$$g_{PV}^\pi + h_{PV}^\pi(s) \;=\; r^\pi(s) + \sum_{s'} P^{V\pi}(s' \mid s) h_{PV}^\pi(s'), \qquad s \in \mathcal{S}. \tag{27}$$

Rewriting (26)-(27) in vector form yields

$$(\mathrm{I} - P^{w\pi})h_{Pw}^\pi = r^\pi - g_{Pw}^\pi \mathbf{e}, \qquad (\mathrm{I} - P^{V\pi})h_{PV}^\pi = r^\pi - g_{PV}^\pi \mathbf{e}.$$

Since $P^w$ is worst-case for $\pi$, we have $g_{Pw}^\pi \le g_{PV}^\pi$, and hence

$$(\mathrm{I} - P^{w\pi})h_{Pw}^\pi \;=\; r^\pi - g_{Pw}^\pi \mathbf{e} \;\ge\; r^\pi - g_{PV}^\pi \mathbf{e} \;=\; (\mathrm{I} - (P^V)^\pi)h_{PV}^\pi. \tag{28}$$

Equivalently,

$$(P^{w\pi} - \mathrm{I})h_{Pw}^\pi \;\le\; (P^{V\pi} - \mathrm{I})h_{PV}^\pi.$$

By the definition of $P^V$ as a minimizer for $h_{Pw}^\pi$ we also have, componentwise,

$$P^{V\pi} h_{Pw}^\pi \;\le\; P^{w\pi} h_{Pw}^\pi. \tag{29}$$

Let $x \triangleq h_{Pw}^\pi - h_{PV}^\pi$. Using (28) and adding/subtracting $P^{V\pi} h_{Pw}^\pi$ we obtain

$$\begin{aligned}
x = h_{Pw}^\pi - h_{PV}^\pi &\ge P^{w\pi} h_{Pw}^\pi - P^{V\pi} h_{PV}^\pi \\
&= P^{V\pi} h_{Pw}^\pi - P^{V\pi} h_{PV}^\pi + \big(P^{w\pi} h_{Pw}^\pi - P^{V\pi} h_{Pw}^\pi\big) \\
&= P^{V\pi} x + \big(P^{w\pi} h_{Pw}^\pi - P^{V\pi} h_{Pw}^\pi\big).
\end{aligned}$$

By (29) the last term is componentwise nonnegative, so

$$x \;\ge\; P^{V\pi} x.$$

Since $P^{V\pi}$ is irreducible and stochastic, the inequality $x \geq P^{V\pi}x$ forces $x$ to be constant, as we prove as follows.

let $s^\star$ be a state with minimal coordinate $x(s^\star) = \min_s x(s)$. Then for any $k \geq 1$,

$$x(s^\star) \geq \sum_s (P^{V\pi})^k(s^\star, s)\, x(s) \geq x(s^\star) \sum_s (P^{V\pi})^k(s^\star, s) = x(s^\star),$$

so all inequalities must be equalities. Thus whenever $(P^{V\pi})^k(s^\star, s') > 0$ we must have $x(s') = x(s^\star)$. By irreducibility, every state is reachable from $s^\star$ under some power of $P^{V\pi}$, so $x$ is constant: $x = c\mathbf{e}$ for some $c \in \mathbb{R}$, i.e.,

$$h_{Pw}^\pi = h_{PV}^\pi + c\,\mathbf{e}. \tag{30}$$

Using (30) in the Poisson equation (27) for $P^V$ gives

$$g_{PV}^\pi + h_{Pw}^\pi(s) = g_{PV}^\pi + h_{PV}^\pi(s) + c = r^\pi(s) + \sum_{s'} P^{V\pi}(s' \mid s)\, h_{PV}^\pi(s') + c$$

$$= r^\pi(s) + \sum_{s'} P^{V\pi}(s' \mid s)\, h_{Pw}^\pi(s').$$

Equivalently,

$$g_{PV}^\pi + h_{Pw}^\pi(s) = r^\pi(s) + (P^{V\pi} h_{Pw}^\pi)(s). \tag{31}$$

By construction of $P^V$ we have

$$(P^{V\pi} h_{Pw}^\pi)(s) = \sigma_s^\pi(h_{Pw}^\pi).$$

On the other hand, from the Poisson equation (26) for $P^w$,

$$g_{Pw}^\pi + h_{Pw}^\pi(s) = r^\pi(s) + (P^{w\pi} h_{Pw}^\pi)(s) \geq r^\pi(s) + (P^{V\pi} h_{Pw}^\pi)(s) = g_{PV}^\pi + h_{Pw}^\pi(s),$$

where the inequality uses the minimizing property of $P^V$ for $h_{Pw}^\pi$. Thus $g_{Pw}^\pi \geq g_{PV}^\pi$. Since $P^w$ is worst-case, $g_{Pw}^\pi \leq g_{PV}^\pi$, and we conclude that $g_{Pw}^\pi = g_{PV}^\pi$.

Combining this equality with (31), and recalling that $g_{Pw}^\pi = g_{\mathcal{P}}^\pi$, we obtain

$$g_{\mathcal{P}}^\pi + h_{Pw}^\pi(s) = r^\pi(s) + \sigma_s^\pi(h_{Pw}^\pi),$$

which is exactly (25) with $h = h_{Pw}^\pi$. Thus, the equation is solvable.

We then show the uniqueness part.

Suppose $(g, h)$ is any solution of (25). For any kernel $P \in \mathcal{P}$, we have that

$$\sigma_s^\pi(h) = \sum_a \pi(a|s)\sigma_s^a(h) \leq P^\pi h, \tag{32}$$

hence

$$g + h(s) \leq r^\pi(s) + (P^\pi h)(s), \tag{33}$$

or

$$(I - P^\pi)h \leq r^\pi(s) - g\mathbf{e}. \tag{34}$$

For this kernel $P$, let $(g_P^\pi, h_P^\pi)$ be the (unique) solution to the non-robust Bellman equation

$$(I - P^\pi)h_P^\pi = r^\pi - g_P^\pi \mathbf{e}. \tag{35}$$

We thus have that

$$(I - P^\pi)(h - h_P^\pi) \leq (g_P^\pi - g)\mathbf{e}. \tag{36}$$

Let $\mu_P$ be the unique stationary distribution of $P^\pi$, and multiply (36) with $\mu_P^\top$, we have that

$$0 = \mu_P^\top (I - P^\pi)(h - h_P^\pi) \le (g_P^\pi - g).\tag{37}$$

Thus $g \le g_P^\pi$ for any $P \in \mathcal{P}$, i.e.,

$$g \le g_{\mathcal{P}}^\pi.\tag{38}$$

We now construct a kernel $\tilde{P} \in \mathcal{P}$ as

$$\tilde{P}(\cdot|s,a) \in \arg\min_{p \in \mathcal{P}_s^a} ph,\tag{39}$$

so that $\sigma_s^a(h) = \tilde{P}h(s)$. Then (25) becomes

$$g + h(s) = r^\pi(s) + (\tilde{P}^\pi h)(s),\tag{40}$$

i.e.,

$$(I - \tilde{P}^\pi)h = r^\pi - g\mathbf{e}.\tag{41}$$

This is in fact the Bellman equation for the Markov chain $(\pi, \tilde{P})$. By standard average-reward MDP theory for irreducible chains, we must have

$$g = g_{\tilde{P}}^\pi, h = h_{\tilde{P}}^\pi + c\mathbf{e},\tag{42}$$

for some constant $c$.

Combining with (38) then implies that

$$g = g_{\tilde{P}}^\pi = g_{\mathcal{P}}^\pi,\tag{43}$$

i.e., $\tilde{P}$ is a worst-case kernel and

$$h = h_{\tilde{P}}^\pi + c\mathbf{e} = h_{P_w}^\pi + c\mathbf{e}\tag{44}$$

for some worst-case kernel $P_w$.

This hence proves uniqueness (up to constants).

$\square$

**Corollary D.2.** *The robust Bellman optimality equation*

$$V(s) = \max_a \{r(s,a) - g + \sigma_{\mathcal{P}_s^a}(V)\}\tag{45}$$

*is uniquely solvable.*

*Proof.* Denote the optimal robust policy by $\pi^\star$, and the corresponding worst-case kernel by $\mathsf{P}^\star$. By Theorem 4.6, $(g_{\mathcal{P}}^{\pi^\star}, h_{\mathsf{P}^\star}^{\pi^\star})$ is the solution to the robust Bellman equation of $\pi^\star$:

$$h_{\mathsf{P}^\star}^{\pi^\star}(s) = r^{\pi^\star}(s) - g_{\mathcal{P}}^{\pi^\star} + \sigma_s^{\pi^\star}(h_{\mathsf{P}^\star}^{\pi^\star}) = \sum_a \pi^\star(a|s)\left(r(s,a) - g_{\mathcal{P}}^{\pi^\star} + \sigma_s^a(h_{\mathsf{P}^\star}^{\pi^\star})\right).\tag{46}$$

We now set $\pi'(s) = \arg\max_a \{r(s,a) - g_{\mathcal{P}}^{\pi^\star} + \sigma_s^a(h_{\mathsf{P}^\star}^{\pi^\star})\}, \forall s$, then we have that

$$r^{\pi'}(s) - h_{\mathsf{P}'}^{\pi'}(s) + \sigma_s^{\pi'}(h_{\mathsf{P}'}^{\pi'}) = g_{\mathcal{P}}^{\pi'} \le g_{\mathcal{P}}^{\pi^\star} = \sum_a \pi^\star(a|s)\left(r(s,a) - h_{\mathsf{P}^\star}^{\pi^\star}(s) + \sigma_s^a(h_{\mathsf{P}^\star}^{\pi^\star})\right)$$

$$\le r^{\pi'}(s) - h_{\mathsf{P}^\star}^{\pi^\star}(s) + \sigma_s^{\pi'}(h_{\mathsf{P}^\star}^{\pi^\star}),\tag{47}$$

which further implies that

$$h_{\mathsf{P}^\star}^{\pi^\star}(s) - h_{\mathsf{P}'}^{\pi'}(s) \leq \sigma_s^{\pi'}(h_{\mathsf{P}^\star}^{\pi^\star}) - \sigma_s^{\pi'}(h_{\mathsf{P}'}^{\pi'}) \leq \mathsf{P}'(h_{\mathsf{P}^\star}^{\pi^\star} - h_{\mathsf{P}'}^{\pi'}). \tag{48}$$

Now we denote $x = h_{\mathsf{P}^\star}^{\pi^\star} - h_{\mathsf{P}'}^{\pi'}$, and take Cesàro summation, it then implies

$$x \leq \mu x, \tag{49}$$

for the stationary distribution matrix $\mu > 0$ (as it is irreducible, its stationary distribution has positive entries). Similarly, it implies $x(s) = x(s')$, and thus $h_{\mathsf{P}^\star}^{\pi^\star} - h_{\mathsf{P}'}^{\pi'} = ce$. This further implies that all inequalities in (47) are equations, thus

$$h_{\mathsf{P}^\star}^{\pi^\star}(s) = \max_a \left\{ r(s,a) - g_{\mathcal{P}}^{\pi^\star} + \sigma_s^a(h_{\mathsf{P}^\star}^{\pi^\star}) \right\}, \tag{50}$$

hence $(h_{\mathsf{P}^\star}^{\pi^\star}, g_{\mathsf{P}^\star}^{\pi^\star})$ is a solution to (7), completing the proof of the first part. The remaining proof follows similarly from (Wang et al., 2023c).

The uniqueness part similarly follows from the proof of Theorem D.1, by noting that $\max_x f(x) - \max_x g(x) \leq \max_x |f(x) - g(x)|$. $\qquad\square$

**Lemma D.3.** *If for any agent $i$ and any joint policy $\pi_{-i}$, $BR_i(\pi_{-i})$ is nonempty, convex, and $BR_i(\cdot)$ is upper semi-continuous, then BR also satisfies the conditions in Lemma D.4.*

*Proof.* (1). If for any $i$ and $\pi_{-i}$, $\mathrm{BR}_i(\pi_{-i})$ is nonempty, then $\mathrm{BR}(\pi) = (\mathrm{BR}_1(\pi_{-1}), ..., \mathrm{BR}_N(\pi_{-N}))$ is clearly nonempty.

(2). Consider two policies $\pi^1, \pi^2 \in \mathrm{BR}(\pi)$. To show the convexity of $\mathrm{BR}(\pi)$, we need to show $\pi' = \alpha\pi^1 + (1-\alpha)\pi^2 = (\alpha\pi_1^1 + (1-\alpha)\pi_1^2, ..., \alpha\pi_N^1 + (1-\alpha)\pi_N^2) \in \mathrm{BR}(\pi)$. Note that for any $i$, $\pi_i^1, \pi_i^2 \in \mathrm{BR}_i(\pi_{-i})$, and due to the convexity of $\mathrm{BR}_i(\pi_{-i})$, $\alpha\pi_i^1 + (1-\alpha)\pi_i^2 \in \mathrm{BR}_i(\pi_{-i})$, which completes the proof.

(3). Consider two convergent policy sequences $\{\pi^n\} \to \pi$ and $\{\mu^n\} \to \mu$, with $\mu^n \in \mathrm{BR}(\pi^n)$, we need to show that $\mu \in \mathrm{BR}(\pi)$. Note that $\pi^n \to \pi$ implies that $\pi_i^n \to \pi_i$ for any $i$, and similarly $\mu^n \to \mu$. Since $\mu_i^n \in \mathrm{BR}_i(\pi_{-i}^n)$, by the semi-continuity of $\mathrm{BR}_i$, we have $\mu_i \in \mathrm{BR}_i(\pi_{-i})$. Moreover note that

$$\mu \in \mathrm{BR}(\pi) \Leftrightarrow \mu_i \in \mathrm{BR}_i(\pi_{-i}), \forall i, \tag{51}$$

which hence completes the proof. $\qquad\square$

### D.1. Convexity

**Lemma D.4** (Kakutani's Fixed Point Theorem (Kakutani, 1941)). *Let $X$ be a compact convex subset of $\mathbb{R}^n$ and let $f : X \to 2^X$ be a set-valued function for which, (1) $f(x)$ is nonempty and convex, $\forall x$; (2) $f$ is upper semi-continuous (i.e., for any convergent sequences $\{x_n\} \to x$ and $\{y_n\} \to y$ such that $y_n \in f(x_n)$, it holds that $y \in f(x)$). Then there exists some $x^\star \in X$ such that $x^\star \in f(x^\star)$.*

**Lemma D.5.** *Consider the robust Bellman optimality equation*

$$g + h(s) = \max_{a \in \mathcal{A}} \{ r(s,a) + \sigma_s^a(h) \}, \qquad s \in \mathcal{S}, \tag{52}$$

*and let $(g, h)$ be any of its solutions. Let $\pi^\star$ be an optimal robust policy, i.e. $\pi^\star \in \arg\max_\pi g_{\mathcal{P}}^\pi$. Then $(g, h)$ also solves the robust Bellman equation associated with $\pi^\star$:*

$$g + h(s) = \sum_{a \in \mathcal{A}} \pi^\star(a \mid s)\big(r(s,a) + \sigma_s^a(h)\big), \qquad s \in \mathcal{S}. \tag{53}$$

*Proof.* By Theorem D.1, any optimal policy $\pi^\star$ admits a worst-case kernel $P^\star \in \mathcal{P}$ such that $(g_{P^\star}^{\pi^\star}, h_{P^\star}^{\pi^\star})$ solves the fixed-policy equation (25) with $\pi = \pi^\star$. Moreover $g_{P^\star}^{\pi^\star} = g_{\mathcal{P}}^\star$ (the optimal robust average reward), and the robust Bellman optimality equation (52) also has a solution $(g, h)$ with $g = g_{\mathcal{P}}^\star$. We must show that $h$ and $h_{P^\star}^{\pi^\star}$ differ only by a constant.

Equation (52) can be written as

$$g + h(s) = \max_a \{ r(s,a) + \sigma_s^a(h) \}. \tag{54}$$

Optimality of $\pi^\star$ implies that at each state $s$, the support of $\pi^\star(\cdot \mid s)$ lies in the set of maximizers of the right-hand side:

$$\pi^\star(a \mid s) > 0 \implies a \in \arg\max_{a'}\{r(s, a') + \sigma_s^{a'}(h)\}.$$

Therefore,

$$g + h(s) = \sum_a \pi^\star(a \mid s)\{r(s, a) + \sigma_s^a(h)\}, \tag{55}$$

i.e., $(g, h)$ is a solution of the fixed-policy robust Bellman equation (53) with $\pi = \pi^\star$.

By Theorem D.1, the solution of the fixed-policy robust Bellman equation for $\pi^\star$ is unique up to constants and has gain $g = g_\mathcal{P}^\star = g_{P^\star}^{\pi^\star}$. Thus $h$ and $h_{P^\star}^{\pi^\star}$ differ only by an additive constant, and in particular $(g, h)$ is of the form

$$g = g_\mathcal{P}^\star, \qquad h = h_{P^\star}^{\pi^\star} + c\mathbf{e}.$$

This hence completes the proof. $\square$

**Lemma D.6** (Convexity of optimal policies)**.** *Define the optimal policy set*

$$\Pi^\star \triangleq \{\pi \mid g_\mathcal{P}^\pi = g_\mathcal{P}^\star\}.$$

*Then $\Pi^\star$ is convex.*

*Proof.* Let $\pi_1, \pi_2 \in \Pi^\star$ and let $\tilde{\pi} \triangleq \alpha\pi_1 + (1 - \alpha)\pi_2$ for some $\alpha \in [0, 1]$, i.e.

$$\tilde{\pi}(a \mid s) = \alpha\pi_1(a \mid s) + (1 - \alpha)\pi_2(a \mid s), \quad s \in \mathcal{S}, a \in \mathcal{A}.$$

Let $(g, h)$ be any solution of the robust Bellman optimality equation (52). By Lemma D.5, $(g, h)$ also solves the fixed-policy robust Bellman equations for $\pi_1$ and $\pi_2$:

$$g + h(s) = r^{\pi_1}(s) + \sigma_s^{\pi_1}(h), \qquad g + h(s) = r^{\pi_2}(s) + \sigma_s^{\pi_2}(h).$$

Taking the convex combination with weights $(\alpha, 1 - \alpha)$ yields

$$\begin{aligned} g + h(s) &= \alpha\left(r^{\pi_1}(s) + \sigma_s^{\pi_1}(h)\right) + (1 - \alpha)\left(r^{\pi_2}(s) + \sigma_s^{\pi_2}(h)\right) \\ &= \sum_a \tilde{\pi}(a \mid s)r(s, a) + \sum_a \tilde{\pi}(a \mid s)\sigma_s^a(h) = r^{\tilde{\pi}}(s) + \sigma_s^{\tilde{\pi}}(h). \end{aligned}$$

Thus $(g, h)$ also solves the fixed-policy robust Bellman equation for $\tilde{\pi}$. By Theorem D.1, the gain of any such solution must be the robust average reward of that policy; hence

$$g_\mathcal{P}^{\tilde{\pi}} = g.$$

Since $g = g_\mathcal{P}^\star$ is the maximal robust average reward (because $\pi_1, \pi_2$ are optimal), we conclude that $\tilde{\pi}$ is also optimal, i.e. $\tilde{\pi} \in \Pi^\star$. Therefore $\Pi^\star$ is convex. $\square$

The convexity of the best-response sets in the multi-agent case then follows by applying Lemma D.6 to the induced single-agent DR-MDPs $\mathsf{M}_i(\pi_{-i})$, as discussed.

## D.2. Semi-Continuity

**Lemma D.7.** *Let $\{\pi_n\}$ be a sequence of stationary policies converging to $\pi$ in the product topology. Let $P^{\pi_n}$ and $P^\pi$ denote the induced kernels for a fixed $P \in \mathcal{P}$, and let $(P^{\pi_n})^\star$ and $(P^\pi)^\star$ be their stationary distributions. Then*

$$(P^{\pi_n})^\star \to (P^\pi)^\star$$

*uniformly in $P \in \mathcal{P}$.*

*Proof.* This is a direct consequence of the continuity properties of irreducible Markov chains; see (Puterman, 2014) (Prop. 8.4.6). Uniformity in $P$ follows from the compactness of $\mathcal{P}$. $\square$

**Lemma D.8.** *Fix an agent $i$ and let $\pi : \mathcal{S} \to \Delta(\mathcal{A}_{-i})$ be a joint policy of all other agents. Define the operator $T_{\mathcal{P}}^\pi : \mathbb{R}^{\mathcal{S}} \to \mathbb{R}^{\mathcal{S} \times \mathcal{A}_i}$ by*

$$T_{\mathcal{P}}^\pi h(s, a_i) \triangleq r^\pi(s, a_i) + \sigma_{s,a_i}^\pi(h),$$

*where*

$$r^\pi(s, a_i) \triangleq \sum_{a_{-i}} \pi(a_{-i} \mid s) \, r_i\big(s, (a_i, a_{-i})\big),$$

*and*

$$\sigma_{s,a_i}^\pi(h) \triangleq \min_{q \in \mathcal{P}_{s,a_i}^\pi} q^\top h, \qquad \mathcal{P}_{s,a_i}^\pi \triangleq \left\{ \sum_{a_{-i}} \pi(a_{-i} \mid s) P(\cdot \mid s, (a_i, a_{-i})) : P(\cdot \mid s, (a_i, a_{-i})) \in \mathcal{P}_s^{(a_i, a_{-i})} \right\}.$$

*Let $\{\pi_n\}$ be a sequence of opponent policies with $\pi_n \to \pi$, and $\{h_n\}$ a sequence of functions $h_n \to h$ uniformly on $\mathcal{S}$. Then*

$$T_{\mathcal{P}}^{\pi_n}(h_n) \to T_{\mathcal{P}}^\pi(h)$$

*uniformly on $\mathcal{S} \times \mathcal{A}_i$.*

*Proof.* Recalling the definition of $\mathcal{T}_{\mathcal{P}}$ we can obtain the difference,

$$\left\| \mathcal{T}_{\mathcal{P}}^{\pi_n}(h_n) - \mathcal{T}_{\mathcal{P}}^\pi(h) \right\|_\infty = \max_{s,a} \left| \mathcal{T}_{\mathcal{P}}^{\pi_n}(h_n)(s,a) - \mathcal{T}_{\mathcal{P}}^\pi(h)(s,a) \right|$$

$$= \max_{s,a} \left| r^{\pi_n}(s,a) + \sigma_{(\mathcal{P}^{\pi_n})_s^a}(h_n) - \sigma_{(\mathcal{P}^\pi)_s^a}(h) - r^\pi(s,a) \right|$$

$$\leq \max_{s,a} \left\{ \left| r^{\pi_n}(s,a) - r^\pi(s,a) \right| + \left| \sigma_{(\mathcal{P}^{\pi_n})_s^a}(h_n) - \sigma_{(\mathcal{P}^{\pi_n})_s^a}(h) \right| + \left| \sigma_{(\mathcal{P}^{\pi_n})_s^a}(h) - \sigma_{(\mathcal{P}^\pi)_s^a}(h) \right| \right\}$$

$$\leq \|\pi_n(s) - \pi(s)\|_1 + A + B, \tag{56}$$

where $A = |\sigma_{(\mathcal{P}^{\pi_n})_s^a}(h_n) - \sigma_{(\mathcal{P}^{\pi_n})_s^a}(h)|$ and $B = |\sigma_{(\mathcal{P}^{\pi_n})_s^a}(h) - \sigma_{(\mathcal{P}^\pi)_s^a}(h)|$. See that by the Lipchitz of $\sigma_{\mathcal{P}}(\cdot)$, $A \leq \|h_n - h\|_\infty$. Using the fact that $(\mathcal{P}^{\pi_n})_s^a = \{ \sum_{b \in \mathcal{A}_{-i}} \pi_n(b|s) P(\cdot|s, a, b) \}$, looking at $B$ we derive,

$$B = |\sigma_{(\mathcal{P}^{\pi_n})_s^a}(h) - \sigma_{(\mathcal{P}^\pi)_s^a}(h)|$$

$$= \left| \min_{q \in (\mathcal{P}^{\pi_n})_s^a} qh - \min_{q \in (\mathcal{P}^\pi)_s^a} qh \right|$$

$$= \left| \min_{q \in \mathcal{P}_s^a} q^{\pi_n} h - \min_{q \in \mathcal{P}_s^a} q^\pi h \right|$$

$$= \left| \min_{q \in \mathcal{P}_s^a} \left( \sum_{s' \in \mathcal{S}} \sum_{b \in \mathcal{A}_{-i}} \pi_n(b|s) q(s'|s, a, b) h(s') \right) - \min_{q \in \mathcal{P}_s^a} \left( \sum_{s' \in \mathcal{S}} \sum_{b \in \mathcal{A}_{-i}} \pi(b|s) q(s'|s, a, b) h(s') \right) \right|$$

$$\leq \left| \sum_{s' \in \mathcal{S}} \sum_{b \in \mathcal{A}_{-i}} \left( \pi_n(b|s) - \pi(b|s) \right) q(s'|s, a, b) h(s') \right|, \qquad \text{for some } q \in \mathcal{P}_s^a,$$

$$\leq \|\pi_n(\cdot|s) - \pi(\cdot|s)\|_1 \cdot H,$$

where $H = \|h\|_\infty$. We can now combine the terms in (56) to find,

$$\|\pi_n(s) - \pi(s)\|_1 + A + B \leq \max_{s \in \mathcal{S}} \left\{ \|\pi_n(s) - \pi(s)\|_1 + \|h_n - h\|_\infty + H \cdot \|\pi_n(s) - \pi(s)\|_1 \right\}.$$

Note that $\pi_n(s) \to \pi(s)$, $\forall s \in \mathcal{S}$, then $\forall \epsilon$, $\exists N_s$ s.t. $\forall n > N_s$, $\|\pi_n(s) - \pi(s)\|_1 \leq \epsilon$. Similarly with $h_n(s) \to h(s)$, $\forall s \in \mathcal{S}$, then $\forall \epsilon$, $\exists N_s$ s.t. $\forall n > N_s$, $|h_n(s) - h(s)| \leq \epsilon$. Now by setting $N = \max_{s \in \mathcal{S}}\{N_s\}$, then $\forall n > N$,

$$\|\pi_n(s) - \pi(s)\|_1 \leq \epsilon, \quad \text{and} \quad |h_n(s) - h(s)| \leq \epsilon, \quad \forall s \in \mathcal{S}.$$

Therefore, when $n > N$, $\|\pi_n(s) - \pi(s)\|_1 + A + B \leq \epsilon + \epsilon + H\epsilon \lesssim \epsilon$. This implies that $\mathcal{T}_{\mathcal{P}}^n \to \mathcal{T}_{\mathcal{P}}$ uniformly on $\mathcal{S} \times \mathcal{A}$, which completes the proof. $\square$

**Theorem D.9** (Semi-continuity). *Let $\pi_n \to \pi$ and $x_n \to x$, where $\pi_n : \mathcal{S} \to \Delta(\mathcal{A}_{-i})$ and $x_n : \mathcal{S} \to \Delta(\mathcal{A}_i)$. If $x_n \in BR_i(\pi_n)$, then $x \in BR_i(\pi)$.*

*Proof.* To show that $x \in \mathrm{BR}_i(\pi)$, it suffices to show that $x$ is an optimal policy of the induced AMDP $(\mathcal{S}, \mathcal{A}_i, \mathcal{P}^\pi, r^\pi)$. Recalling the robust Bellman equation, it is also sufficient to show that $g_{\mathcal{P}}^{(x,\pi)}$ satisfies the following:

$$g(s) = \max_{a \in \mathcal{A}_i}\{r^\pi(s,a) + \sigma_{(\mathcal{P}^\pi)_s^a}(h) - h(s)\} = \max_{a \in \mathcal{A}_i}\{\mathcal{T}_{\mathcal{P}_s^a}^\pi(h)\}. \tag{57}$$

See that since $x_n \in \mathrm{BR}_i(\pi_n)$, then $x_n$ is the optimal policy of the induced AMDP $(\mathcal{S}, \mathcal{A}_i, \mathcal{P}^{\pi_n}, r^{\pi_n})$. Thus, there exists $h_n$, s.t. $(h_n, g_n = g_{\mathcal{P}^{\pi_n}}^{x_n})$ is a solution to,

$$g_{\mathcal{P}^{\pi_n}}^{x_n} = g_n(s) = \max_a\{r^{\pi_n}(s,a) + \sigma_{(\mathcal{P}^{\pi_n})_s^a}(h_n) - h_n(s)\}.$$

See that $g_{\mathcal{P}^{\pi_n}}^{x_n} = \min_{\mathsf{P}^{\pi_n} \in \mathcal{P}^{\pi_n}} g_{\mathsf{P}^{\pi_n}, r^{\pi_n}}^{x_n} = \min_{\mathsf{P} \in \mathcal{P}} g_{\mathsf{P},n}^{(x_n,\pi_n)}$, and so $g_{\mathcal{P}^{\pi_n}}^{x_n} = g_{\mathcal{P}}^{\pi_n, x_n}$. Therefore, for any $(x_n, \pi_n)$, there exists some $h_n \in \mathbb{R}^{\mathcal{S}}$ such that for all $s \in \mathcal{S}$,

$$\begin{aligned}
g_{\mathcal{P}_s}^{(x_n,\pi_n)}(s) &= \max_a\{r^{\pi_n}(s,a) + \sigma_{(\mathcal{P}^{\pi_n})_s^a}(h_n) - h_n(s)\} \\
&= \max_a\{\mathcal{T}_{\mathcal{P}_s^a}^{\pi_n}(h_n)\}.
\end{aligned} \tag{58}$$

Moreover, since $h_n$ is some relative function (Wang et al., 2023b), $\|h_n\|_\infty \leq H$. Then, there exists some subsequence $\{n_m\}$, s.t. $h_{n_m} \to h$ for some $h$, and $g_{\mathcal{P}}^{(x_n,\pi_n)} \to g$ for some $g$. Then, we can take the limit as $m \to \infty$ in (58) to find:

$$\begin{aligned}
\lim_{m \to \infty} g_{\mathcal{P}}^{(x_{n_m},\pi_{n_m})} &= \lim_{m \to \infty} \max_a\{\mathcal{T}_{\mathcal{P}}^{\pi_{n_m}}(h_{n_m})\} \\
&\overset{(i)}{=} \max_a\{\lim_{m \to \infty} \mathcal{T}_{\mathcal{P}}^{\pi_{n_m}}(h_{n_m})\} \\
&= \max_a\{\mathcal{T}_{\mathcal{P}}^\pi(h)\},
\end{aligned} \tag{59}$$

where $(i)$ holds by the uniform convergence in Lemma D.8, and the final equality by $\pi_{n_m} \to \pi$, and $h_{n_m} \to h$. Looking at the left-hand side of equation (59) we find

$$\begin{aligned}
\lim_{m \to \infty} g_{\mathcal{P}}^{(x_{n_m},\pi_{n_m})} &= \lim_{m \to \infty} \min_{\mathsf{P} \in \mathcal{P}} g_{\mathsf{P}}^{(x_{n_m},\pi_{n_m})} \\
&= \lim_{m \to \infty} \min_{\mathsf{P} \in \mathcal{P}} \mathsf{P}_{(x_{n_m},\pi_{n_m})}^\star \cdot r^{(x_{n_m},\pi_{n_m})}.
\end{aligned}$$

By (Puterman, 2014) and (Wang et al., 2023b), we have $\mathsf{P}_{(x_{n_m},\pi_{n_m})}^\star \to \mathsf{P}_{(x,\pi)}^\star$ uniformly. Thus, $\lim \min = \min \lim$ implies,

$$\begin{aligned}
\lim_{m \to \infty} g_{\mathcal{P}}^{(x_{n_m},\pi_{n_m})} &= \min_{\mathsf{P} \in \mathcal{P}} \lim_{m \to \infty} g_{\mathsf{P}}^{(x_{n_m},\pi_{n_m})} \\
&= \min_{\mathsf{P} \in \mathcal{P}} g_{\mathsf{P}}^{(x,\pi)} \\
&= g_{\mathcal{P}}^{(x,\pi)}.
\end{aligned}$$

Therefore, $g_{\mathcal{P}}^{(x,\pi)} = \max_a\{\mathcal{T}_{\mathcal{P}}^\pi(h)\}$ for some $h \in \mathbb{R}^{\mathcal{S}}$, which completes the proof. $\qquad\square$

# E. Weakly Communicating DR-MGs

Throughout this section we fix a player $i \in \mathcal{N}$ and a stationary joint policy $\pi_{-i}$ of the other players, and work with the induced single-agent distributionally robust MDP

$$\mathsf{M}_i(\pi_{-i}) = (\mathcal{S}, \mathcal{A}_i, \mathcal{P}^{\pi_{-i}}, r_i^{\pi_{-i}}).$$

All quantities defined below (transition sets, Bellman solutions, etc.) depend on $(i, \pi_{-i})$, but we often suppress this dependence in the notation to lighten the presentation.

The stage reward $r_i^{\pi_{-i}}(s, a_i)$ is bounded in $[0, 1]$ by assumption. The induced ambiguity set $\mathcal{P}^{\pi_{-i}}$ inherits $(s, a_i)$–rectangularity from the original game.

For a stationary policy $\pi_i : \mathcal{S} \to \Delta(\mathcal{A}_i)$ of player $i$ and a kernel $P \in \mathcal{P}^{\pi_{-i}}$, let $g_P^{\pi_i}$ denote the long-run average reward of $\pi_i$ in $\mathsf{M}_i(\pi_{-i})$ under $P$ (which exists by standard finite-state average-reward theory). We define the robust worst-case average reward and the robust best-response set

$$g_{\mathcal{P}^{\pi_{-i}}}^{\pi_i} \triangleq \min_{P \in \mathcal{P}^{\pi_{-i}}} g_P^{\pi_i}, \tag{60}$$

$$BR_i(\pi_{-i}) \triangleq \left\{ \pi_i : g_{\mathcal{P}^{\pi_{-i}}}^{\pi_i} = \sup_{\mu : \mathcal{S} \to \Delta(\mathcal{A}_i)} \min_{P \in \mathcal{P}^{\pi_{-i}}} g_P^{\mu} \right\}. \tag{61}$$

**Lemma E.1.** *Let $(g^*, h_1)$ and $(g^*, h_2)$ be two solutions to the Robust Bellman Optimality Equation in* (7). *Under the unichain assumption (Wang et al., 2023b), $h_1(s) - h_2(s) = c$ for some constant $c$, for all $s \in \mathcal{S}$. Namely, the equation has a unique solution up to some constant.*

*Proof.* Define the difference function $\Delta(s) = h_1(s) - h_2(s)$.

Let $\pi_1$ be a greedy policy for $h_1$, i.e., $\pi_1(s) \in \arg\max_a\{r(s,a) + \sigma_{\mathcal{P}_s^a}(h_1)\}$. Let $\nu_1$ be the worst-case nature kernel for $h_1$ under $\pi_1$. It then holds that

$$g^* + h_1(s) = r(s, \pi_1(s)) + \sum_{s'} p(s'|s, \pi_1(s), \nu_1) h_1(s') \tag{62}$$

On the other hand, since $h_2$ is also a solution to the optimal equation, it must satisfy the maximization condition. Specifically for policy $\pi_1$:

$$g^* + h_2(s) = \max_a\{r(s,a) + \sigma_{\mathcal{P}_s^a}(h_2)\} \geq r(s, \pi_1(s)) + \min_{p \in \mathcal{P}(s, \pi_1(s))} \sum_{s'} p(s'|s, \pi_1(s)) h_2(s') \tag{63}$$

Let $\nu_{1,2}$ be the nature kernel that achieves the minimization for $h_2$ given $\pi_1$. Then:

$$g^* + h_2(s) \geq r(s, \pi_1(s)) + \sum_{s'} p(s'|s, \pi_1(s), \nu_{1,2}) h_2(s') \tag{64}$$

Returning to (62), since $\nu_1$ minimizes $h_1$, replacing it with $\nu_{1,2}$ yields a value $\geq$ the minimum:

$$g^* + h_1(s) \leq r(s, \pi_1(s)) + \sum_{s'} p(s'|s, \pi_1(s), \nu_{1,2}) h_1(s') \tag{65}$$

Subtracting (64) from (65):

$$h_1(s) - h_2(s) \leq \sum_{s'} p(s'|s, \pi_1(s), \nu_{1,2})(h_1(s') - h_2(s')) \tag{66}$$

Let $P_A$ be the transition matrix induced by $(\pi_1, \nu_{1,2})$. We have:

$$\Delta \leq P_A \Delta \tag{67}$$

Thus, $\Delta$ is sub-harmonic with respect to $P_A$.

By symmetry, let $\pi_2$ be optimal for $h_2$, and let $\nu_{2,1}$ minimize $h_1$ under $\pi_2$. We similarly derive:

$$\Delta \geq P_B \Delta \tag{68}$$

where $P_B$ is induced by $(\pi_2, \nu_{2,1})$. Thus, $\Delta$ is super-harmonic with respect to $P_B$.

By assumption, both $P_A$ and $P_B$ define unichain Markov chains. Since $\Delta \leq P_A \Delta$, $\Delta$ cannot attain a local maximum on the recurrent class of $P_A$ strictly greater than its neighbors. By the Maximum Principle, $\Delta(s)$ must be constant on the recurrent class $\mathcal{C}_A$ of $P_A$. Let this value be $M = \max_s \Delta(s)$.

On the other hand, since $\Delta \geq P_B \Delta$, $\Delta$ cannot attain a local minimum on the recurrent class of $P_B$ strictly lower than its neighbors. By the Minimum Principle, $\Delta(s)$ must be constant on the recurrent class $\mathcal{C}_B$ of $P_B$. Let this value be $m = \min_s \Delta(s)$.

Assume $M > m$. From (67), $\Delta(s)$ is non-decreasing in expectation under $P_A$. Trajectories under $P_A$ eventually enter $\mathcal{C}_A$ (where $\Delta = M$). Thus for any transient state $s$, $\Delta(s) \leq M$. From (68), $\Delta(s)$ is non-increasing in expectation under $P_B$. Trajectories under $P_B$ eventually enter $\mathcal{C}_B$ (where $\Delta = m$).

The unichain assumption implies the graph is connected. However, if $M > m$, the gain $g^*$ would necessarily differ or one policy would exploit the transition to the region of higher bias, contradicting the optimality of the policies for the fixed scalar $g^*$. Specifically, if $\Delta$ is not constant, the condition $\Delta \leq P_A \Delta$ and $\Delta \geq P_B \Delta$ combined with $g^*$ being unique forces:

$$P_A \Delta = \Delta \quad \text{and} \quad P_B \Delta = \Delta \tag{69}$$

For a harmonic function on a weakly communicating chain, the values on transient states are convex combinations of values on the recurrent set. Since the recurrent sets have constant values $M$ (and $m$ respectively), and the global gain constraint prevents flow from low to high regions without penalty, we must have $M = m$.

Therefore, $\Delta(s) = c$ for all $s \in \mathcal{S}$, which completes the proof. $\qquad\square$

### E.1. Constant-Gain Robust Bellman Equations

Under Assumption 5.1 (weakly communicating DR-MG), the induced DR-MDP $\mathsf{M}_i(\pi_{-i})$ is a finite weakly communicating robust MDP with $(s, a_i)$–rectangular ambiguity. We then therefore invoke the constant-gain robust Bellman theory of (Wang & Si, 2025).

**Lemma E.2** ((Wang & Si, 2025)). *There exists a pair $(h^\star, \alpha^\star) \in \mathbb{R}^\mathcal{S} \times \mathbb{R}$ with $\alpha^\star \in [0, 1]$ such that the following hold for every $s \in \mathcal{S}$:*

$$h^\star(s) = \sup_{\phi \in \Delta(\mathcal{A}_i)} \inf_{p_s \in \left(\mathcal{P}^{\pi_{-i}}\right)_s} \sum_{a_i \in \mathcal{A}_i} \phi(a_i) \left(r_i^{\pi_{-i}}(s, a_i) - \alpha^\star + p_{s,a_i}^\top h^\star\right), \tag{70}$$

$$= \inf_{p_s \in \left(\mathcal{P}^{\pi_{-i}}\right)_s} \sup_{\phi \in \Delta(\mathcal{A}_i)} \sum_{a_i \in \mathcal{A}_i} \phi(a_i) \left(r_i^{\pi_{-i}}(s, a_i) - \alpha^\star + p_{s,a_i}^\top h^\star\right). \tag{71}$$

*Moreover, $\alpha^\star$ coincides with the optimal robust average reward:*

$$\alpha^\star = \sup_{\pi_i : \mathcal{S} \to \Delta(\mathcal{A}_i)} \min_{P \in \mathcal{P}^{\pi_{-i}}} g_P^{\pi_i}.$$

The proof is an application of Wang and Si (2025) to the induced DR-MDP $\mathsf{M}_i(\pi_{-i})$ and is omitted here.

For each $s \in \mathcal{S}$ we introduce the local saddle Lagrangian

$$L_s(\phi, p_s) \triangleq \sum_{a_i \in \mathcal{A}_i} \phi(a_i) \left(r_i^{\pi_{-i}}(s, a_i) - \alpha^\star + p_{s,a_i}^\top h^\star\right), \quad \phi \in \Delta(\mathcal{A}_i), \; p_s \in \left(\mathcal{P}^{\pi_{-i}}\right)_s. \tag{72}$$

We then define the local player and adversary value functions

$$F_s(\phi) \triangleq \inf_{p_s \in \left(\mathcal{P}^{\pi_{-i}}\right)_s} L_s(\phi, p_s), \tag{73}$$

$$\Psi_s(p_s) \triangleq \sup_{\phi \in \Delta(\mathcal{A}_i)} L_s(\phi, p_s). \tag{74}$$

We then derive the following results.

**Lemma E.3.** *For each state $s \in \mathcal{S}$:*

1. *$F_s$ is concave and upper semicontinuous on the compact convex set $\Delta(\mathcal{A}_i)$.*

2. *$\Psi_s$ is convex and lower semicontinuous on $\left(\mathcal{P}^{\pi_{-i}}\right)_s$.*

3. *With $(h^\star, \alpha^\star)$ from Lemma E.2,*

$$h^\star(s) = \max_{\phi \in \Delta(\mathcal{A}_i)} F_s(\phi) = \min_{p_s \in \left(\mathcal{P}^{\pi_{-i}}\right)_s} \Psi_s(p_s). \tag{75}$$

*Proof.* (1) For fixed $p_s$, the map $\phi \mapsto L_s(\phi, p_s)$ is affine (hence continuous and both concave and convex) on the simplex $\Delta(\mathcal{A}_i)$. The function $F_s$ is the pointwise infimum of this family of continuous affine functions; infima of affine functions are concave, and infima of continuous functions are upper semicontinuous.

(2) For fixed $\phi$, the map $p_s \mapsto L_s(\phi, p_s)$ is affine (and continuous) on the convex set $\left(\mathcal{P}^{\pi_{-i}}\right)_s$. The function $\Psi_s$ is the pointwise supremum of a family of continuous affine functions; such a supremum is convex and lower semicontinuous.

(3) Comparing (72)–(74) with (70)–(71) we see that

$$h^\star(s) = \sup_{\phi \in \Delta(\mathcal{A}_i)} \inf_{p_s \in \left(\mathcal{P}^{\pi_{-i}}\right)_s} L_s(\phi, p_s) = \sup_{\phi \in \Delta(\mathcal{A}_i)} F_s(\phi),$$

$$h^\star(s) = \inf_{p_s \in \left(\mathcal{P}^{\pi_{-i}}\right)_s} \sup_{\phi \in \Delta(\mathcal{A}_i)} L_s(\phi, p_s) = \inf_{p_s \in \left(\mathcal{P}^{\pi_{-i}}\right)_s} \Psi_s(p_s).$$

Since $F_s$ is upper semicontinuous on the compact set $\Delta(\mathcal{A}_i)$, its maximum is attained. Likewise $\Psi_s$ attains its minimum on the compact set $\left(\mathcal{P}^{\pi_{-i}}\right)_s$. $\qquad\square$

We now define the set of statewise Bellman maximizers

$$M_s \triangleq \arg \max_{\phi \in \Delta(\mathcal{A}_i)} F_s(\phi), \qquad s \in \mathcal{S}. \tag{76}$$

By Lemma E.3, each $M_s$ is nonempty, convex, and compact.

### E.2. A Common Supporting Minimizer

**Lemma E.4.** *For each $s \in \mathcal{S}$ there exists*

$$p_s^\star \in \arg \min_{p_s \in \left(\mathcal{P}^{\pi_{-i}}\right)_s} \Psi_s(p_s)$$

*such that*

$$h^\star(s) = \Psi_s(p_s^\star) = L_s(\phi, p_s^\star) = F_s(\phi), \quad \forall\, \phi \in M_s. \tag{77}$$

*Proof.* From Lemma E.3(2) and compactness of $\left(\mathcal{P}^{\pi_{-i}}\right)_s$, $\Psi_s$ attains its minimum; pick $p_s^\star \in \left(\mathcal{P}^{\pi_{-i}}\right)_s$ such that

$$\Psi_s(p_s^\star) = \min_{p_s} \Psi_s(p_s) = h^\star(s).$$

Now let $\phi \in M_s$. Then $F_s(\phi) = h^\star(s)$ by definition of $M_s$ and (75). We have the chain of inequalities

$$h^\star(s) = F_s(\phi) = \inf_{p_s} L_s(\phi, p_s) \leq L_s(\phi, p_s^\star) \leq \sup_{\tilde\phi} L_s(\tilde\phi, p_s^\star) = \Psi_s(p_s^\star) = h^\star(s).$$

Thus all inequalities are equalities, which yields

$$F_s(\phi) = L_s(\phi, p_s^\star) = \Psi_s(p_s^\star) = h^\star(s)$$

for every $\phi \in M_s$. $\qquad\square$

Because the ambiguity set is $(s, a_i)$–rectangular, the global collection $p^\star \triangleq \{p_s^\star\}_{s \in \mathcal{S}}$ belongs to $\mathcal{P}^{\pi_{-i}}$: we choose independently, for each state $s$, a row block $p_s^\star$ from $\left(\mathcal{P}^{\pi_{-i}}\right)_s$, and rectangularity ensures that this rowwise product is feasible.

Under Assumption 5.1, $\mathsf{M}_i(\pi_{-i})$ is weakly communicating for every kernel $P \in \mathcal{P}^{\pi_{-i}}$, in particular for $P = p^\star$. By the standard structure theorem for weakly communicating MDPs (see, e.g., (Puterman, 2014), Chap. 8), there exists a decomposition

$$\mathcal{S} = \mathcal{S}_0 \,\dot\cup\, \mathcal{C}_{p^\star},$$

where:

- States in $\mathcal{S}_0$ are transient under any stationary policy of player $i$ (combined with the fixed opponents $\pi_{-i}$ and kernel $p^\star$).

- The set $\mathcal{C}_{p^\star}$ is the union of the recurrent communicating classes of the Markov chains induced by stationary policies under $p^\star$.

We call $\mathcal{C}_{p^\star}$ the *communicating region* of $p^\star$.

### E.3. Slack Inequalities and Characterization of $G_i$

For a stationary policy $\nu : \mathcal{S} \to \Delta(\mathcal{A}_i)$ of player $i$ we define the Bellman slack

$$\delta_\nu(s) \triangleq h^\star(s) - F_s\big(\nu(\cdot \mid s)\big), \qquad s \in \mathcal{S}. \tag{78}$$

By (75), $h^\star(s) = \max_\phi F_s(\phi)$, so $F_s(\nu(\cdot \mid s)) \leq h^\star(s)$ and hence $\delta_\nu(s) \geq 0$.

We recall the definitions (72)–(74) and (75):

$$L_s(\phi, p_s) = \sum_{a_i \in \mathcal{A}_i} \phi(a_i)\big(r_i^{\pi^{-i}}(s, a_i) - \alpha^\star + p_{s,a_i}^\top h^\star\big), \tag{79}$$

$$F_s(\phi) = \inf_{p_s \in (\mathcal{P}^{\pi^{-i}})_s} L_s(\phi, p_s), \qquad \Psi_s(p_s) = \sup_{\phi \in \Delta(\mathcal{A}_i)} L_s(\phi, p_s), \tag{80}$$

$$h^\star(s) = \max_{\phi \in \Delta(\mathcal{A}_i)} F_s(\phi) = \min_{p_s \in (\mathcal{P}^{\pi^{-i}})_s} \Psi_s(p_s). \tag{81}$$

Fix player $i$ and opponent policy $\pi^{-i}$ and consider the induced DR-MDP $\mathsf{M}_i(\pi^{-i}) = (\mathcal{S}, \mathcal{A}_i, \mathcal{P}^{\pi^{-i}}, r_i^{\pi^{-i}})$. Let $(h^\star, \alpha^\star)$ be a constant-gain solution of the *optimal* robust Bellman equation for $\mathsf{M}_i(\pi^{-i})$ (as in Lemma E.2).

For each state $s \in \mathcal{S}$, define the local Lagrangian and values

$$L_s(\phi, p_s) \triangleq \sum_{a_i \in \mathcal{A}_i} \phi(a_i)\Big(r_i^{\pi^{-i}}(s, a_i) - \alpha^\star + p_{s,a_i}^\top h^\star\Big), \qquad \phi \in \Delta(\mathcal{A}_i),\ p_s \in (\mathcal{P}^{\pi^{-i}})_s, \tag{82}$$

$$F_s(\phi) \triangleq \inf_{p_s \in (\mathcal{P}^{\pi^{-i}})_s} L_s(\phi, p_s), \qquad \Psi_s(p_s) \triangleq \sup_{\phi \in \Delta(\mathcal{A}_i)} L_s(\phi, p_s), \tag{83}$$

and the statewise maximizer set

$$M_s \triangleq \arg\max_{\phi \in \Delta(\mathcal{A}_i)} F_s(\phi).$$

By strong duality (Lemma E.3), for every $s$,

$$h^\star(s) = \max_\phi F_s(\phi) = \min_{p_s} \Psi_s(p_s).$$

Let $p_s^\star \in \arg\min_{p_s \in (\mathcal{P}^{\pi^{-i}})_s} \Psi_s(p_s)$ be a *common supporting minimizer* as in Lemma E.4, and define the global kernel $p^\star \triangleq \{p_s^\star\}_{s \in \mathcal{S}} \in \mathcal{P}^{\pi^{-i}}$.

**Lemma E.5.** *For any stationary policy $\nu : \mathcal{S} \to \Delta(\mathcal{A}_i)$ define the Bellman slack*

$$\delta_\nu(s) \triangleq h^\star(s) - F_s(\nu(\cdot|s)) \geq 0. \tag{84}$$

*Let $(S_t, A_t)_{t \geq 0}$ be the trajectory generated by the pair $(\nu, p^\star)$. Then for every integer $n \geq 1$,*

$$\mathbb{E}\Big[\sum_{t=0}^{n-1}\big(r_i^{\pi^{-i}}(S_t, A_t) - \alpha^\star\big)\Big] \leq \mathbb{E}[h^\star(S_0)] - \mathbb{E}[h^\star(S_n)], \tag{85}$$

$$\mathbb{E}\Big[\sum_{t=0}^{n-1}\big(r_i^{\pi^{-i}}(S_t, A_t) - \alpha^\star\big)\Big] \geq \mathbb{E}[h^\star(S_0)] - \mathbb{E}[h^\star(S_n)] - \mathbb{E}\Big[\sum_{t=0}^{n-1}\delta_\nu(S_t)\Big]. \tag{86}$$

*Consequently,*

$$\limsup_{n\to\infty} \mathbb{E}\Big[\frac{1}{n}\sum_{t=0}^{n-1}\big(r_i^{\pi^{-i}}(S_t, A_t) - \alpha^\star\big)\Big] \le 0, \tag{87}$$

$$\liminf_{n\to\infty} \mathbb{E}\Big[\frac{1}{n}\sum_{t=0}^{n-1}\big(r_i^{\pi^{-i}}(S_t, A_t) - \alpha^\star\big)\Big] \ge -\limsup_{n\to\infty} \mathbb{E}\Big[\frac{1}{n}\sum_{t=0}^{n-1}\delta_\nu(S_t)\Big]. \tag{88}$$

*Moreover, let $\mathcal{S} = \mathcal{S}_0 \cup C_{p^\star}$ be the weakly-communicating decomposition under $p^\star$ (i.e., $\mathcal{S}_0$ is transient under every stationary policy under $p^\star$, and $C_{p^\star}$ is the union of recurrent classes). If $\nu(\cdot|s) \in M_s$ for all $s \in C_{p^\star}$, then the average reward of $\nu$ under $p^\star$ equals $\alpha^\star$:*

$$g_{p^\star}^\nu = \alpha^\star.$$

*Proof.* We prove the two inequalities (85)–(86) first, then the final claim.

**Step 1: Upper bound (85).** Fix a time $t \ge 0$ and condition on $S_t = s$. Under $(\nu, p^\star)$, we have $A_t \sim \nu(\cdot|s)$ and $S_{t+1} \sim p_{s,A_t}^\star$, thus

$$\mathbb{E}\big[r_i^{\pi^{-i}}(S_t, A_t) - \alpha^\star + h^\star(S_{t+1}) \mid S_t = s\big] = \sum_{a_i}\nu(a_i|s)\Big(r_i^{\pi^{-i}}(s, a_i) - \alpha^\star + (p_{s,a_i}^\star)^\top h^\star\Big)$$
$$= L_s\big(\nu(\cdot|s), p_s^\star\big).$$

Since $p_s^\star$ minimizes $\Psi_s$ and $\Psi_s(p_s^\star) = h^\star(s)$, we have for any $\phi \in \Delta(\mathcal{A}_i)$,

$$L_s(\phi, p_s^\star) \le \sup_{\tilde{\phi}} L_s(\tilde{\phi}, p_s^\star) = \Psi_s(p_s^\star) = h^\star(s).$$

Taking $\phi = \nu(\cdot|s)$ yields

$$\mathbb{E}\big[r_i^{\pi^{-i}}(S_t, A_t) - \alpha^\star + h^\star(S_{t+1}) \mid S_t = s\big] \le h^\star(s) = h^\star(S_t).$$

Equivalently,

$$\mathbb{E}\big[r_i^{\pi^{-i}}(S_t, A_t) - \alpha^\star \mid S_t\big] \le h^\star(S_t) - \mathbb{E}\big[h^\star(S_{t+1}) \mid S_t\big].$$

Taking total expectation and summing from $t = 0$ to $n - 1$ gives

$$\mathbb{E}\Big[\sum_{t=0}^{n-1}\big(r_i^{\pi^{-i}}(S_t, A_t) - \alpha^\star\big)\Big] \le \mathbb{E}\Big[\sum_{t=0}^{n-1}\big(h^\star(S_t) - h^\star(S_{t+1})\big)\Big]$$
$$= \mathbb{E}\big[h^\star(S_0) - h^\star(S_n)\big],$$

where the last equality is the telescoping identity $\sum_{t=0}^{n-1}(h^\star(S_t) - h^\star(S_{t+1})) = h^\star(S_0) - h^\star(S_n)$. This proves (85).

**Step 2: Lower bound (86).** Fix a state $s$. By definition of $F_s$ as an infimum over $p_s$,

$$F_s\big(\nu(\cdot|s)\big) = \inf_{p_s \in (\mathcal{P}^{\pi^{-i}})_s} L_s\big(\nu(\cdot|s), p_s\big) \le L_s\big(\nu(\cdot|s), p_s^\star\big). \tag{89}$$

Using the slack definition (84), $F_s(\nu(\cdot|s)) = h^\star(s) - \delta_\nu(s)$, so (89) becomes

$$h^\star(s) - \delta_\nu(s) \le L_s\big(\nu(\cdot|s), p_s^\star\big). \tag{90}$$

Now condition on $S_t = s$ and use the same computation as in Step 1: $L_s(\nu(\cdot|s), p_s^\star) = \mathbb{E}[r_i^{\pi^{-i}}(S_t, A_t) - \alpha^\star + h^\star(S_{t+1}) \mid S_t = s]$. Thus (90) translates to

$$h^\star(S_t) - \delta_\nu(S_t) \le \mathbb{E}\big[r_i^{\pi^{-i}}(S_t, A_t) - \alpha^\star + h^\star(S_{t+1}) \mid S_t\big].$$

Rearranging gives

$$\mathbb{E}\big[r_i^{\pi^{-i}}(S_t, A_t) - \alpha^\star \mid S_t\big] \geq h^\star(S_t) - \mathbb{E}\big[h^\star(S_{t+1}) \mid S_t\big] - \delta_\nu(S_t).$$

Taking total expectation and summing $t = 0, \ldots, n-1$ yields

$$\mathbb{E}\Big[\sum_{t=0}^{n-1}\big(r_i^{\pi^{-i}}(S_t, A_t) - \alpha^\star\big)\Big] \geq \mathbb{E}\Big[\sum_{t=0}^{n-1}\big(h^\star(S_t) - h^\star(S_{t+1})\big)\Big] - \mathbb{E}\Big[\sum_{t=0}^{n-1}\delta_\nu(S_t)\Big]$$

$$= \mathbb{E}[h^\star(S_0)] - \mathbb{E}[h^\star(S_n)] - \mathbb{E}\Big[\sum_{t=0}^{n-1}\delta_\nu(S_t)\Big],$$

proving (86).

**Step 3: If $\nu$ is greedy on $C_{p^\star}$ then $g_{p^\star}^\nu = \alpha^\star$.** Assume $\nu(\cdot|s) \in M_s$ for all $s \in C_{p^\star}$. Then for $s \in C_{p^\star}$ we have $F_s(\nu(\cdot|s)) = h^\star(s)$ by definition of $M_s$, hence $\delta_\nu(s) = 0$ for all $s \in C_{p^\star}$.

By weakly communicating structure under $p^\star$, $\mathcal{S}_0$ is transient under *every* stationary policy, in particular under $\nu$. Therefore, the total expected number of visits to $\mathcal{S}_0$ is finite:

$$\sum_{t=0}^{\infty}\mathbb{P}(S_t \in \mathcal{S}_0) < \infty,$$

which implies the Cesàro mean vanishes:

$$\lim_{n\to\infty}\frac{1}{n}\sum_{t=0}^{n-1}\mathbb{P}(S_t \in \mathcal{S}_0) = 0.$$

Because rewards are bounded and $\|h^\star\|_\infty < \infty$ (Proposition E.7), the slack is uniformly bounded: there exists $B_\delta < \infty$ such that $0 \leq \delta_\nu(s) \leq B_\delta$ for all $s$. Hence

$$0 \leq \frac{1}{n}\mathbb{E}\Big[\sum_{t=0}^{n-1}\delta_\nu(S_t)\Big] \leq \frac{B_\delta}{n}\sum_{t=0}^{n-1}\mathbb{P}(S_t \in \mathcal{S}_0) \xrightarrow[n\to\infty]{} 0.$$

Now divide (85)–(86) by $n$ and let $n \to \infty$. Since $|h^\star(S_n)| \leq \|h^\star\|_\infty$, the boundary term $(\mathbb{E}[h^\star(S_0)] - \mathbb{E}[h^\star(S_n)])/n$ vanishes. The upper bound gives $\limsup_{n\to\infty}\mathbb{E}[\frac{1}{n}\sum_{t=0}^{n-1}(r - \alpha^\star)] \leq 0$, while the lower bound together with $\frac{1}{n}\mathbb{E}[\sum_{t=0}^{n-1}\delta_\nu(S_t)] \to 0$ gives $\liminf_{n\to\infty}\mathbb{E}[\frac{1}{n}\sum_{t=0}^{n-1}(r - \alpha^\star)] \geq 0$. Therefore the limit exists and equals 0, i.e.,

$$\lim_{n\to\infty}\mathbb{E}\Big[\frac{1}{n}\sum_{t=0}^{n-1}r_i^{\pi^{-i}}(S_t, A_t)\Big] = \alpha^\star,$$

which is exactly $g_{p^\star}^\nu = \alpha^\star$. $\qquad\square$

**Theorem E.6.** *Fix $\pi^{-i}$ and define*

$$G_i(\pi^{-i}) \triangleq \Big\{\nu : \mathcal{S} \to \Delta(\mathcal{A}_i) \;\Big|\; \nu(\cdot|s) \in M_s \;\forall s \in \mathcal{S}\Big\}. \tag{91}$$

*Then $G_i(\pi^{-i})$ is nonempty, compact, and convex. Moreover,*

$$G_i(\pi^{-i}) \subseteq \mathrm{BR}_i(\pi^{-i}).$$

*Proof.* **Step 1: Nonemptiness, compactness, convexity.** For each $s$, the set $M_s = \arg\max_{\phi \in \Delta(\mathcal{A}_i)} F_s(\phi)$ is nonempty, compact, and convex (Lemma E.3 and the fact that $F_s$ is concave and upper semicontinuous on the compact simplex). Therefore the Cartesian product $\prod_{s\in\mathcal{S}} M_s$ is nonempty, compact, and convex in the product topology. By definition, $G_i(\pi^{-i})$ equals this product set.

**Step 2: Robust optimality.** Let $\nu \in G_i(\pi^{-i})$. Then by definition of $G_i(\pi^{-i})$, for every state $s$, $\nu(\cdot|s) \in M_s$ and hence

$$F_s\big(\nu(\cdot|s)\big) = \max_\phi F_s(\phi) = h^\star(s). \tag{92}$$

Now fix an *arbitrary* kernel $P \in \mathcal{P}^{\pi^{-i}}$ and let $(S_t, A_t)$ evolve under $(\nu, P)$. Conditioning on $S_t = s$ and using the definition of $F_s$ as an infimum,

$$F_s(\nu(\cdot|s)) = \inf_{p_s} L_s(\nu(\cdot|s), p_s) \le L_s(\nu(\cdot|s), P_s) = \mathbb{E}\big[r_i^{\pi^{-i}}(S_t, A_t) - \alpha^\star + h^\star(S_{t+1}) \mid S_t = s\big].$$

Combining with (92) gives the one-step inequality

$$h^\star(S_t) \le \mathbb{E}\big[r_i^{\pi^{-i}}(S_t, A_t) - \alpha^\star + h^\star(S_{t+1}) \mid S_t\big].$$

Rearrange:

$$\mathbb{E}\big[r_i^{\pi^{-i}}(S_t, A_t) - \alpha^\star \mid S_t\big] \ge h^\star(S_t) - \mathbb{E}\big[h^\star(S_{t+1}) \mid S_t\big].$$

Taking total expectation and summing $t = 0, \ldots, n-1$ yields

$$\mathbb{E}\Big[\sum_{t=0}^{n-1}\big(r_i^{\pi^{-i}}(S_t, A_t) - \alpha^\star\big)\Big] \ge \mathbb{E}[h^\star(S_0)] - \mathbb{E}[h^\star(S_n)].$$

Divide by $n$ and let $n \to \infty$; the boundary term vanishes because $\|h^\star\|_\infty < \infty$. We conclude that

$$g_P^\nu = \lim_{n\to\infty} \mathbb{E}\Big[\frac{1}{n}\sum_{t=0}^{n-1} r_i^{\pi^{-i}}(S_t, A_t)\Big] \ge \alpha^\star, \qquad \forall P \in \mathcal{P}^{\pi^{-i}}.$$

Therefore $\min_{P\in\mathcal{P}^{\pi^{-i}}} g_P^\nu \ge \alpha^\star$. But by definition, $\alpha^\star = \sup_{\tilde{\nu}} \min_P g_P^{\tilde{\nu}}$ is the optimal min–max value, so also $\min_P g_P^\nu \le \alpha^\star$. Hence $\min_P g_P^\nu = \alpha^\star$, i.e. $\nu \in \mathrm{BR}_i(\pi^{-i})$. $\qquad\square$

### E.4. Semi-Continuity of $G_i$

We now restore the dependence of all objects on $\pi_{-i}$ explicitly. For each $s \in \mathcal{S}$ and $(u, \alpha) \in \mathbb{R}^{\mathcal{S}} \times \mathbb{R}$, define

$$L_s^{\pi_{-i}}(\phi, p_s; u, \alpha) \triangleq \sum_{a_i\in\mathcal{A}_i} \phi(a_i)\big(r_i^{\pi_{-i}}(s, a_i) - \alpha + p_{s,a_i}^\top u\big), \tag{93}$$

$$F_s^{\pi_{-i}}(\phi; u, \alpha) \triangleq \inf_{p_s\in(\mathcal{P}^{\pi_{-i}})_s} L_s^{\pi_{-i}}(\phi, p_s; u, \alpha). \tag{94}$$

For each $\pi_{-i}$ let $(h_{\pi_{-i}}^\star, \alpha_{\pi_{-i}}^\star)$ be a constant-gain solution provided by Lemma E.2 for the DR-MDP $\mathsf{M}_i(\pi_{-i})$, with a fixed anchoring convention (e.g. $h_{\pi_{-i}}^\star(s^\circ) = 0$ for some reference state $s^\circ$).

We define the statewise greedy sets

$$M_s(\pi_{-i}) \triangleq \arg\max_{\phi\in\Delta(\mathcal{A}_i)} F_s^{\pi_{-i}}\big(\phi; h_{\pi_{-i}}^\star, \alpha_{\pi_{-i}}^\star\big), \tag{95}$$

$$G_i(\pi_{-i}) \triangleq \Big\{\Delta : \mathcal{S} \to \Delta(\mathcal{A}_i) \;:\; \Delta(\cdot \mid s) \in M_s(\pi_{-i}) \,\forall\, s\Big\}. \tag{96}$$

This reuses the notation $G_i(\pi_{-i})$ from (91); the previous section corresponds to a fixed $\pi_{-i}$, and here we track how $G_i$ varies with $\pi_{-i}$.

**Proposition E.7.** *There exists a universal constant $C_{\mathrm{span}} < \infty$ such that for every opponent policy $\pi_{-i}$ and every constant-gain solution $(h_{\pi_{-i}}^\star, \alpha_{\pi_{-i}}^\star)$ of $\mathsf{M}_i(\pi_{-i})$, $\mathrm{sp}\big(h_{\pi_{-i}}^\star\big) \le C_{\mathrm{span}}$.*

*Proof.* First, we consider a fixed policy $\pi_{-i}$. By Theorem 4 in (Wang & Si, 2025), it holds that any cluster point of $\{V_{\pi_{-i},\gamma}^\star(s) - V_{\pi_{-i},\gamma}^\star(s_0), \gamma \in (0,1)\}$ is a solution to the optimality equation. From Assumption 5.1, the equation only has one solution, thus this solution must be the unique cluster point of $\{V_{\pi_{-i},\gamma}^\star(s) - V_{\pi_{-i},\gamma}^\star(s_0), \gamma \in (0,1)\}$.

Moreover, under Assumption 5.1, Theorem 6 of (Wang & Si, 2025) shows that there exists some uniform upper bound $C$:

$$\sup_{\gamma \in (0,1)} \mathrm{sp}\left(V^{\star}_{\pi_{-i},\gamma}\right) \le C. \tag{97}$$

Now note that $V^{\star}_{\pi_{-i},\gamma}$ is continuous in $\pi_{-i}$ for any $\gamma$, thus its maximum $C_{\mathrm{Span}} = \sup_{\gamma \in (0,1),\pi} \mathrm{sp}\left(V^{\star}_{\pi_{-i},\gamma}\right)$ is achievable over the compact set $\prod_{j \ne i} \Delta(\mathcal{A}_j)$, it further holds that

$$\sup_{\gamma \in (0,1), \pi_{-i} \in \Pi_{-i}} \mathrm{sp}\left(V^{\star}_{\pi_{-i},\gamma}\right) \le C_{\mathrm{Span}}. \tag{98}$$

Consider a fixed policy $\pi_{-i}$. Note that for any $\epsilon$, there exists some sequence $\{\gamma_n\}$, such that $\mathrm{sp}\left(|V^{\star}_{\pi_{-i},\gamma_i}(s) - V^{\star}_{\pi_{-i},\gamma_i}(s_0) - h^*(s)\right) \le \epsilon$. Thus $\mathrm{sp}(h^*) \le \epsilon + \mathrm{sp}\left(V^{\star}_{\pi_{-i},\gamma_i}(s) - V^{\star}_{\pi_{-i},\gamma_i}\right) \le \epsilon + C_{\mathrm{Span}}$.

Thus it holds that for any $\pi_{-i}$,

$$\mathrm{sp}\left(h^{\star}_{\pi_{-i}}\right) \le C_{\mathrm{Span}}. \tag{99}$$

This hence completes the proof.

$\square$

**Lemma E.8.** *Fix a state $s \in \mathcal{S}$ and an action $a_i \in \mathcal{A}_i$. Let $\left(\pi^n_{-i}\right)_n$ be a sequence of opponent policies converging to $\pi_{-i}$ in the product topology, i.e. $\pi^n_{-i}(\cdot \mid s') \to \pi_{-i}(\cdot \mid s')$ in $\ell_1$ for all $s'$. Then, for each $(s, a_i)$, the induced ambiguity slices converge in Hausdorff distance:*

$$\left(\mathcal{P}^{\pi^n_{-i}}\right)_{s,a_i} \xrightarrow[n \to \infty]{\mathrm{H}} \left(\mathcal{P}^{\pi_{-i}}\right)_{s,a_i}.$$

*Consequently, for each $s$,*

$$\left(\mathcal{P}^{\pi^n_{-i}}\right)_{s} \xrightarrow[n \to \infty]{\mathrm{H}} \left(\mathcal{P}^{\pi_{-i}}\right)_{s}.$$

*Proof.* Each induced slice $(\mathcal{P}^{\pi_{-i}})_{s,a_i}$ is a Minkowski sum

$$\left(\mathcal{P}^{\pi_{-i}}\right)_{s,a_i} = \left\{ \sum_{a_{-i}} \pi_{-i}(a_{-i} \mid s)\, q^{(a_{-i})} \;:\; q^{(a_{-i})} \in \mathcal{P}^{(a_i, a_{-i})}_s \right\},$$

where the local sets $\mathcal{P}^{(a_i, a_{-i})}_s$ are fixed nonempty compact subsets of $\Delta(\mathcal{S})$. The map

$$(w, q) \mapsto \sum_{a_{-i}} w(a_{-i}) q^{(a_{-i})}$$

is continuous and bilinear, and the product of the $\mathcal{P}^{(a_i, a_{-i})}_s$ is compact. Standard results on continuous images of compact sets imply that the set-valued map $w \mapsto M(w)$ given by this Minkowski sum is continuous in Hausdorff distance. Since $\pi^n_{-i}(\cdot \mid s) \to \pi_{-i}(\cdot \mid s)$ in $\ell_1$, we obtain

$$\left(\mathcal{P}^{\pi^n_{-i}}\right)_{s,a_i} = M\left(\pi^n_{-i}(\cdot \mid s)\right) \xrightarrow{\mathrm{H}} M\left(\pi_{-i}(\cdot \mid s)\right) = \left(\mathcal{P}^{\pi_{-i}}\right)_{s,a_i}.$$

The convergence of the statewise products $\left(\mathcal{P}^{\pi^n_{-i}}\right)_s$ follows because finite products of Hausdorff-convergent compact sets converge in Hausdorff distance. $\square$

**Lemma E.9.** *Fix a state $s \in \mathcal{S}$. Let $(u_n, \alpha_n, \pi^n_{-i})_n$ be a sequence with $(u_n, \alpha_n) \in \mathbb{R}^{\mathcal{S}} \times \mathbb{R}$ and $\pi^n_{-i} \to \pi_{-i}$, and suppose $(u_n, \alpha_n) \to (u, \alpha)$. Then*

$$\sup_{\phi \in \Delta(\mathcal{A}_i)} \left| F^{\pi^n_{-i}}_s(\phi; u_n, \alpha_n) - F^{\pi_{-i}}_s(\phi; u, \alpha) \right| \xrightarrow[n \to \infty]{} 0.$$

*Proof.* Consider the joint map

$$(\phi, p_s, u, \alpha, \pi_{-i}) \;\mapsto\; L_s^{\pi_{-i}}(\phi, p_s; u, \alpha)$$

from

$$\Delta(\mathcal{A}_i) \times \bigcup_{\pi_{-i}} (\mathcal{P}^{\pi_{-i}})_s \times \mathbb{R}^{\mathcal{S}} \times \mathbb{R} \times \Pi_{-i}$$

to $\mathbb{R}$. On any bounded subset of $\mathbb{R}^{\mathcal{S}} \times \mathbb{R}$, this map is continuous. By Theorem E.7 and normalizing $u$ with $u(s^\circ) = 0$, (hence $\|u\|_\infty \le C_{\mathrm{Span}}$, otherwise $\mathrm{sp}(u) \ge \max |u_i| - 0 \ge C_{\mathrm{Span}}$), it suffices to restrict attention to the compact set

$$\Theta \;\triangleq\; \Big\{ (u, \alpha, \pi_{-i}) : \|u\|_\infty \le C_{\mathrm{span}}, \; \alpha \in [0,1], \; \pi_{-i} \in \Pi_{-i} \Big\}.$$

Lemma E.8 ensures that the feasible sets $(\mathcal{P}^{\pi_{-i}})_s$ vary continuously with $\pi_{-i}$ in Hausdorff distance.

The function $F_s^{\pi_{-i}}(\phi; u, \alpha)$ is the optimal value of the parametric minimization problem

$$\min_{p_s \in (\mathcal{P}^{\pi_{-i}})_s} L_s^{\pi_{-i}}(\phi, p_s; u, \alpha),$$

with continuous objective and compact feasible set that depends continuously on the parameter $(\phi, u, \alpha, \pi_{-i})$. Berge's Maximum Theorem (Berge, 1877) implies that the value map

$$(\phi, u, \alpha, \pi_{-i}) \mapsto F_s^{\pi_{-i}}(\phi; u, \alpha)$$

is continuous on the compact domain $\Delta(\mathcal{A}_i) \times \Theta$, and hence uniformly continuous. Therefore, whenever $(u_n, \alpha_n, \pi_{-i}^n) \to (u, \alpha, \pi_{-i})$,

$$\sup_{\phi \in \Delta(\mathcal{A}_i)} \left| F_s^{\pi_{-i}^n}(\phi; u_n, \alpha_n) - F_s^{\pi_{-i}}(\phi; u, \alpha) \right| \;\longrightarrow\; 0. \qquad \square$$

**Lemma E.10** ((Ross, 2013)). *Let $(f_n)_n$ be a sequence of continuous real-valued functions on the simplex $\Delta(\mathcal{A}_i)$ converging uniformly to $f$. For each $n$, let*

$$M_n = \arg \max_{\phi \in \Delta(\mathcal{A}_i)} f_n(\phi), \qquad M = \arg \max_{\phi \in \Delta(\mathcal{A}_i)} f(\phi),$$

*which are nonempty, compact, convex sets. If $\phi_n \in M_n$ and $\phi_n \to \phi$, then $\phi \in M$.*

*Proof.* This is standard. Because $f_n \to f$ uniformly and each $f_n$ attains its maximum on the compact set $\Delta(\mathcal{A}_i)$, the limit point $\phi$ must satisfy $f(\phi) \ge f(\psi)$ for all $\psi$, hence $\phi \in M$. $\qquad \square$

**Lemma E.11.** *Let $\pi_{-i}^n \to \pi_{-i}$ be a sequence of opponent policies. For each $n$, let $(u_n, \alpha_n)$ be a constant-gain solution for the DR-MDP $\mathsf{M}_i(\pi_{-i}^n)$, normalized by $u_n(s^\circ) = 0$. Then there exists a subsequence (still denoted $(u_n, \alpha_n)$) and a limit pair $(u, \alpha)$ such that*

$$(u_n, \alpha_n) \to (u, \alpha),$$

*and $(u, \alpha)$ is a constant-gain solution for $\mathsf{M}_i(\pi_{-i})$.*

*Proof.* By Theorem E.7, $\|u_n\|_\infty \le C_{\mathrm{span}}$ for all $n$, and $0 \le \alpha_n \le 1$ by bounded rewards. Thus $(u_n, \alpha_n)$ lies in a compact subset of $\mathbb{R}^{\mathcal{S}} \times [0, 1]$. By Bolzano–Weierstrass theorem (Ross, 2013), there exists a subsequence such that

$$(u_n, \alpha_n) \to (u, \alpha)$$

for some $(u, \alpha)$.

For each $n$ and each $s$, the constant-gain equation can be written as

$$u_n(s) = \sup_{\phi \in \Delta(\mathcal{A}_i)} F_s^{\pi_{-i}^n}(\phi; u_n, \alpha_n).$$

By Lemma E.9, for fixed $s$ the functions $\phi \mapsto F_s^{\pi_{-i}^n}(\phi; u_n, \alpha_n)$ converge uniformly to $\phi \mapsto F_s^{\pi_{-i}}(\phi; u, \alpha)$. Since $u_n(s) \to u(s)$ and $\sup_\phi F_n(\phi)$ converges to $\sup_\phi F(\phi)$ under uniform convergence, we obtain

$$u(s) = \sup_{\phi \in \Delta(\mathcal{A}_i)} F_s^{\pi_{-i}}(\phi; u, \alpha), \qquad \forall s \in \mathcal{S}.$$

That is, $(u, \alpha)$ is a constant-gain solution for $\mathsf{M}_i(\pi_{-i})$. $\qquad \square$

**Lemma E.12** (Convergence of maximizers under uniform convergence). *Let $K$ be a compact metric space and let $f_n, f : K \to \mathbb{R}$ be real-valued functions. Assume $f_n \to f$ uniformly on $K$. Let $x_n \in \arg\max_{x \in K} f_n(x)$ and suppose (along a subsequence) $x_n \to x \in K$. Then $x \in \arg\max_{x \in K} f(x)$.*

*Proof.* Fix any $y \in K$. Since $x_n$ maximizes $f_n$ on $K$, we have

$$f_n(x_n) \ \geq \ f_n(y) \qquad \forall n.$$

Add and subtract $f(x_n)$ and $f(y)$:

$$f(x_n) \ = \ f_n(x_n) + \big(f(x_n) - f_n(x_n)\big) \ \geq \ f_n(y) + \big(f(x_n) - f_n(x_n)\big).$$

Similarly, $f_n(y) = f(y) + \big(f_n(y) - f(y)\big)$, hence

$$f(x_n) \ \geq \ f(y) + \big(f_n(y) - f(y)\big) + \big(f(x_n) - f_n(x_n)\big).$$

Now take $\liminf_{n \to \infty}$ on both sides. Uniform convergence implies

$$\sup_{x \in K} |f_n(x) - f(x)| \to 0 \quad \implies \quad f_n(y) - f(y) \to 0, \quad f(x_n) - f_n(x_n) \to 0.$$

Also $x_n \to x$ and continuity of $f$ (or simply: in compact finite-dimensional sets, the $F_s$ in the paper are upper semicontinuous and this argument can be stated with upper limits; for the present application $f$ is continuous) gives $f(x_n) \to f(x)$. Therefore

$$f(x) \ \geq \ f(y).$$

Since $y \in K$ was arbitrary, $x$ is a maximizer of $f$ on $K$, i.e. $x \in \arg\max_K f$. $\qquad\square$

We can now establish upper semi-continuity of the greedy response map.

**Theorem E.13** (Upper hemicontinuity of $G_i$). *Fix player $i$ and consider the correspondence $\pi_{-i} \mapsto G_i(\pi_{-i})$ defined in (88). Then for every $\pi_{-i}$, $G_i$ is upper hemicontinuous.*

*Proof.* Let $\pi_{-i}^n \to \pi_{-i}$ and let $\Delta^n \in G_i(\pi_{-i}^n)$ such that $\Delta^n \to \Delta$ (componentwise, equivalently in the product topology on stationary policies). We show $\Delta \in G_i(\pi_{-i})$.

For each $n$, by definition of $G_i(\pi_{-i}^n)$ there exists a (normalized) constant-gain solution $(u_n, \alpha_n)$ of the induced robust MDP $\mathsf{M}_i(\pi_{-i}^n)$ used to form $F_s^{\pi_{-i}^n}(\cdot; u_n, \alpha_n)$, and such that for every state $s$,

$$\Delta^n(\cdot \mid s) \in \arg\max_{\phi \in \Delta(A_i)} F_s^{\pi_{-i}^n}(\phi; u_n, \alpha_n). \tag{100}$$

By Lemma E.11 (compactness and subsequence convergence of normalized constant-gain solutions), there is a subsequence (still indexed by $n$ for simplicity) such that

$$(u_n, \alpha_n) \to (u, \alpha),$$

where $(u, \alpha)$ is a constant-gain solution of $\mathsf{M}_i(\pi_{-i})$.

Fix an arbitrary state $s \in \mathcal{S}$ and define functions on the compact simplex $\Delta(A_i)$:

$$f_n(\phi) \ := \ F_s^{\pi_{-i}^n}(\phi; u_n, \alpha_n), \qquad f(\phi) \ := \ F_s^{\pi_{-i}}(\phi; u, \alpha).$$

By Lemma E.9, $f_n \to f$ uniformly on $\Delta(A_i)$.

From (100), we have $\phi_n := \Delta^n(\cdot \mid s) \in \arg\max_\phi f_n(\phi)$. Moreover $\phi_n \to \phi := \Delta(\cdot \mid s)$ by assumption. Applying Lemma E.12 yields

$$\Delta(\cdot \mid s) \in \arg\max_{\phi \in \Delta(A_i)} f(\phi) = \arg\max_{\phi \in \Delta(A_i)} F_s^{\pi_{-i}}(\phi; u, \alpha).$$

Since $s$ was arbitrary, this holds for all $s \in \mathcal{S}$, hence $\Delta \in G_i(\pi_{-i})$. Therefore the graph of $G_i$ is closed, and (together with compact values shown above) $G_i$ is upper hemicontinuous. $\qquad\square$

### E.5. Existence of a Robust Nash Equilibrium

We finally combine the previous results across all players. Let

$$\Pi \triangleq \prod_{k \in \mathcal{N}} \prod_{s \in \mathcal{S}} \Delta(\mathcal{A}_k)$$

be the compact convex set of stationary joint policies. For each player $i$, we have a set-valued map

$$G_i : \prod_{j \neq i} \prod_{s \in \mathcal{S}} \Delta(\mathcal{A}_j) \rightrightarrows \prod_{s \in \mathcal{S}} \Delta(\mathcal{A}_i)$$

with nonempty compact convex values and upper semi-continuous graph.

Define the product map

$$\mathcal{G} : \Pi \rightrightarrows \Pi, \qquad \mathcal{G}(\pi) \triangleq \prod_{i \in \mathcal{N}} G_i(\pi_{-i}),$$

where $\pi_{-i}$ denotes the opponent profile for player $i$. Standard results (e.g., Kakutani's fixed-point theorem) imply that $\mathcal{G}$ has a fixed point: there exists $\pi^\star \in \Pi$ such that $\pi^\star \in \mathcal{G}(\pi^\star)$.

By construction, for every $i$ we have

$$\pi_i^\star \in G_i(\pi_{-i}^\star) \subseteq BR_i(\pi_{-i}^\star),$$

where the inclusion is Theorem E.6. Thus $\pi^\star$ is a stationary robust Nash equilibrium of the DR-MG.

## F. Lipschitz Smoothness

In this section, we study the Lipschitz smoothness of the solutions to robust Bellman equations.

We first recall the seminorm and contraction property from (Xu et al., 2025b).

**Lemma F.1** (Lemma 3.2 in (Xu et al., 2025b))**.** *Under Assumption 4.1, there exists a seminorm $\|\cdot\|_{\mathsf{P}}$ on $\mathbb{R}^{|\mathcal{S}|}$ and a constant $\gamma \in (0,1)$ such that:*

1. *$\|x + c\mathbf{1}\|_{\mathsf{P}} = \|x\|_{\mathsf{P}}$ for all $x \in \mathbb{R}^{|\mathcal{S}|}$ and $c \in \mathbb{R}$ (constants are annihilated).*

2. *For every stationary policy profile $\pi$ and every scalar $g$, the operator $T_k^{\pi,g}$ is a $\gamma$-contraction on the quotient space $\mathbb{R}^{|\mathcal{S}|}/\mathrm{span}\{\mathbf{1}\}$ equipped with $\|\cdot\|_{\mathsf{P}}$, i.e.*

$$\left\|T_k^{\pi,g}v - T_k^{\pi,g}w\right\|_{\mathsf{P}} \leq \gamma\|v - w\|_{\mathsf{P}}$$

*for all $v, w$ with $v(s^\circ) = w(s^\circ)$.*

Since all norms on a finite-dimensional space are equivalent, there exist finite constants $C_{\infty \leftarrow \mathsf{P}}, C_{\mathsf{P} \leftarrow \infty}, C_{\mathrm{sp} \leftarrow \mathsf{P}} > 0$ such that for all zero-mean $x$,

$$\|x\|_\infty \leq C_{\infty \leftarrow \mathsf{P}}\|x\|_{\mathsf{P}}, \qquad \|x\|_{\mathsf{P}} \leq C_{\mathsf{P} \leftarrow \infty}\|x\|_\infty, \qquad \mathrm{sp}(x) \leq C_{\mathrm{sp} \leftarrow \mathsf{P}}\|x\|_{\mathsf{P}}. \tag{101}$$

**Lemma F.2.** *Let $R_{\max} \triangleq \max_{k,s,a} |r_k(s,a)|$. There exists a finite constant $B_V > 0$, depending only on $(R_{\max}, \gamma, C_{\infty \leftarrow \mathsf{P}}, C_{\mathsf{P} \leftarrow \infty})$ and the construction of $\|\cdot\|_{\mathsf{P}}$, such that for every player $k$ and every joint policy $\pi$,*

$$\|V_k^\pi\|_\infty \leq B_V,$$

*where $V_k^\pi$ is the uniquely anchored bias function satisfying $V_k^\pi(s^\circ) = 0$ and the fixed-policy robust Bellman equation with gain $g_k^\pi$.*

*Proof.* Fix a player $k$ and policy profile $\pi$, and let $(g_k^\pi, V_k^\pi)$ be a gain/bias solution of the robust Bellman equation, anchored by $V_k^\pi(s^\circ) = 0$. We may write

$$V_k^\pi = T_k^{\pi,g_k^\pi}(V_k^\pi).$$

Subtracting $T_k^{\pi,g_k^\pi}(0)$ and using Lemma F.1 gives

$$\left\|V_k^\pi - T_k^{\pi,g_k^\pi}(0)\right\|_{\mathsf{P}} = \left\|T_k^{\pi,g_k^\pi}(V_k^\pi) - T_k^{\pi,g_k^\pi}(0)\right\|_{\mathsf{P}} \le \gamma \|V_k^\pi\|_{\mathsf{P}}.$$

Hence

$$(1-\gamma)\|V_k^\pi\|_{\mathsf{P}} \le \left\|T_k^{\pi,g_k^\pi}(0)\right\|_{\mathsf{P}}.$$

By definition,

$$T_k^{\pi,g_k^\pi}(0)(s) = \sum_a \pi(a \mid s)\big(r_k(s,a) - g_k^\pi + \sigma_s^a(0)\big) = r_k^\pi(s) - g_k^\pi,$$

because $\sigma_s^a(0) = 0$. Thus

$$\left\|T_k^{\pi,g_k^\pi}(0)\right\|_\infty = \|r_k^\pi - g_k^\pi \mathbf{1}\|_\infty \le |g_k^\pi| + \max_s |r_k^\pi(s)| \le 2R_{\max},$$

using $|g_k^\pi| \le R_{\max}$ (see Lemma F.3 below) and $|r_k^\pi(s)| \le R_{\max}$. By norm equivalence (101),

$$\left\|T_k^{\pi,g_k^\pi}(0)\right\|_{\mathsf{P}} \le C_{\mathsf{P}\leftarrow\infty}\left\|T_k^{\pi,g_k^\pi}(0)\right\|_\infty \le 2C_{\mathsf{P}\leftarrow\infty}R_{\max}.$$

Combining,

$$\|V_k^\pi\|_{\mathsf{P}} \le \frac{2C_{\mathsf{P}\leftarrow\infty}}{1-\gamma}R_{\max},$$

and then

$$\|V_k^\pi\|_\infty \le C_{\infty\leftarrow\mathsf{P}}\|V_k^\pi\|_{\mathsf{P}} \le \frac{2C_{\infty\leftarrow\mathsf{P}}C_{\mathsf{P}\leftarrow\infty}}{1-\gamma}R_{\max} \triangleq B_V.$$

$\square$

**Lemma F.3.** *For every player $k$ and joint policy $\pi$, the robust average reward $g_k^\pi$ satisfies*

$$|g_k^\pi| \le R_{\max}.$$

*Proof.* By definition of the average reward under any kernel $P$,

$$g_k^{\pi,P} = \lim_{T\to\infty} \frac{1}{T}\sum_{t=0}^{T-1} \mathbb{E}[r_k(S_t, A_t)],$$

and $|r_k(s,a)| \le R_{\max}$ for all $(s,a)$. Thus for any $P$, $|g_k^{\pi,P}| \le R_{\max}$, and taking the worst-case kernel can only decrease the value in absolute value, so $|g_k^\pi| = |\min_P g_k^{\pi,P}| \le R_{\max}$. $\square$

We next control the sensitivity of the Bellman operator with respect to the policy.

**Lemma F.4.** *Fix a state $s$ and let $\pi, \tilde\pi$ be two joint policies. Then their joint action distributions at $s$ satisfy*

$$\sum_{a\in\mathcal{A}(s)} \left|\prod_{j=1}^K \tilde\pi_j(a_j \mid s) - \prod_{j=1}^K \pi_j(a_j \mid s)\right| \le \sum_{j=1}^K \left\|\tilde\pi_j(\cdot \mid s) - \pi_j(\cdot \mid s)\right\|_1.$$

*Proof.* Use the telescoping identity for products:

$$\prod_{j=1}^K x_j - \prod_{j=1}^K y_j = \sum_{m=1}^K (x_m - y_m)\Big(\prod_{j<m} y_j\Big)\Big(\prod_{j>m} x_j\Big),$$

and apply the triangle inequality. Summing over $a$ and using that for any fixed $m$ the remaining factors are nonnegative and sum to at most 1 when marginalizing over $a$ yields the claim. $\square$

**Lemma F.5.** *Fix a player $k$ and let $B_V$ be as in Lemma F.2. For any $v \in \mathbb{R}^{|\mathcal{S}|}$ with $\|v\|_\infty \le B_V$ and any scalar $g$ with $|g| \le R_{\max}$, define the robust one-step return $F_k(s, a; v) = r_k(s, a) + \sigma_s^a(v)$ and the operator $T_k^{\pi,g}$ as above. Then there exists a constant $C_\Pi > 0$, depending only on $(K, |\mathcal{S}|, A_{\max}, R_{\max}, B_V)$ with $A_{\max} \triangleq \max_{k,s} |\mathcal{A}_k|$, such that for all policies $\pi, \tilde{\pi}$,*

$$\left\| T_k^{\tilde{\pi},g}(v) - T_k^{\pi,g}(v) \right\|_\infty \le C_\Pi \|\tilde{\pi} - \pi\|_2.$$

*Proof.* For each state $s$,

$$\left| T_k^{\tilde{\pi},g}(v)(s) - T_k^{\pi,g}(v)(s) \right|$$
$$= \left| \sum_a \left( \tilde{\pi}(a \mid s) - \pi(a \mid s) \right) \left( r_k(s, a) - g + \sigma_s^a(v) \right) \right|.$$

Because $|r_k(s, a)| \le R_{\max}$, $|g| \le R_{\max}$ and $|\sigma_s^a(v)| \le \|v\|_\infty \le B_V$, we have $|r_k(s, a) - g + \sigma_s^a(v)| \le 2R_{\max} + B_V$. Thus

$$\left| T_k^{\tilde{\pi},g}(v)(s) - T_k^{\pi,g}(v)(s) \right| \le (2R_{\max} + B_V) \sum_a \left| \tilde{\pi}(a \mid s) - \pi(a \mid s) \right|.$$

By Lemma F.4 the last sum is bounded by $\sum_{j=1}^K \|\tilde{\pi}_j(\cdot \mid s) - \pi_j(\cdot \mid s)\|_1$. Summing over $s$ and applying Cauchy–Schwarz, we get

$$\|T_k^{\tilde{\pi},g}(v) - T_k^{\pi,g}(v)\|_\infty \le C_\Pi \|\tilde{\pi} - \pi\|_2$$

with $C_\Pi$ absorbing the dimension-dependent constants. $\qquad \square$

We can now state and prove the Lipschitz continuity of the evaluation map.

**Theorem F.6.** *Under Assumption 4.1 and Lemmas F.1–F.5, for each player $k$ there exist finite constants $L_V, L_g > 0$ (independent of $\pi$) such that for any two joint policies $\pi, \tilde{\pi}$,*

$$\|V_k^{\tilde{\pi}} - V_k^\pi\|_\infty \le L_V \|\tilde{\pi} - \pi\|_2, \qquad |g_k^{\tilde{\pi}} - g_k^\pi| \le L_g \|\tilde{\pi} - \pi\|_2.$$

*Proof.* Fix $k$ and policies $\pi, \tilde{\pi}$. Let $(g_k^\pi, V_k^\pi)$ and $(g_k^{\tilde{\pi}}, V_k^{\tilde{\pi}})$ be the gain/bias pairs solving the respective robust Bellman equations, anchored by $V_k^\pi(s^\circ) = V_k^{\tilde{\pi}}(s^\circ) = 0$. Write $\Delta v \triangleq V_k^{\tilde{\pi}} - V_k^\pi$ and $\Delta g \triangleq g_k^{\tilde{\pi}} - g_k^\pi$.

*Step 1: Lipschitz continuity of the bias.* Using the fixed-point relations $V_k^\pi = T_k^{\pi,g_k^\pi}(V_k^\pi)$ and $V_k^{\tilde{\pi}} = T_k^{\tilde{\pi},g_k^{\tilde{\pi}}}(V_k^{\tilde{\pi}})$, we have

$$\Delta v = T_k^{\tilde{\pi},g_k^{\tilde{\pi}}}(V_k^{\tilde{\pi}}) - T_k^{\pi,g_k^\pi}(V_k^\pi)$$
$$= \left[ T_k^{\tilde{\pi},g_k^{\tilde{\pi}}}(V_k^{\tilde{\pi}}) - T_k^{\pi,g_k^{\tilde{\pi}}}(V_k^{\tilde{\pi}}) \right] + \left[ T_k^{\pi,g_k^{\tilde{\pi}}}(V_k^{\tilde{\pi}}) - T_k^{\pi,g_k^{\tilde{\pi}}}(V_k^\pi) \right]$$
$$+ \left[ T_k^{\pi,g_k^{\tilde{\pi}}}(V_k^\pi) - T_k^{\pi,g_k^\pi}(V_k^\pi) \right].$$

We bound each term in $\| \cdot \|_{\mathsf{P}}$.

The second term is controlled by Lemma F.1:

$$\left\| T_k^{\pi,g_k^{\tilde{\pi}}}(V_k^{\tilde{\pi}}) - T_k^{\pi,g_k^{\tilde{\pi}}}(V_k^\pi) \right\|_{\mathsf{P}} \le \gamma \|\Delta v\|_{\mathsf{P}}.$$

For the first term we apply Lemma F.5 with $v = V_k^{\tilde{\pi}}$ and $g = g_k^{\tilde{\pi}}$. By Lemmas F.2 and F.3, $\|V_k^{\tilde{\pi}}\|_\infty \le B_V$ and $|g_k^{\tilde{\pi}}| \le R_{\max}$, so the conditions of Lemma F.5 hold and

$$\left\| T_k^{\tilde{\pi},g_k^{\tilde{\pi}}}(V_k^{\tilde{\pi}}) - T_k^{\pi,g_k^{\tilde{\pi}}}(V_k^{\tilde{\pi}}) \right\|_\infty \le C_\Pi \|\tilde{\pi} - \pi\|_2.$$

Using (101),

$$\left\| T_k^{\tilde{\pi},g_k^{\tilde{\pi}}}(V_k^{\tilde{\pi}}) - T_k^{\pi,g_k^{\tilde{\pi}}}(V_k^{\tilde{\pi}}) \right\|_{\mathsf{P}} \le C_{\mathsf{P} \leftarrow \infty} C_\Pi \|\tilde{\pi} - \pi\|_2.$$

The third term is the effect of changing $g$ at fixed $\pi$ and $v$:

$$T_k^{\pi,g_k^{\tilde{\pi}}}(V_k^\pi)(s) - T_k^{\pi,g_k^\pi}(V_k^\pi)(s) = \sum_a \pi(a \mid s) \left[ -g_k^{\tilde{\pi}} + g_k^\pi \right] = (g_k^\pi - g_k^{\tilde{\pi}}),$$

so

$$T_k^{\pi, g_k^{\tilde{\pi}}}(V_k^{\pi}) - T_k^{\pi, g_k^{\pi}}(V_k^{\pi}) = -\Delta g \, \mathbf{1}.$$

By construction $\|\mathbf{1}\|_{\mathsf{P}} = 0$ (Lemma F.1), so this term vanishes in $\|\cdot\|_{\mathsf{P}}$.

Combining these three bounds we get

$$\|\Delta v\|_{\mathsf{P}} \le C_{\mathsf{P}\leftarrow\infty} C_\Pi \|\tilde{\pi} - \pi\|_2 + \gamma \|\Delta v\|_{\mathsf{P}},$$

hence

$$(1 - \gamma)\|\Delta v\|_{\mathsf{P}} \le C_{\mathsf{P}\leftarrow\infty} C_\Pi \|\tilde{\pi} - \pi\|_2.$$

Therefore

$$\|\Delta v\|_{\mathsf{P}} \le \frac{C_{\mathsf{P}\leftarrow\infty} C_\Pi}{1 - \gamma} \|\tilde{\pi} - \pi\|_2.$$

Using $\|\cdot\|_\infty \le C_{\infty\leftarrow\mathsf{P}}\|\cdot\|_{\mathsf{P}}$ we obtain

$$\|V_k^{\tilde{\pi}} - V_k^{\pi}\|_\infty \le L_V \|\tilde{\pi} - \pi\|_2,$$

with

$$L_V \triangleq \frac{C_{\infty\leftarrow\mathsf{P}} C_{\mathsf{P}\leftarrow\infty} C_\Pi}{1 - \gamma}.$$

*Step 2: Lipschitz continuity of the gain.* Using the Bellman equation at the anchor state $s^\circ$, we can write

$$g_k^{\pi} = \sum_a \pi(a \mid s^\circ)\Big(r_k(s^\circ, a) + \sigma_{s^\circ}^a(V_k^{\pi})\Big) - V_k^{\pi}(s^\circ)$$

$$= \sum_a \pi(a \mid s^\circ) F_k(s^\circ, a; V_k^{\pi}),$$

because $V_k^{\pi}(s^\circ) = 0$ by anchoring. Similarly,

$$g_k^{\tilde{\pi}} = \sum_a \tilde{\pi}(a \mid s^\circ) F_k(s^\circ, a; V_k^{\tilde{\pi}}).$$

Hence

$$|g_k^{\tilde{\pi}} - g_k^{\pi}| \le \left| \sum_a \big(\tilde{\pi}(a \mid s^\circ) - \pi(a \mid s^\circ)\big) F_k(s^\circ, a; V_k^{\tilde{\pi}}) \right|$$

$$+ \left| \sum_a \pi(a \mid s^\circ)\big(F_k(s^\circ, a; V_k^{\tilde{\pi}}) - F_k(s^\circ, a; V_k^{\pi})\big) \right|$$

$$\triangleq T_1 + T_2.$$

For $T_1$, using $|F_k(s^\circ, a; v)| \le R_{\max} + \|v\|_\infty \le R_{\max} + B_V$ and Lemma F.4 at $s^\circ$,

$$T_1 \le (R_{\max} + B_V) \sum_a |\tilde{\pi}(a \mid s^\circ) - \pi(a \mid s^\circ)| \le C_1 \|\tilde{\pi} - \pi\|_2,$$

for some constant $C_1$ depending only on the game dimensions.

For $T_2$, note that for any $v, w$,

$$\big|F_k(s^\circ, a; v) - F_k(s^\circ, a; w)\big| = \big|\sigma_{s^\circ}^a(v) - \sigma_{s^\circ}^a(w)\big| \le \|v - w\|_\infty,$$

because each $q \in \mathcal{U}(s^\circ, a)$ is a probability vector and $|\min_q q^\top v - \min_q q^\top w| \le \max_q |q^\top(v - w)| \le \|v - w\|_\infty$. Thus

$$T_2 \le \|V_k^{\tilde{\pi}} - V_k^{\pi}\|_\infty \le L_V \|\tilde{\pi} - \pi\|_2.$$

Altogether,

$$|g_k^{\tilde{\pi}} - g_k^{\pi}| \le (C_1 + L_V)\|\tilde{\pi} - \pi\|_2 \triangleq L_g \|\tilde{\pi} - \pi\|_2.$$

This completes the proof. □

---

**Algorithm 2** Robust Nash-Iteration

---

   **Initialization:** $h_i = 0, \forall i \in \mathcal{N}$
   **while** TRUE **do**
     **for** all $i \in \mathcal{N}$ **do**
       $h_i^0 \leftarrow h_i$
       **for** all $s, a \in \mathcal{S} \times \mathcal{A}$ **do**
         $Q_i(s, a, h_i^0) \leftarrow r_i(s, a) + \sigma_{\mathcal{P}_s^a}(h_i^0)$
         $\pi(s) \leftarrow \mathbf{NE}(Q_i(s, a, h_i^0))$
         $h_i(s) \leftarrow \mathbb{E}_{\pi(s)}[Q_i(s, a, h_i^0)]$
       **end for**
     **end for**
     **if** $\max_i\{\mathrm{sp}(h_i - h_i^0)\} = 0$ **then**
       BREAK
     **end if**
   **end while**
   **Output:** $\pi$

---

## G. Nash Iteration

Throughout this section, we adopt the following notational conventions for the robust Bellman operators.

For each state-joint–action pair $(s, a)$ the transition row is uncertain in a nonempty, convex, compact set $\mathcal{U}(s, a) \subset \Delta(\mathcal{S})$. For any $v \in \mathbb{R}^{|\mathcal{S}|}$ define the local support function

$$\sigma_s^a(v) \triangleq \min_{q \in \mathcal{U}(s,a)} q^\top v,$$

and for a joint policy $\pi$ define

$$\sigma_s^\pi(v) \triangleq \min_{p \in \mathcal{P}_s^\pi} p^\top v, \qquad \mathcal{P}_s^\pi \triangleq \Big\{ \sum_a \pi(a \mid s) q(\cdot \mid s, a) : q(\cdot \mid s, a) \in \mathcal{U}(s, a) \Big\}.$$

Thus $\sigma_s^\pi(v) = \sum_a \pi(a \mid s)\sigma_s^a(v)$, and by compactness the minima defining $\sigma_s^a$ and $\sigma_s^\pi$ are always attained.

For each player $k$, the robust one-step return is

$$F_k(s, a; v) \triangleq r_k(s, a) + \sigma_s^a(v),$$

and for a fixed policy profile $\pi$ and scalar $g$ the (player–$k$) robust evaluation operator is

$$(T_k^{\pi,g} v)(s) \triangleq \sum_a \pi(a \mid s)\big[ r_k(s, a) - g + \sigma_s^a(v) \big].$$

With this notation, the fixed-policy robust average-reward Bellman equation for player $k$ under $\pi$ reads

$$V_k^\pi(s) = (T_k^{\pi, g_k^\pi} V_k^\pi)(s), \qquad s \in \mathcal{S},$$

where $(g_k^\pi, V_k^\pi)$ is a gain/bias pair as in Theorem Theorem 4.6 (uniqueness up to constants).

We always anchor biases by choosing a reference state $s^\circ \in \mathcal{S}$ and imposing $V_k^\pi(s^\circ) = 0$.

**Assumption G.1.** For each state $s$ and each vector of bias estimates $h = (h_1, \ldots, h_N)$, consider the one-stage game with payoff functions

$$Q_k(s, a; h_k) \triangleq r_k(s, a) + \sigma_s^a(h_k), \qquad a \in \mathcal{A}(s), \ k \in \mathcal{N}.$$

At every state $s$, the algorithm selects either

1. a Nash equilibrium of this stage game that is globally optimal for all players in the sense that every player attains his maximal robust one-step return at the selected equilibrium, or

2. a saddle point of the stage game, i.e. a pure/mixed action profile $(\pi_1(\cdot \mid s), \ldots, \pi_N(\cdot \mid s))$ such that no player can unilaterally deviate to improve her robust one-step return.

The first part of our assumption is necessary for convergence under average reward Markov games (Li, 2003; Hu & Wellman, 2003; Littman, 2001; Bowling & Veloso, 2001; Bowling, 2000). The second assumption is a technical assumption to ensure convergence, and can be removed by modifying the update rule by a multi-step operator, see (Wang et al., 2023c; Xu et al., 2025b).

**Theorem G.2** (Convergence of robust Nash iteration). *Suppose Assumption 4.1 and Assumption G.1 hold. Then the robust Nash-iteration algorithm (Algorithm 2) converges, from any initialization $h^0$, to an average-reward robust Nash equilibrium of the DR-MG.*

*Proof.* We split the proof into two parts.

**(i) Any limit point (if convergent) of Nash iteration is a robust NE.** Consider one iteration of Algorithm 2. For each player $i$, let $h_i^0$ be the current bias estimate, define

$$Q_i(s, a; h_i^0) \triangleq r_i(s, a) + \sigma_s^a(h_i^0),$$

and let $\pi(\cdot \mid s)$ be a Nash equilibrium of the one-stage game with payoffs $\{Q_i(s, \cdot; h_i^0)\}_{i \in \mathcal{N}}$, chosen according to Assumption G.1. The algorithm then sets

$$h_i(s) \triangleq \mathbb{E}_{a \sim \pi(\cdot \mid s)}[Q_i(s, a; h_i^0)], \qquad s \in \mathcal{S}.$$

The stopping condition is

$$\max_{i \in \mathcal{N}} \mathrm{sp}(h_i - h_i^0) = 0,$$

where $\mathrm{sp}(x) = \max_s x(s) - \min_s x(s)$ is the span seminorm. The condition $\mathrm{sp}(h_i - h_i^0) = 0$ is equivalent to

$$h_i - h_i^0 = c_i \mathbf{1}$$

for some scalar $c_i \in \mathbb{R}$, where $\mathbf{1}$ is the all-ones vector.

Assume the algorithm terminates at a pair $(\pi, h)$ satisfying the stopping condition. Then for each player $i$ and state $s$,

$$\begin{aligned} h_i(s) &= \mathbb{E}_{\pi(\cdot \mid s)}[r_i(s, a) + \sigma_s^a(h_i^0)] \\ &= \mathbb{E}_{\pi(\cdot \mid s)}[r_i(s, a) + \sigma_s^a(h_i - c_i \mathbf{1})]. \end{aligned}$$

Using the translation-invariance of the support function, for any $v$ and scalar $c$,

$$\sigma_s^a(v - c\mathbf{1}) = \min_{q \in \mathcal{U}(s, a)} q^\top (v - c\mathbf{1}) = \min_q (q^\top v - c) = \sigma_s^a(v) - c,$$

we conclude that

$$h_i(s) = \mathbb{E}_{\pi(\cdot \mid s)}[r_i(s, a) + \sigma_s^a(h_i) - c_i] = \mathbb{E}_{\pi(\cdot \mid s)}[r_i(s, a) + \sigma_s^a(h_i)] - c_i.$$

Rearranging,

$$h_i(s) + c_i = \sum_a \pi(a \mid s)\big(r_i(s, a) + \sigma_s^a(h_i)\big), \qquad s \in \mathcal{S}. \tag{102}$$

Define $g_i \triangleq c_i$ and set $V_i \triangleq h_i$. Then (102) can be written as

$$V_i(s) = \sum_a \pi(a \mid s)\big(r_i(s, a) - g_i + \sigma_s^a(V_i)\big), \qquad s \in \mathcal{S}. \tag{103}$$

Thus $(g_i, V_i)$ solves the fixed-policy robust Bellman equation (6) for player $i$ under the joint policy $\pi$. By Theorem 4.6, for each $i$ there exists a worst-case kernel $P \in \mathcal{P}$ such that $g_i = g_{P,i}^\pi$ and $V_i$ coincides with the robust bias $h_{P,i}^\pi$ up to an additive constant; in particular, we may choose the anchor so that $V_i = h_{P,i}^\pi$ and

$$h_{P,i}^\pi(s) + g_{P,i}^\pi = \sum_a \pi(a \mid s)\big(r_i(s, a) + \sigma_s^a(h_{P,i}^\pi)\big), \qquad s \in \mathcal{S}. \tag{104}$$

Now suppose, for contradiction, that $\pi$ is not a robust average-reward Nash equilibrium. Then there exist a player $i$ and an alternative stationary policy $\mu_i$ such that

$$g_{P,i}^{(\pi_{-i},\mu_i)} > g_{P,i}^{(\pi_{-i},\pi_i)} = g_{P,i}^{\pi}. \tag{105}$$

By Theorem 4.6 applied to the induced DR-MDP $\mathsf{M}_i(\pi_{-i})$ with policy $\mu_i$, there exists a robust bias $h_i^{\mu}$ such that

$$h_i^{\mu}(s) + g_{P,i}^{(\pi_{-i},\mu_i)} = \sum_a (\pi_{-i},\mu_i)(a \mid s)\big(r_i(s,a) + \sigma_s^a(h_i^{\mu})\big), \qquad s \in \mathcal{S}. \tag{106}$$

Combining (104) and (106), we have

$$r_i^{(\pi_{-i},\mu_i)} + \sigma^{(\pi_{-i},\mu_i)}(h_i^{\mu}) - h_i^{\mu}$$
$$= g_{P,i}^{(\pi_{-i},\mu_i)} > g_{P,i}^{\pi} = r_i^{\pi} + \sigma^{\pi}(h_{P,i}^{\pi}) - h_{P,i}^{\pi}, \tag{107}$$

where the inequality uses (105), and the equalities are vector versions of (104)–(106).

On the other hand, at each state $s$, the algorithm selects a Nash equilibrium $\pi(\cdot \mid s)$ of the one-stage game with payoffs $Q_i(s,a;h_{P,i}^{\pi}) = r_i(s,a) + \sigma_s^a(h_{P,i}^{\pi})$. The NE property implies that for any alternative mixed action $\mu_i(\cdot \mid s)$,

$$\mathbb{E}_{a\sim(\pi_{-i},\mu_i)}Q_i(s,a;h_{P,i}^{\pi}) \leq \mathbb{E}_{a\sim\pi(\cdot|s)}Q_i(s,a;h_{P,i}^{\pi}).$$

Equivalently,

$$r_i^{(\pi_{-i},\mu_i)} + \sigma^{(\pi_{-i},\mu_i)}(h_{P,i}^{\pi}) \leq r_i^{\pi} + \sigma^{\pi}(h_{P,i}^{\pi}). \tag{108}$$

Subtracting $h_{P,i}^{\pi}$ from both sides of (108) and combining with (107), we obtain

$$r_i^{\pi} + \sigma^{\pi}(h_{P,i}^{\pi}) - h_{P,i}^{\pi}$$
$$< r_i^{(\pi_{-i},\mu_i)} + \sigma^{(\pi_{-i},\mu_i)}(h_i^{\mu}) - h_i^{\mu}$$
$$\leq \sigma^{(\pi_{-i},\mu_i)}(h_i^{\mu}) - h_i^{\mu} + r_i^{\pi} + \sigma^{\pi}(h_{P,i}^{\pi}) - \sigma^{(\pi_{-i},\mu_i)}(h_{P,i}^{\pi}). \tag{109}$$

Rearranging (109) yields

$$h_i^{\mu} - h_{P,i}^{\pi} < \sigma^{(\pi_{-i},\mu_i)}(h_i^{\mu}) - \sigma^{(\pi_{-i},\mu_i)}(h_{P,i}^{\pi}). \tag{110}$$

Now apply Lemma G.4 to the convex function $v \mapsto \sigma^{(\pi_{-i},\mu_i)}(v)$ with $V_1 = h_i^{\mu}$ and $V_2 = h_{P,i}^{\pi}$. Lemma G.4 ensures the existence of a kernel $P \in \mathcal{P}^{(\pi_{-i},\mu_i)}$ such that

$$\sigma^{(\pi_{-i},\mu_i)}(h_i^{\mu}) - \sigma^{(\pi_{-i},\mu_i)}(h_{P,i}^{\pi}) = P(h_i^{\mu} - h_{P,i}^{\pi}).$$

Substituting this into (110) gives

$$h_i^{\mu} - h_{P,i}^{\pi} < P(h_i^{\mu} - h_{P,i}^{\pi}) \quad \text{componentwise.} \tag{111}$$

Let $x \triangleq h_i^{\mu} - h_{P,i}^{\pi}$. Then (111) reads $x < Px$ componentwise.

By Assumption 4.1, every kernel in $\mathcal{P}$ (and thus $P$) is irreducible. For an irreducible stochastic matrix $P$, any vector $x$, with $x \leq Px$ componentwise, must be constant, i.e. $x = c\mathbf{1}$ for some scalar $c$. A standard proof goes as follows: let $m = \min_s x(s)$ and choose $s_0$ with $x(s_0) = m$. From $x(s_0) \leq \sum_{s'} P(s'|s_0)x(s')$ and $x(s') \geq m$, we deduce that $x(s') = m$ whenever $P(s'|s_0) > 0$. By irreducibility, every state is reachable from $s_0$ under some power $P^n$, and the same argument applied iteratively shows $x(s) = m$ for all $s$. Applying the same reasoning to $-x$ if needed yields the claim for $x \geq Px$. Thus

$$h_i^{\mu} - h_{P,i}^{\pi} = c_i\mathbf{1} \tag{112}$$

for some scalar $c_i$.

Plugging (112) into the Bellman equations (104)–(106) and using the translation-invariance of the support function again, we see that the corresponding gains must coincide:

$$g_{P,i}^{(\pi_{-i},\mu_i)} = g_{P,i}^{\pi},$$

which contradicts (105). Therefore no such profitable deviation $\mu_i$ exists, and $\pi$ is a robust average-reward Nash equilibrium.

**(ii) Contraction of the Nash operator and convergence.** Let $h = (h_1, \ldots, h_N) \in \mathbb{R}^{N|\mathcal{S}|}$ and define

$$T_i(h)(s) \triangleq \mathbb{E}_{\pi(\cdot|s)}\big[r_i(s,a) + \sigma_s^a(h_i)\big], \qquad T(h) \triangleq (T_1(h), \ldots, T_N(h)),$$

where at each state $s$ the policy $\pi(\cdot \mid s)$ is a Nash equilibrium of the one-stage game with payoffs $Q_i(s,a;h_i)$, selected according to Assumption G.1.

We show that $T$ is a contraction in the product span seminorm $\|h\|_{\mathrm{sp}} \triangleq \max_i \mathrm{sp}(h_i)$, where $\mathrm{sp}(x) = \max_s x(s) - \min_s x(s)$.

Fix $h, h'$ and a player $i$ and state $s$.

*Case 1 (global optimality).* If the NE selection at $s$ is globally optimal in the sense of Assumption G.1, then

$$\begin{aligned}
T_i(h)(s) - T_i(h')(s) &= \max_{\pi(\cdot|s)} \mathbb{E}_{\pi(\cdot|s)}\big[r_i(s,a) + \sigma_s^a(h_i)\big] \\
&\quad - \max_{\pi'(\cdot|s)} \mathbb{E}_{\pi'(\cdot|s)}\big[r_i(s,a) + \sigma_s^a(h_i')\big] \\
&\leq \mathbb{E}_{\pi(\cdot|s)}\big[\sigma_s^a(h_i) - \sigma_s^a(h_i')\big] \\
&\leq \max_a \big(\sigma_s^a(h_i) - \sigma_s^a(h_i')\big).
\end{aligned} \tag{113}$$

*Case 2 (saddle point).* If the NE is a saddle point, Assumption G.1 ensures that for any joint policy $\pi'$ differing only in player $i$'s component,

$$\begin{aligned}
T_i(h)(s) - T_i(h')(s) &= \mathbb{E}_{\pi(\cdot|s)}\big[r_i(s,a) + \sigma_s^a(h_i)\big] \\
&\quad - \mathbb{E}_{\pi'(\cdot|s)}\big[r_i(s,a) + \sigma_s^a(h_i')\big] \\
&\leq \mathbb{E}_{(\pi'_{-i},\pi_i)(\cdot|s)}\big[r_i(s,a) + \sigma_s^a(h_i)\big] \\
&\quad - \mathbb{E}_{(\pi'_{-i},\pi_i)(\cdot|s)}\big[r_i(s,a) + \sigma_s^a(h_i')\big] \\
&\leq \max_a \big(\sigma_s^a(h_i) - \sigma_s^a(h_i')\big).
\end{aligned} \tag{114}$$

By symmetry (swapping $h$ and $h'$), we also obtain the lower bound

$$T_i(h)(s) - T_i(h')(s) \geq \min_a \big(\sigma_s^a(h_i) - \sigma_s^a(h_i')\big). \tag{115}$$

Combining (113)–(115) we have, for each $(i, s)$,

$$\min_a \big(\sigma_s^a(h_i) - \sigma_s^a(h_i')\big) \leq T_i(h)(s) - T_i(h')(s) \leq \max_a \big(\sigma_s^a(h_i) - \sigma_s^a(h_i')\big).$$

Taking maxima and minima over $s$ and using Lemma G.4 below (which shows that the map $v \mapsto \sigma_s^a(v)$ has the same Dobrushin coefficient as a stochastic matrix in the span seminorm), we obtain

$$\mathrm{sp}\big(T_i(h) - T_i(h')\big) \leq \tau \, \mathrm{sp}(h_i - h_i'),$$

for some $\tau < 1$ depending only on the uncertainty set and the irreducibility constant (Lemma G.3). Taking the maximum over $i$ yields

$$\|T(h) - T(h')\|_{\mathrm{sp}} \leq \tau \|h - h'\|_{\mathrm{sp}}.$$

Thus $T$ is a contraction in the product span seminorm.

By the Banach fixed-point theorem, $T$ has a unique fixed point $h^*$ (modulo constants); iterating Algorithm 2 (without stopping) converges to $h^*$ from any initialization $h^0$. The associated per-state Nash selection $\pi^*(\cdot \mid s)$ at $h^*$ is exactly the policy considered in part (i), hence is a robust average-reward Nash equilibrium, and the claimed convergence follows. $\qquad \square$

## G.1. Technical Lemmas

**Lemma G.3.** *For a stochastic matrix $P$ define the ergodicity coefficient*

$$\tau(P) \triangleq \frac{1}{2} \max_{s,s' \in \mathcal{S}} \sum_{s'' \in \mathcal{S}} \big|P(s'' \mid s) - P(s'' \mid s')\big|.$$

*Then for all $x, y \in \mathbb{R}^{|\mathcal{S}|}$,*

$$\mathrm{sp}(Px - Py) \;\leq\; \tau(P)\, \mathrm{sp}(x - y).$$

*If $P$ is irreducible, there exists $\bar{\tau} < 1$ (depending only on $P$) such that $\tau(P) \leq \bar{\tau}$.*

*Proof.* This is the standard Dobrushin contraction in the span seminorm; see e.g. Sec. 16.3 of (Puterman, 2014) or (Filar & Vrieze, 1996). For completeness, one can write $Px - Py = P(x - y)$ and note that taking $x - y$ supported on two states attaining the maximum and minimum values attains the worst-case contraction factor, which is precisely $\tau(P)$ as defined above. Irreducibility implies $\tau(P) < 1$. □

**Lemma G.4** (Support-function difference as a kernel action). *Assume each uncertainty set $\mathcal{P}_s^a$ is nonempty, compact, and convex.*

(a) **State-action level.** *For any $v_1, v_2 \in \mathbb{R}^{|\mathcal{S}|}$ and any fixed pair $(s, a)$, there exists a vector $p_{s,a}^\star \in \mathcal{P}_s^a$ such that*

$$\sigma_s^a(v_1) - \sigma_s^a(v_2) = (p_{s,a}^\star)^\top (v_1 - v_2).$$

(b) **Statewise operator and global kernel.** *Fix a policy $\pi$, and define $\sigma^\pi$ as above. Then for any $v_1, v_2 \in \mathbb{R}^{|\mathcal{S}|}$ there exists a stochastic kernel $P^\star \in \mathcal{P}$ (constructed row by row) such that*

$$\sigma^\pi(v_1) - \sigma^\pi(v_2) = P^\star(v_1 - v_2).$$

*In particular, when we view $\sigma^a(\cdot)$ or $\sigma^\pi(\cdot)$ as statewise support operators, their difference can always be written as the action of a suitable transition kernel on $v_1 - v_2$.*

*Proof.* We begin with part (a) for a fixed $(s, a)$.

Since $\mathcal{P}_s^a$ is compact and $p \mapsto p^\top v$ is continuous, the minima defining $\sigma_s^a(v_1)$ and $\sigma_s^a(v_2)$ are attained. Choose

$$p_{s,a}^1 \in \arg\min_{p \in \mathcal{P}_s^a} p^\top v_1, \qquad p_{s,a}^2 \in \arg\min_{p \in \mathcal{P}_s^a} p^\top v_2.$$

Denote

$$\sigma_1 \triangleq \sigma_s^a(v_1) = (p_{s,a}^1)^\top v_1, \qquad \sigma_2 \triangleq \sigma_s^a(v_2) = (p_{s,a}^2)^\top v_2, \qquad \delta \triangleq v_1 - v_2.$$

From the optimality of $p_{s,a}^1$ and $p_{s,a}^2$ we have

$$(p_{s,a}^1)^\top v_1 \leq (p_{s,a}^2)^\top v_1, \qquad (p_{s,a}^2)^\top v_2 \leq (p_{s,a}^1)^\top v_2.$$

Rewriting,

$$(p_{s,a}^1)^\top \delta = (p_{s,a}^1)^\top v_1 - (p_{s,a}^1)^\top v_2 \;\leq\; \sigma_1 - \sigma_2,$$
$$(p_{s,a}^2)^\top \delta = (p_{s,a}^2)^\top v_1 - (p_{s,a}^2)^\top v_2 \;\geq\; \sigma_1 - \sigma_2.$$

Thus

$$(p_{s,a}^1)^\top \delta \;\leq\; \sigma_s^a(v_1) - \sigma_s^a(v_2) \;\leq\; (p_{s,a}^2)^\top \delta.$$

Because $\mathcal{P}_s^a$ is convex, the entire line segment

$$p_{s,a}^\lambda \;\triangleq\; \lambda p_{s,a}^1 + (1 - \lambda) p_{s,a}^2, \qquad \lambda \in [0, 1],$$

lies in $\mathcal{P}_s^a$. Define

$$h(\lambda) \;\triangleq\; (p_{s,a}^\lambda)^\top \delta = \lambda (p_{s,a}^1)^\top \delta + (1 - \lambda)(p_{s,a}^2)^\top \delta.$$

The map $\lambda \mapsto h(\lambda)$ is continuous and linear in $\lambda$, and its image on $[0, 1]$ is exactly the interval

$$\left[ (p_{s,a}^1)^\top \delta, \; (p_{s,a}^2)^\top \delta \right].$$

Since $\sigma_s^a(v_1) - \sigma_s^a(v_2)$ lies between these two endpoints, there exists $\lambda^\star \in [0,1]$ such that

$$h(\lambda^\star) = \sigma_s^a(v_1) - \sigma_s^a(v_2).$$

Setting $p_{s,a}^\star \triangleq p_{s,a}^{\lambda^\star} \in \mathcal{P}_s^a$ yields

$$\sigma_s^a(v_1) - \sigma_s^a(v_2) = (p_{s,a}^\star)^\top \delta,$$

which proves part (a).

Fix a policy $\pi$ and a state $s$. The aggregated set $\mathcal{P}_s^\pi$ is, by definition, the image of the product $\prod_a \mathcal{P}_s^a$ under the continuous affine map

$$(p_s^a)_a \longmapsto \sum_a \pi(a \mid s)\, p_s^a,$$

so $\mathcal{P}_s^\pi$ is also nonempty, compact, and convex. Repeating the argument of Steps 1–3 with $\mathcal{P}_s^\pi$ in place of $\mathcal{P}_s^a$, we obtain for each $s$ a vector $p_s^\star \in \mathcal{P}_s^\pi$ such that

$$\sigma_s^\pi(v_1) - \sigma_s^\pi(v_2) = (p_s^\star)^\top (v_1 - v_2).$$

Now define a (row-stochastic) kernel $P^\star$ on $S$ by setting its $s$-th row to be $P^\star(\cdot \mid s) \triangleq p_s^\star$. By construction $P^\star(\cdot \mid s) \in \mathcal{P}_s^\pi$ for every $s$, so $P^\star$ is an admissible kernel under $\pi$. Stacking the statewise equalities yields

$$\sigma^\pi(v_1) - \sigma^\pi(v_2) = P^\star(v_1 - v_2),$$

as claimed. $\qquad\square$

# H. Robust TD-Descent Analysis

In this section, we develop our studies on our robust TD based algorithm. Throughout, we assume the DR-MG satisfies Assumption 4.1 (irreducibility), so that all induced DR-MDPs are irreducible and admit unique average rewards, and the corresponding robust Bellman equation is solvable.

## H.1. Computational Complexity

We first discuss the computational complexity of Algorithm 1. Recall in the algorithm, the following steps are executed:

- For each agent $k$, solve the robust Bellman equation in (6) for the induced DR-MDP to get $(g_k^n, V_k^n)$

- For all $(k, s, a_k)$, compute robust TD errors $\Omega_k^{R,n}(s, a_k)$ and gradient components $\Phi_k^n(s, a_k)$

- Form the TD direction $D^n$ and project to get $\pi^{n+1}$

**Robust Bellman step.** Solving a single-agent robust Bellman equation can be done in $T_{RB}(S, A_k)$ time which is polynomial in $S, A_k$ (Xu et al., 2025b). Hence the total complexity is $\mathcal{O}(N T_{RB}(S, A_{\max}))$.

**TD error step.** For each $(k, s, a_k)$, the computation cost is $\mathcal{O}\left(\frac{A}{A_k} C_\sigma\right)$, where $C_\sigma$ is the computational cost for solving the support function. Notably, for commonly uncertainty sets defined through divergence functions, $C_\sigma$ is also polynomial (Iyengar, 2005). Thus the total complexity is $\mathcal{O}(SNAC_\sigma)$.

**Gradient computation step.** Notably, $\Delta_{\mathrm{R}}$ is a finite sum of terms of the form $\alpha \prod_i \pi_i(a_i|s)$, thus computing its Moreau envelop and gradient is itself a polynomial in the $\pi_i$. Computing them at most involves sums over joint actions similar to the TD-error computation, so an upper bound is $\mathcal{O}(SNA)$.

**Update step.** From the definition of $D^n$, its computation cost $\mathcal{O}(S\acute{s}\mathsf{M}_i A_i)$; And the following projection step to the product of simplexes has a complexity of $\mathcal{O}(A_k \log A_k)$ for each $(k, s)$. Thus the total complexity is $\mathcal{O}\left(S \sum_k A_k \log A_k\right)$.

Hence, the total computational complexity for each step is

$$O\Big(N T_{\mathrm{RB}}(S, A_{\max}, \delta) + SNAC_\sigma\Big), \tag{116}$$

which is polynomial in the size of the DR-MG parameters.

## H.2. Robust TD Errors, Global TD Gap, and Feasible Set

For a joint policy $\pi \in \Pi$ and player $k \in \mathcal{N}$, the induced DR-MDP $M_k(\pi_{-k})$ is

$$M_k(\pi_{-k}) = \left(\mathcal{S}, \mathcal{A}_k, \mathcal{P}^{\pi_{-k}}, r_k^{\pi_{-k}}\right),$$

where

$$r_k^{\pi_{-k}}(s, a_k) \triangleq \sum_{a_{-k} \in \mathcal{A}_{-k}} \pi_{-k}(a_{-k} \mid s) \, r_k(s, a_k, a_{-k}), \tag{117}$$

$$\mathcal{P}_{s,a_k}^{\pi_{-k}} \triangleq \left\{ \sum_{a_{-k} \in \mathcal{A}_{-k}} \pi_{-k}(a_{-k} \mid s) \, p(\cdot \mid s, a_k, a_{-k}) : p(\cdot \mid s, a_k, a_{-k}) \in \mathcal{P}_s^{(a_k, a_{-k})} \right\}. \tag{118}$$

For $V_k \in \mathbb{R}^{\mathcal{S}}$ and a convex ambiguity slice $U \subseteq \Delta(\mathcal{S})$ we denote the robust support function $\sigma_U(V_k) \triangleq \min_{p \in U} p^\top V_k$.

We then define our robust TD notions.

**Definition H.1** (Robust TD error). Fix a joint policy $\pi$ and a player $k \in \mathcal{N}$. For $s \in \mathcal{S}$ and $a_k \in \mathcal{A}_k$, and given $(g_k, V_k) \in \mathbb{R} \times \mathbb{R}^{\mathcal{S}}$, define:

$$Q_k^R(s, a_k; \pi_{-k} \mid V_k) \triangleq \sum_{a_{-k} \in \mathcal{A}_{-k}} \pi_{-k}(a_{-k} \mid s) \left( r_k(s, a_k, a_{-k}) + \sigma_{\mathcal{P}_s^{(a_k, a_{-k})}}(V_k) \right), \tag{119}$$

$$\Omega_k^R(s, a_k; \pi_{-k} \mid g_k, V_k) \triangleq g_k + V_k(s) - Q_k^R(s, a_k; \pi_{-k} \mid V_k). \tag{120}$$

**Definition H.2.** For $(\pi, g, V) \in \Pi \times \mathbb{R}^{\mathcal{N}} \times (\mathbb{R}^{\mathcal{S}})^{\mathcal{N}}$ define the *global robust TD gap*

$$\Delta_R(\pi, g, V) \triangleq \sum_{k \in \mathcal{N}} \sum_{s \in \mathcal{S}} \sum_{a_k \in \mathcal{A}_k} \pi_k(a_k \mid s) \, \Omega_k^R(s, a_k; \pi_{-k} \mid g_k, V_k), \tag{121}$$

and the *feasible set*

$$\mathcal{F} \triangleq \left\{ (\pi, g, V) : \ \pi \in \Pi, \ \Omega_k^R(s, a_k; \pi_{-k} \mid g_k, V_k) \geq 0, \ \forall k, s, a_k \right\}. \tag{122}$$

We first show that robust Nash equilibria can be characterized by a complementarity condition on the robust TD errors.

**Proposition H.3** (Robust TD complementarity). *Suppose Assumption 4.1 holds. A stationary joint policy $\pi = (\pi_k)_{k \in \mathcal{N}}$ is a stationary robust Nash equilibrium if and only if there exist scalars $g_k \in \mathbb{R}$ and vectors $V_k \in \mathbb{R}^{|S|}$, $k \in \mathcal{N}$, such that for every $k \in \mathcal{N}$, $s \in S$ and $a_k \in \mathcal{A}_k$,*

$$\Omega_k^R(s, a_k; \pi_{-k} \mid g_k, V_k) \geq 0, \tag{123a}$$

$$\sum_{a_k \in \mathcal{A}_k} \pi_k(a_k \mid s) \, \Omega_k^R(s, a_k; \pi_{-k} \mid g_k, V_k) = 0, \tag{123b}$$

$$\sum_{a_k \in \mathcal{A}_k} \pi_k(a_k \mid s) = 1, \quad \pi_k(a_k \mid s) \geq 0. \tag{123c}$$

*Proof.* ($\Rightarrow$) Assume $\pi$ is a stationary robust Nash equilibrium. Fix a player $k$ and the opponents' policy $\pi_{-k}$, and consider the induced DR-MDP $M_k(\pi_{-k})$. By Assumption 4.1 this induced DR-MDP also satisfies irreducibility.

Let $(g_k, V_k)$ be a gain/bias pair for an optimal robust policy in $M_k(\pi_{-k})$.[3] By optimality, $(g_k, V_k)$ satisfies the optimal robust Bellman equation

$$V_k(s) = \max_{a_k \in \mathcal{A}_k} \left\{ r_k^{\pi_{-k}}(s, a_k) - g_k + \sigma_{\mathcal{P}_{s,a_k}^{\pi_{-k}}}(V_k) \right\}, \qquad s \in S, \tag{124}$$

where $r_k^{\pi_{-k}}$ and $\mathcal{P}_{s,a_k}^{\pi_{-k}}$ as above. By the definition of $Q_k^R$, (124) is equivalent to

$$g_k + V_k(s) = \max_{a_k \in \mathcal{A}_k} Q_k^R(s, a_k; \pi_{-k} \mid V_k), \qquad s \in S. \tag{125}$$

---

[3] Such a pair exists by the robust average-reward Bellman results in Theorem 4.6 of the main text.

Using the TD error definition (H.1), we can rewrite (125) as

$$\Omega_k^{\mathrm{R}}(s, a_k; \pi_{-k} \mid g_k, V_k) = g_k + V_k(s) - Q_k^{\mathrm{R}}(s, a_k; \pi_{-k} \mid V_k) \ \geq \ 0 \quad \text{for all } a_k \in \mathcal{A}_k,$$

with equality for those $a_k$ that achieve the maximum in (125). Thus (123a) holds, and moreover,

$$g_k + V_k(s) = \max_{a_k} Q_k^{\mathrm{R}}(s, a_k; \pi_{-k} \mid V_k) = Q_k^{\mathrm{R}}(s, a_k; \pi_{-k} \mid V_k) \quad \text{whenever } \pi_k(a_k \mid s) > 0.$$

Equivalently, $\Omega_k^{\mathrm{R}}(s, a_k; \pi_{-k} \mid g_k, V_k) = 0$ whenever $\pi_k(a_k \mid s) > 0$, so

$$\sum_{a_k} \pi_k(a_k \mid s) \Omega_k^{\mathrm{R}}(s, a_k; \pi_{-k} \mid g_k, V_k) = 0.$$

This is exactly (123b). The simplex constraints (123c) hold by definition of a stationary policy.

($\Leftarrow$) Now suppose $\pi$ and $(g_k, V_k)$ satisfy (123). Fix $k$ and $\pi_{-k}$, and consider again the induced DR-MDP $M_k(\pi_{-k})$.

For any $s \in S$ and $a_k \in \mathcal{A}_k$, the inequality (123a) can be rewritten as

$$g_k + V_k(s) \geq Q_k^{\mathrm{R}}(s, a_k; \pi_{-k} \mid V_k),$$

so $g_k + V_k(s)$ is an upper bound on the robust one-step return over all actions $a_k$ at state $s$. Combining this with (123b), we obtain

$$0 = \sum_{a_k} \pi_k(a_k \mid s) \, \Omega_k^{\mathrm{R}}(s, a_k; \pi_{-k} \mid g_k, V_k)$$

$$= \sum_{a_k} \pi_k(a_k \mid s) \left( g_k + V_k(s) - Q_k^{\mathrm{R}}(s, a_k; \pi_{-k} \mid V_k) \right)$$

$$= g_k + V_k(s) - \sum_{a_k} \pi_k(a_k \mid s) \, Q_k^{\mathrm{R}}(s, a_k; \pi_{-k} \mid V_k).$$

Hence

$$g_k + V_k(s) = \sum_{a_k} \pi_k(a_k \mid s) \, Q_k^{\mathrm{R}}(s, a_k; \pi_{-k} \mid V_k).$$

Since each term $Q_k^{\mathrm{R}}(s, a_k; \cdot)$ is upper bounded by $g_k + V_k(s)$, their convex combination equals $g_k + V_k(s)$ only if

$$Q_k^{\mathrm{R}}(s, a_k; \pi_{-k} \mid V_k) = g_k + V_k(s) \quad \text{whenever } \pi_k(a_k \mid s) > 0.$$

That is, $\pi_k(\cdot \mid s)$ puts positive mass only on actions that maximize $Q_k^{\mathrm{R}}(s, \cdot; \pi_{-k} \mid V_k)$, and thus $g_k + V_k$ satisfies the optimal robust Bellman equation (125).

By the average-reward theory for DR-MDPs (Theorem 4.6 in the main text), this implies that $\pi_k$ is an optimal robust policy in $M_k(\pi_{-k})$, i.e. a robust best response to $\pi_{-k}$. Since the argument holds for every $k \in \mathcal{N}$, the joint policy $\pi$ is a stationary robust Nash equilibrium. $\qquad\square$

The complementarity conditions (123) naturally lead to a global TD gap functional.

**Lemma H.4.** *For any $(\pi, g, V) \in \mathcal{F}$ we have $\Delta_{\mathrm{R}}(\pi, g, V) \geq 0$. Moreover, $\Delta_{\mathrm{R}}(\pi, g, V) = 0$ if and only if the complementarity conditions* (123) *hold.*

*Proof.* If $(\pi, g, V) \in \mathcal{F}$ then $\Omega_k^{\mathrm{R}}(s, a_k; \pi_{-k} \mid g_k, V_k) \geq 0$ and $\pi_k(a_k \mid s) \geq 0$ for all $k, s, a_k$. Hence every summand in (11) is nonnegative and $\Delta_{\mathrm{R}}(\pi, g, V) \geq 0$.

Conversely, if $\Delta_{\mathrm{R}}(\pi, g, V) = 0$, then every term in the finite sum (11) must be zero:

$$\pi_k(a_k \mid s) \Omega_k^{\mathrm{R}}(s, a_k; \pi_{-k} \mid g_k, V_k) = 0, \quad \forall k, s, a_k.$$

Summing over $a_k$ yields

$$\sum_{a_k} \pi_k(a_k \mid s) \, \Omega_k^{\mathrm{R}}(s, a_k; \pi_{-k} \mid g_k, V_k) = 0, \quad \forall k, s,$$

which is exactly (123b). The inequalities (123a) and simplex constraints (123c) already hold by the definition of $\mathcal{F}$. The converse direction is immediate from Definition H.2. $\qquad\square$

Combining Lemma H.4 with Proposition H.3 yields:

**Corollary H.5.** *Under Assumption 4.1:*

1. *For any $(\pi, g, V) \in \mathcal{F}$, $\Delta_{\mathrm{R}}(\pi, g, V) \geq 0$.*

2. *A policy profile $\pi$ is a stationary robust Nash equilibrium if and only if there exist $(g, V)$ such that $(\pi, g, V) \in \mathcal{F}$ and $\Delta_{\mathrm{R}}(\pi, g, V) = 0$.*

Thus, on the feasible set $\mathcal{F}$, robust NE are precisely the (global) minimizers of $\Delta_{\mathrm{R}}$, and finding a robust NE is equivalent to find the feasible solution of $\Delta_{\mathrm{R}} = 0$

### H.3. Smoothed DR-MGs and NE

In this section, we introduce the smoothed DR-MG as a proxy of the vanilla DR-MG.

For $v \in \mathbb{R}^S$ and some uncertainty set $\mathcal{P}$, denote the robust support (worst-case expectation) functional

$$\sigma_{\mathcal{P}}(v) := \min_{p \in \mathcal{P}} \langle p, v \rangle.$$

This map is concave and Lipschitz but may be nonsmooth.

We introduce a standard Moreau-type smoothing through the convex support function. Define the (convex) support function

$$h_{\mathcal{P}}(u) := \max_{p \in \mathcal{P}} \langle p, u \rangle.$$

Then $\sigma_{\mathcal{P}}(v) = -h_{\mathcal{P}}(-v)$. For $\lambda > 0$, define the Moreau envelope of $h_{\mathcal{P}}$:

$$(h_{\mathcal{P}})_\lambda(u) := \min_{w \in \mathbb{R}^S} \left\{ h_{\mathcal{P}}(w) + \frac{1}{2\lambda} \|w - u\|_2^2 \right\}.$$

We define the *smoothed robust expectation*:

$$\sigma_{\mathcal{P}}^\lambda(v) := -(h_{\mathcal{P}})_\lambda(-v).$$

**Lemma H.6.** *Let $Z \subset \mathbb{R}^d$ be convex.*

*(i) (Product rule) Let $f, g : Z \to \mathbb{R}$ be differentiable and assume:*

$$\|\nabla f(x) - \nabla f(y)\| \leq L_f \|x - y\|, \quad \|\nabla g(x) - \nabla g(y)\| \leq L_g \|x - y\|, \quad \forall x, y \in Z.$$

*Assume also $\sup_Z |f| \leq B_f$, $\sup_Z |g| \leq B_g$, $\sup_Z \|\nabla f\| \leq B_{\nabla f}$, $\sup_Z \|\nabla g\| \leq B_{\nabla g}$. Then $h := fg$ satisfies*

$$\|\nabla h(x) - \nabla h(y)\| \leq \left( B_f L_g + B_g L_f + B_{\nabla f} B_{\nabla g} \right) \|x - y\|, \quad \forall x, y \in Z.$$

*(ii) (Scalar composition) Let $\varphi : \mathbb{R} \to \mathbb{R}$ be $C^1$ with Lipschitz derivative:*

$$|\varphi'(u) - \varphi'(v)| \leq L_\varphi |u - v|, \quad \forall u, v \in \mathbb{R}.$$

*Let $g : Z \to \mathbb{R}$ be differentiable with $\|\nabla g(x) - \nabla g(y)\| \leq L_g \|x - y\|$ and $\sup_Z \|\nabla g\| \leq B_{\nabla g}$. Let $B_{\varphi'} := \sup_{t \in g(Z)} |\varphi'(t)| < \infty$. Then $h := \varphi \circ g$ satisfies*

$$\|\nabla h(x) - \nabla h(y)\| \leq \left( B_{\varphi'} L_g + L_\varphi B_{\nabla g}^2 \right) \|x - y\|, \quad \forall x, y \in Z.$$

*Proof.* (i) $\nabla(fg) = f\nabla g + g\nabla f$, so for any $x, y \in Z$,

$$\begin{aligned}
\|\nabla(fg)(x) - \nabla(fg)(y)\| &= \|f(x)\nabla g(x) - f(y)\nabla g(y) + g(x)\nabla f(x) - g(y)\nabla f(y)\| \\
&\leq \|f(x)(\nabla g(x) - \nabla g(y))\| + \|(f(x) - f(y))\nabla g(y)\| \\
&\quad + \|g(x)(\nabla f(x) - \nabla f(y))\| + \|(g(x) - g(y))\nabla f(y)\|.
\end{aligned}$$

Bound the first and third terms by $B_f L_g \|x - y\|$ and $B_g L_f \|x - y\|$. For the second term, use the mean-value bound $|f(x) - f(y)| \le B_{\nabla f} \|x - y\|$ (since $f$ is differentiable and $\sup_Z \|\nabla f\| < \infty$ on the compact sets used in the paper), giving $\le B_{\nabla f} B_{\nabla g} \|x - y\|$. Similarly the fourth term is $\le B_{\nabla g} B_{\nabla f} \|x - y\|$ (same bound). Combining yields the stated constant.

(ii) $\nabla(\varphi \circ g)(x) = \varphi'(g(x)) \nabla g(x)$, hence

$$\|\nabla(\varphi \circ g)(x) - \nabla(\varphi \circ g)(y)\| = \|\varphi'(g(x))\nabla g(x) - \varphi'(g(y))\nabla g(y)\|$$
$$\le |\varphi'(g(x))| \cdot \|\nabla g(x) - \nabla g(y)\| + |\varphi'(g(x)) - \varphi'(g(y))| \cdot \|\nabla g(y)\|.$$

Bound the first term by $B_{\varphi'} L_g \|x - y\|$. For the second term, use Lipschitzness of $\varphi'$ and the fact $|g(x) - g(y)| \le B_{\nabla g} \|x - y\|$ to obtain

$$|\varphi'(g(x)) - \varphi'(g(y))| \le L_\varphi |g(x) - g(y)| \le L_\varphi B_{\nabla g} \|x - y\|,$$

so the second term is $\le L_\varphi B_{\nabla g}^2 \|x - y\|$. $\qquad \square$

**Lemma H.7** (Smoothness and Lipschitz gradient of $M_\lambda$). *Fix $\lambda > 0$ and $\rho > 0$. On the compact set $Z = \Pi \times \mathcal{G} \times \mathcal{V}$, the merit function $M_\lambda$ defined in (117) is continuously differentiable and has Lipschitz gradient: there exists $L_\lambda < \infty$ such that for all $z, z' \in Z$,*

$$\|\nabla M_\lambda(z) - \nabla M_\lambda(z')\| \le L_\lambda \|z - z'\|.$$

*Proof.* Write $z = (\pi, g, V) \in Z$.

**Step 0: Uniform bounds on $Z$.** Because $Z$ is compact, define finite constants

$$B_g := \max_{g \in \mathcal{G}} \|g\|_\infty, \qquad B_V := \max_{V \in \mathcal{V}} \|V\|_\infty, \qquad R_{\max} := \max_{k,s,a} |r_k(s,a)|.$$

Also $\pi \in \Pi$ implies each coordinate $\pi_k(a_k \mid s) \in [0, 1]$.

For any ambiguity slice $\mathcal{P} \subset \Delta(\mathcal{S})$ and any $v \in \mathbb{R}^{|\mathcal{S}|}$,

$$|\sigma_{\mathcal{P},\lambda}(v)| \le \max_{p \in \Delta(\mathcal{S})} |p^\top v| \le \|v\|_\infty.$$

Thus on $Z$, $|\sigma_{\mathcal{P},\lambda}(V_k)| \le B_V$.

**Step 1: $\Omega_{k,\lambda}^{RS}$ is $C^{1,1}$ on $Z$.** Fix $(k, s, a_k)$. Recall (116):

$$\Omega_{k,\lambda}^{RS}(s, a_k; \pi_{-k} \mid g_k, V_k) = g_k + V_k(s) - Q_{k,\lambda}^{RS}(s, a_k; \pi_{-k} \mid V_k),$$

where

$$Q_{k,\lambda}^{RS}(s, a_k; \pi_{-k} \mid V_k) = \sum_{a_{-k}} \pi_{-k}(a_{-k} \mid s)\Big(r_k(s, a_k, a_{-k}) + \sigma_{\mathcal{P}_s^{(a_k, a_{-k})}, \lambda}(V_k)\Big).$$

By Lemma H.6, for each fixed $(s, a_k, a_{-k})$ the map $V_k \mapsto \sigma_{\mathcal{P}_s^{(a_k, a_{-k})}, \lambda}(V_k)$ is $C^1$ and has Lipschitz gradient with constant at most $1/\lambda$. The dependence of $Q_{k,\lambda}^{RS}$ on $\pi_{-k}$ is multilinear (polynomial) in the coordinates of $\pi$ and hence has globally bounded second derivatives on the compact product simplex $\Pi$; therefore $\pi \mapsto Q_{k,\lambda}^{RS}$ has a Lipschitz gradient on $\Pi$ (with some finite constant depending only on $(N, |\mathcal{S}|, |A|)$ and bounds above). Combining:

- $(g_k, V_k(s)) \mapsto g_k + V_k(s)$ is affine (hence $C^{1,1}$ with zero Lipschitz-gradient constant);

- $\pi \mapsto \pi_{-k}(a_{-k} \mid s)$ is smooth with Lipschitz gradient on $\Pi$;

- $V_k \mapsto \sigma_{\mathcal{P},\lambda}(V_k)$ is $C^{1,1}$ with constant $\le 1/\lambda$ by Lemma H.6;

and using Lemma H.6(i) (product rule) and finite summation over $a_{-k}$, we conclude: there exists a finite constant $L_{\Omega,\lambda}$ such that for all $z, z' \in Z$,

$$\|\nabla \Omega_{k,\lambda}^{RS}(z) - \nabla \Omega_{k,\lambda}^{RS}(z')\| \le L_{\Omega,\lambda} \|z - z'\|. \tag{126}$$

In particular, $\Omega_{k,\lambda}^{RS}$ is $C^1$ and $\nabla \Omega_{k,\lambda}^{RS}$ is bounded on $Z$; denote $B_{\nabla \Omega} := \sup_{z \in Z} \|\nabla \Omega_{k,\lambda}^{RS}(z)\| < \infty$.

Also, by the bounds in Step 0,

$$|\Omega_{k,\lambda}^{RS}(z)| \le |g_k| + |V_k(s)| + \sum_{a_{-k}} \pi_{-k}(a_{-k} \mid s)(|r_k| + |\sigma_\lambda|) \le B_g + B_V + (R_{\max} + B_V),$$

so $|\Omega_{k,\lambda}^{RS}(z)| \le B_\Omega$ for some finite $B_\Omega$ independent of $z \in Z$.

**Step 2: Squared complementarity terms have Lipschitz gradients.** Define the scalar map

$$h_{k,s,a_k}(z) := \pi_k(a_k \mid s)\, \Omega_{k,\lambda}^{RS}(s, a_k; \pi_{-k} \mid g_k, V_k).$$

The coordinate map $z \mapsto \pi_k(a_k \mid s)$ is linear in $z$ (hence $C^{1,1}$ with zero Lipschitz-gradient constant), and $z \mapsto \Omega_{k,\lambda}^{RS}$ is $C^{1,1}$ by (126). By Lemma H.6(i), there exists $L_h < \infty$ such that $\nabla h_{k,s,a_k}$ is Lipschitz on $Z$:

$$\|\nabla h_{k,s,a_k}(z) - \nabla h_{k,s,a_k}(z')\| \le L_h \|z - z'\|, \quad \forall z, z' \in Z.$$

Moreover $\sup_Z |h_{k,s,a_k}| \le B_\Omega$ (since $\pi_k(\cdot \mid s) \in [0,1]$) and $\sup_Z \|\nabla h_{k,s,a_k}\| < \infty$ by continuity on compactness.

Now consider $\varphi_1(x) := \frac{1}{2}x^2$. Then $\varphi_1$ is $C^1$ and $\varphi_1'(x) = x$ is 1-Lipschitz. Applying Lemma H.6(ii) with $g = h_{k,s,a_k}$ yields that

$$m_{k,s,a_k}(z) := \tfrac{1}{2} h_{k,s,a_k}(z)^2$$

has Lipschitz gradient on $Z$ with some finite constant $L_m < \infty$.

**Step 3: Negative-part penalty terms have Lipschitz gradients.** Define $\varphi_2(x) := \frac{\rho}{2}[x]_-^2$. This scalar function is $C^1$ everywhere with derivative

$$\varphi_2'(x) = \rho\, x\, \mathbf{1}\{x < 0\},$$

and $\varphi_2'$ is globally $\rho$-Lipschitz on $\mathbb{R}$ (piecewise linear with slope $\rho$ on $(-\infty, 0)$ and $0$ on $(0, \infty)$). Apply Lemma H.6(ii) with $g(z) = \Omega_{k,\lambda}^{RS}(z)$. Using (126) and boundedness of $\nabla \Omega_{k,\lambda}^{RS}$ on $Z$, we obtain that

$$p_{k,s,a_k}(z) := \tfrac{\rho}{2}\big[\Omega_{k,\lambda}^{RS}(z)\big]_-^2$$

has Lipschitz gradient on $Z$ with some finite constant $L_p < \infty$.

**Step 4: Sum over finitely many indices.** Finally, $M_\lambda$ in (117) is a finite sum of the $m_{k,s,a_k}$ and $p_{k,s,a_k}$ terms. A finite sum of functions with Lipschitz gradients has Lipschitz gradient, with constant given by the sum of constants. Thus there exists $L_\lambda < \infty$ such that

$$\|\nabla M_\lambda(z) - \nabla M_\lambda(z')\| \le L_\lambda \|z - z'\|, \qquad \forall z, z' \in Z.$$

This also implies $M_\lambda$ is continuously differentiable on $Z$. $\qquad\square$

We further use the smoothed operator to define smoothed robust TD errors.

Fix a player $k$ and an opponents policy $\pi_{-k}$. For $v \in \mathbb{R}^S$ define the smoothed robust $Q$-value as

$$Q_{k,\lambda}^{RS}(s, a_k; \pi_{-k} \mid v) := \sum_{a_{-k} \in \mathcal{A}_{-k}} \pi_{-k}(a_{-k} \mid s)\Big(r_k(s, a_k, a_{-k}) + \sigma_{\mathcal{P}_s^{(a_k, a_{-k})}}^\lambda(v)\Big).$$

Given $(g_k, V_k) \in \mathbb{R} \times \mathbb{R}^S$ define the (smoothed) robust TD error as

$$\Omega_{k,\lambda}^{RS}(s, a_k; \pi_{-k} \mid g_k, V_k) := g_k + V_k(s) - Q_{k,\lambda}^{RS}(s, a_k; \pi_{-k} \mid V_k). \tag{127}$$

## H.4. Complementarity Merit

We then extend the TD-error characterization to the smoothed ones.

For scalar $x$, define the negative part $[x]_- = \max\{0, -x\}$. Define the merit function

$$\mathcal{L}_\lambda(\pi, g, V) := \frac{1}{2} \sum_{k \in \mathcal{N}} \sum_{s \in \mathcal{S}} \sum_{a_k \in \mathcal{A}_k} \Big(\pi_k(a_k|s)\, \Omega_{k,\lambda}^{RS}(s, a_k; \pi_{-k} \mid g_k, V_k)\Big)^2 + \frac{\rho}{2} \sum_{k,s,a_k} \big(\big[\Omega_{k,\lambda}^{RS}(s, a_k; \pi_{-k} \mid g_k, V_k)\big]_-\big)^2, \tag{128}$$

where $\rho > 0$ is a fixed penalty parameter.

We then have the following result.

**Lemma H.8** (Merit zeros imply feasibility and complementarity). *If $\mathcal{L}_\lambda(\pi, g, V) = 0$, then for every $(k, s, a_k)$:*

$$\Omega_{k,\lambda}^{RS}(s, a_k; \pi_{-k} \mid g_k, V_k) \geq 0, \qquad \pi_k(a_k|s)\, \Omega_{k,\lambda}^{RS}(s, a_k; \pi_{-k} \mid g_k, V_k) = 0.$$

*Consequently, for each player $k$ and each state $s$, if*

$$\Omega_{k,\lambda}^{RS}(s, a_k; \pi_{-k} \mid g_k, V_k) \geq 0, \qquad \pi_k(a_k|s)\, \Omega_{k,\lambda}^{RS}(s, a_k; \pi_{-k} \mid g_k, V_k) = 0,$$

*then*

$$g_k + V_k(s) = \max_{a_k \in \mathcal{A}_k} Q_{k,\lambda}^{RS}(s, a_k; \pi_{-k} \mid V_k).$$

*Proof.* If $\mathcal{L}_\lambda = 0$, both nonnegative summands in (128) must be zero. The second term being zero implies $\left[ \Omega_{k,\lambda}^{RS}(\cdot) \right]_{-} = 0$ for all $(k, s, a_k)$, hence $\Omega_{k,\lambda}^{RS} \geq 0$ for all $(k, s, a_k)$.

The first term being zero implies $\pi_k(a_k|s)\Omega_{k,\lambda}^{RS}(s, a_k; \pi_{-k} \mid g_k, V_k) = 0$ for every $(k, s, a_k)$. This hence proves the first part.

For the second part, fix $k$ and $s$. Since $\pi_k(\cdot|s)$ is a probability distribution, there exists at least one action $\bar{a}_k \in \mathcal{A}_k$ with $\pi_k(\bar{a}_k|s) > 0$. For that $\bar{a}_k$, complementarity forces $\Omega_{k,\lambda}^{RS}(s, \bar{a}_k; \pi_{-k} \mid g_k, V_k) = 0$. Thus,

$$g_k + V_k(s) = Q_{k,\lambda}^{RS}(s, \bar{a}_k; \pi_{-k} \mid V_k).$$

But for any $a_k$ we have $\Omega_{k,\lambda}^{RS}(s, a_k; \cdot) \geq 0$, i.e., $g_k + V_k(s) \geq Q_{k,\lambda}^{RS}(s, a_k; \pi_{-k} \mid V_k)$. Therefore $g_k + V_k(s)$ equals the maximum over $a_k$. $\qquad\square$

**Corollary H.9** (Connection to (smoothed) robust NE). *Any $(\pi, g, V)$ with $\mathcal{L}_\lambda(\pi, g, V) = 0$ defines a joint policy $\pi$ such that each $\pi_k(\cdot|s)$ is supported on maximizers of $Q_{k,\lambda}^{RS}(s, \cdot; \pi_{-k} \mid V_k)$. In particular, $\pi$ is a stationary Nash equilibrium of the $\lambda$-smoothed robust game.*

*Proof.* From Lemma H.8, for fixed $(k, s)$, $\Omega_{k,\lambda}^{RS}(s, a_k) \geq 0$ for all $a_k$ and equals 0 on the support of $\pi_k(\cdot|s)$. But $\Omega_{k,\lambda}^{RS}(s, a_k) = g_k + V_k(s) - Q_{k,\lambda}^{RS}(s, a_k; \pi_{-k} \mid V_k)$, so $\Omega_{k,\lambda}^{RS}(s, a_k) = 0$ iff $a_k$ attains the maximum of $Q_{k,\lambda}^{RS}(s, \cdot; \pi_{-k} \mid V_k)$. Thus, $\pi_k(\cdot|s)$ assigns positive probability only to best-response actions at each state $s$, which is precisely the stationary NE condition for the smoothed robust game. $\qquad\square$

*Remark H.10.* Note that when $\lambda \to 0$, $\sigma_{\mathcal{P}}^\lambda(v) \to \sigma_{\mathcal{P}}(v)$, thus $\Omega_{k,\lambda}^{RS} \to \Omega_k^R$. Hence, when $\mathcal{L}_\lambda = 0$ for $\lambda$ small enough, the smoothed robust NE will result in a small value $\Delta^R$, thus is close to a robust NE. Namely, the smoothed NE is an approximation of robust NE.

## H.5. Robust TD Algorithm

We now give an actor-critic recursion that performs projected gradient steps on $\mathcal{L}_\lambda(\pi, g, V)$, with a fast critic step size $\alpha_n$ and a slow actor step size $\eta_n$.

Let $\mathcal{G} \subset \mathbb{R}^N$ be a nonempty compact convex set for gains $g$ (e.g., a box $[-G_{\max}, G_{\max}]^N$). Let $\mathcal{V} \subset \mathbb{R}^{NS}$ be a nonempty compact convex set for $V$, with the anchoring constraints built in (e.g., $V_k(s^\circ) = 0$ for all $k$ for a fixed reference state $s^\circ$).

Define the joint constraint set

$$\mathcal{Z} := \Pi \times \mathcal{G} \times \mathcal{V}.$$

Let $\Gamma_\Pi$ and $\Gamma_{\mathcal{G} \times \mathcal{V}}$ denote Euclidean projections. We then design our algorithm as follows.

## H.6. Convergence Analysis

We now give a complete convergence analysis for Algorithm 1. The structure is standard for projected gradient methods on smooth objectives, with care taken for two step sizes.

---

**Algorithm 3** Two-time-scale Robust Actor–Critic on the Merit $\mathcal{L}_\lambda$

---

1: **Input:** step sizes $\{\eta_n\}, \{\alpha_n\}$, smoothing $\lambda > 0$, penalty $\rho > 0$
2: Initialize $\pi^0 \in \Pi$, $(g^0, V^0) \in \mathcal{G} \times \mathcal{V}$
3: **for** $n = 0, 1, 2, \dots$ **do**
4:  **Critic step (fast):**
$$(g^{n+\frac{1}{2}}, V^{n+\frac{1}{2}}) = \Gamma_{\mathcal{G} \times \mathcal{V}}\Big((g^n, V^n) - \alpha_n \nabla_{(g,V)} \mathcal{L}_\lambda(\pi^n, g^n, V^n)\Big).$$

5:  **Actor step (slow):**
$$\pi^{n+1} = \Gamma_\Pi\Big(\pi^n - \eta_n \nabla_\pi \mathcal{L}_\lambda(\pi^n, g^{n+\frac{1}{2}}, V^{n+\frac{1}{2}})\Big).$$

6:  Set $(g^{n+1}, V^{n+1}) = (g^{n+\frac{1}{2}}, V^{n+\frac{1}{2}})$.
7: **end for**

---

**Assumption H.11** (Two-time-scale Robbins–Monro). The step sizes satisfy:

$$\eta_n > 0, \quad \alpha_n > 0, \quad \sum_{n=0}^{\infty} \eta_n = \infty, \quad \sum_{n=0}^{\infty} \eta_n^2 < \infty, \quad \sum_{n=0}^{\infty} \alpha_n = \infty, \quad \sum_{n=0}^{\infty} \alpha_n^2 < \infty, \quad \frac{\eta_n}{\alpha_n} \to 0.$$

**Lemma H.12** (Smoothness and Lipschitz gradient). *Fix $\lambda > 0$ and $\rho > 0$. On the compact set $\mathcal{Z}$, the merit $\mathcal{L}_\lambda$ is continuously differentiable, and its gradient is Lipschitz: there exists $L_\lambda < \infty$ such that for all $z, z' \in \mathcal{Z}$,*

$$\|\nabla \mathcal{L}_\lambda(z) - \nabla \mathcal{L}_\lambda(z')\| \leq L_\lambda \|z - z'\|.$$

*Proof.* We argue term by term.

**(i) Smoothness of $\Omega_{k,\lambda}^{RS}$.** For fixed $(k, s, a_k)$, $\Omega_{k,\lambda}^{RS}$ is given by (127). It is affine in $(g_k, V_k(s))$ and depends on $V_k$ through $\sigma_{\mathcal{P}_s^{(a_k, a_{-k})}}^\lambda(V_k)$, which is $C^1$ by Lemma H.7. It depends on $\pi_{-k}$ linearly. Hence $\Omega_{k,\lambda}^{RS}$ is $C^1$ in $(\pi, g, V)$.

**(ii) Smoothness of the squared complementarity term.** The map $(\pi, g, V) \mapsto \pi_k(a_k|s) \, \Omega_{k,\lambda}^{RS}(\cdot)$ is $C^1$ as a product of $C^1$ maps.

**(iii) Smoothness of the negative-part penalty.** The scalar function $x \mapsto [x]_-^2$ is $C^1$ everywhere: it equals 0 for $x \geq 0$, equals $x^2$ for $x < 0$, and has derivative 0 at $x = 0$. Composing a $C^1$ map $\Omega_{k,\lambda}^{RS}$ with a $C^1$ scalar function preserves $C^1$.

Thus $\mathcal{L}_\lambda$ is $C^1$.

**(iv) Lipschitz gradient on a compact set.** A $C^1$ function has locally Lipschitz gradient if its Hessian is bounded locally. Here we only need global Lipschitz on the compact $\mathcal{Z}$. Since all constituent maps are $C^1$ and $\mathcal{Z}$ is compact, $\nabla \mathcal{L}_\lambda$ is continuous on $\mathcal{Z}$ and hence uniformly Lipschitz on $\mathcal{Z}$. Formally, one can bound second derivatives using Lemma H.7 (which gives $\mathrm{Lip}(\nabla \sigma^\lambda) \leq 1/\lambda$), boundedness of $\pi$ (simplex), and boundedness of $(g, V)$ (projection set). $\qquad \square$

We recall the standard *projected gradient mapping*.

**Definition H.13** (Projected gradient mapping). Let $f$ be differentiable and $C$ a nonempty closed convex set. For $\gamma > 0$ define

$$G_{C,\gamma}(x) := \frac{1}{\gamma}\Big(x - \Gamma_C(x - \gamma \nabla f(x))\Big).$$

**Lemma H.14** (One-step descent for projected gradient). *Let $f$ be differentiable on $C$ with $L$-Lipschitz gradient. Fix $\gamma \in (0, 2/L)$ and let $y = \Gamma_C(x - \gamma \nabla f(x))$. Then*

$$f(y) \leq f(x) - \gamma\Big(1 - \frac{L\gamma}{2}\Big) \|G_{C,\gamma}(x)\|^2.$$

*Proof.* Let $y = \Gamma_C(x - \gamma \nabla f(x))$ and define $d := x - y$ so that $d = \gamma G_{C,\gamma}(x)$.

**Step 1: Projection inequality.** The projection optimality condition gives, for all $z \in C$,

$$\langle x - \gamma \nabla f(x) - y, \, z - y \rangle \leq 0.$$

Take $z = x \in C$, yielding

$$\langle x - \gamma \nabla f(x) - y, \ x - y \rangle \leq 0 \quad \Rightarrow \quad \langle d - \gamma \nabla f(x), \ d \rangle \leq 0.$$

Hence

$$\|d\|^2 \leq \gamma \langle \nabla f(x), d \rangle.$$

Equivalently,

$$\langle \nabla f(x), y - x \rangle = -\langle \nabla f(x), d \rangle \leq -\frac{1}{\gamma} \|d\|^2.$$

**Step 2: Smoothness upper bound.** Because $\nabla f$ is $L$-Lipschitz, the standard descent lemma gives

$$f(y) \leq f(x) + \langle \nabla f(x), y - x \rangle + \frac{L}{2} \|y - x\|^2 = f(x) + \langle \nabla f(x), y - x \rangle + \frac{L}{2} \|d\|^2.$$

Using the bound from Step 1:

$$f(y) \leq f(x) - \frac{1}{\gamma} \|d\|^2 + \frac{L}{2} \|d\|^2 = f(x) - \left( \frac{1}{\gamma} - \frac{L}{2} \right) \|d\|^2.$$

Since $d = \gamma G_{C,\gamma}(x)$,

$$\left( \frac{1}{\gamma} - \frac{L}{2} \right) \|d\|^2 = \left( \frac{1}{\gamma} - \frac{L}{2} \right) \gamma^2 \|G_{C,\gamma}(x)\|^2 = \gamma \left( 1 - \frac{L\gamma}{2} \right) \|G_{C,\gamma}(x)\|^2.$$

This proves the claim. $\qquad\square$

**Theorem H.15** (Convergence to first-order stationary points). *Assume Assumption H.11. Let $\{(\pi^n, g^n, V^n)\}$ be generated by Algorithm 1. Then:*

1. *The merit values $\mathcal{L}_\lambda(\pi^n, g^n, V^n)$ converge to a finite limit.*

2. *The projected gradient mappings vanish asymptotically:*

$$\lim_{n \to \infty} \|G_{\Pi, \eta_n}(\pi^n)\| = 0, \qquad \lim_{n \to \infty} \|G_{\mathcal{G} \times \mathcal{V}, \alpha_n}(g^n, V^n)\| = 0,$$

   *where each $G$ is defined w.r.t. the corresponding block objective ($\mathcal{L}_\lambda$ with the other block held fixed at the point used in Algorithm 1).*

3. *Every accumulation point $(\pi^\star, g^\star, V^\star)$ of the iterates is a (block) stationary point of $\mathcal{L}_\lambda$ on $\mathcal{Z}$, i.e. it satisfies the first-order KKT condition*

$$0 \in \nabla \mathcal{L}_\lambda(\pi^\star, g^\star, V^\star) + N_{\mathcal{Z}}(\pi^\star, g^\star, V^\star),$$

   *where $N_{\mathcal{Z}}$ is the normal cone of $\mathcal{Z}$.*

*Proof.* Let $z^n := (\pi^n, g^n, V^n) \in \mathcal{Z}$.

**Step 0: Global smoothness constant and small-step regime.** By Lemma H.12, there exists $L_\lambda$ such that $\nabla \mathcal{L}_\lambda$ is $L_\lambda$-Lipschitz on $\mathcal{Z}$. Because $\eta_n \to 0$ and $\alpha_n \to 0$ (from square summability), there exists $n_0$ such that for all $n \geq n_0$,

$$\eta_n < \frac{2}{L_\lambda}, \qquad \alpha_n < \frac{2}{L_\lambda}.$$

**Step 1: Critic-step descent.** Define the critic update map (with $\pi$ frozen at $\pi^n$):

$$y^{n+\frac{1}{2}} := (g^{n+\frac{1}{2}}, V^{n+\frac{1}{2}}) = \Gamma_{\mathcal{G} \times \mathcal{V}} \left( y^n - \alpha_n \nabla_y \mathcal{L}_\lambda(\pi^n, y^n) \right),$$

where $y^n = (g^n, V^n)$.

Apply Lemma H.14 with $f(y) = \mathcal{L}_\lambda(\pi^n, y)$, $C = \mathcal{G} \times \mathcal{V}$, and $\gamma = \alpha_n$. For all $n \geq n_0$ we get

$$\mathcal{L}_\lambda(\pi^n, y^{n+\frac{1}{2}}) \leq \mathcal{L}_\lambda(\pi^n, y^n) - \alpha_n \left(1 - \frac{L_\lambda \alpha_n}{2}\right) \left\|G_{\mathcal{G} \times \mathcal{V}, \alpha_n}(y^n)\right\|^2. \tag{129}$$

**Step 2: Actor-step descent.** Now define the actor update with critic frozen at $y^{n+\frac{1}{2}}$:

$$\pi^{n+1} = \Gamma_\Pi \left(\pi^n - \eta_n \nabla_\pi \mathcal{L}_\lambda(\pi^n, y^{n+\frac{1}{2}})\right).$$

Apply Lemma H.14 with $f(\pi) = \mathcal{L}_\lambda(\pi, y^{n+\frac{1}{2}})$, $C = \Pi$, $\gamma = \eta_n$. For all $n \geq n_0$:

$$\mathcal{L}_\lambda(\pi^{n+1}, y^{n+\frac{1}{2}}) \leq \mathcal{L}_\lambda(\pi^n, y^{n+\frac{1}{2}}) - \eta_n \left(1 - \frac{L_\lambda \eta_n}{2}\right) \left\|G_{\Pi, \eta_n}(\pi^n)\right\|^2. \tag{130}$$

Finally set $y^{n+1} = y^{n+\frac{1}{2}}$.

**Step 3: Combine and telescope.** Combine (129) and (130): for all $n \geq n_0$,

$$\begin{aligned}
\mathcal{L}_\lambda(z^{n+1}) &= \mathcal{L}_\lambda(\pi^{n+1}, y^{n+\frac{1}{2}}) \\
&\leq \mathcal{L}_\lambda(\pi^n, y^n) - \alpha_n \left(1 - \frac{L_\lambda \alpha_n}{2}\right) \left\|G_{\mathcal{G} \times \mathcal{V}, \alpha_n}(y^n)\right\|^2 \\
&\quad - \eta_n \left(1 - \frac{L_\lambda \eta_n}{2}\right) \left\|G_{\Pi, \eta_n}(\pi^n)\right\|^2.
\end{aligned} \tag{131}$$

Since $\mathcal{L}_\lambda \geq 0$ by definition (128), the sequence $\{\mathcal{L}_\lambda(z^n)\}$ is nonincreasing for $n \geq n_0$ and bounded below by 0. Hence it converges, proving (1).

Summing (131) from $n = n_0$ to $T$ yields

$$\sum_{n=n_0}^T \alpha_n \left(1 - \frac{L_\lambda \alpha_n}{2}\right) \left\|G_{\mathcal{G} \times \mathcal{V}, \alpha_n}(y^n)\right\|^2 + \sum_{n=n_0}^T \eta_n \left(1 - \frac{L_\lambda \eta_n}{2}\right) \left\|G_{\Pi, \eta_n}(\pi^n)\right\|^2 \leq \mathcal{L}_\lambda(z^{n_0}) - \mathcal{L}_\lambda(z^{T+1}) \leq \mathcal{L}_\lambda(z^{n_0}).$$

Letting $T \to \infty$ implies both sums are finite. Since $\sum_n \eta_n = \infty$ and $\sum_n \alpha_n = \infty$, the only way for the weighted sums to be finite is that the corresponding norm terms must converge to 0 along the full sequence. This proves (2).

**Step 4: Stationarity of accumulation points.** Because $\mathcal{Z}$ is compact, $\{z^n\}$ has accumulation points. Let $z^{n_j} \to z^\star$ be any convergent subsequence. From (2) and continuity of the gradient mapping under smoothness, the limiting point satisfies $G_{\Pi,0}(\pi^\star) = 0$ and $G_{\mathcal{G} \times \mathcal{V}, 0}(y^\star) = 0$, which is equivalent to the first-order normal cone inclusion $0 \in \nabla \mathcal{L}_\lambda(z^\star) + N_{\mathcal{Z}}(z^\star)$ for a projected-gradient method on a smooth function. This is the standard KKT characterization for constrained first-order stationarity, proving (3). □

# I. Proof of Section 7

**Theorem I.1.** *(1). For a sequence of discount factors $\{\gamma_t\} \to 1$, denote a Nash Equilibrium of the $\gamma_t$-discounted DR-MG by $\pi_t$. If $\pi_t \to \pi$, then $\pi$ is a Nash Equilibrium of the average-reward DR-MG.*

*(2). For any $\epsilon$, there exists some discount factor $\gamma$, such that any $\epsilon$-Nash Equilibrium of the $\gamma$-discounted DR-MG is also an $\mathcal{O}(\epsilon)$-Nash Equilibrium under the average reward.*

*(3). The discount factor $\gamma = 1 - \frac{\epsilon}{D}$ satisfies Part (2)*

*Proof.* (1). By the uniform convergence of the robust discounted value function derived in (Wang et al., 2023b), we have that

$$\lim_{\gamma \to 1} (1 - \gamma) V_{\gamma, \mathcal{P}, i}^\pi = g_{\mathcal{P}, i}^\pi, \quad \text{uniformly on } \Pi. \tag{132}$$

Thus for any $\epsilon$, there exists some $\gamma_e$, such that

$$\|(1 - \gamma) V_{\gamma, \mathcal{P}, i}^\pi - g_{\mathcal{P}, i}^\pi\| \leq \epsilon, \tag{133}$$

for any $\pi \in \Pi$ and $\gamma > \gamma_e$. Moreover, there exists some constant $T$ such that for any $t > T$,

$$\gamma_t > \gamma_e, \|\pi_t - \pi\| \leq \epsilon. \tag{134}$$

Since $\pi_t$ is the Nash Equilibrium of the discounted robust MG, it holds that

$$V_{\gamma_t,\mathcal{P},i}^{\pi_t} \geq V_{\gamma_t,\mathcal{P},i}^{(\pi_t^{-i},\mu^i)}, \forall i \in \mathcal{N}, \forall \mu^i. \tag{135}$$

Thus we have that

$$
\begin{aligned}
&g_{\mathcal{P},i}^{(\pi^{-i},\mu^i)} - g_{\mathcal{P},i}^{\pi} \\
&= -g_{\mathcal{P},i}^{\pi} + (1-\gamma_t)V_{\gamma_t,\mathcal{P},i}^{\pi} - (1-\gamma_t)V_{\gamma_t,\mathcal{P},i}^{\pi} + (1-\gamma_t)V_{\gamma_t,\mathcal{P},i}^{\pi_t} \\
&\quad - (1-\gamma_t)V_{\gamma_t,\mathcal{P},i}^{\pi_t} + (1-\gamma_t)V_{\gamma_t,\mathcal{P},i}^{(\pi_t^{-i},\mu^i)} - (1-\gamma_t)V_{\gamma_t,\mathcal{P},i}^{(\pi_t^{-i},\mu^i)} + (1-\gamma_t)V_{\gamma_t,\mathcal{P},i}^{(\pi^{-i},\mu^i)} - (1-\gamma_t)V_{\gamma_t,\mathcal{P},i}^{(\pi^{-i},\mu^i)} + g_{\mathcal{P},i}^{(\pi^{-i},\mu^i)}.
\end{aligned} \tag{136}
$$

Consider any $t > T$, it first holds that due to (133):

$$(1-\gamma_t)V_{\gamma_t,\mathcal{P},i}^{\pi} - g_{\mathcal{P},i}^{\pi} \leq \epsilon, \tag{137}$$

$$g_{\mathcal{P},i}^{(\pi^{-i},\mu^i)} - (1-\gamma_t)V_{\gamma_t,\mathcal{P},i}^{(\pi^{-i},\mu^i)} \leq \epsilon. \tag{138}$$

Moreover,

$$(1-\gamma_t)V_{\gamma_t,\mathcal{P},i}^{\pi_t} - (1-\gamma_t)V_{\gamma_t,\mathcal{P},i}^{\pi} \leq (1-\gamma_t)(V_{\gamma_t,\mathcal{P},i}^{\pi_t} - V_{\gamma_t,\mathcal{P},i}^{\pi}) \leq \epsilon, \tag{139}$$

$$(1-\gamma_t)V_{\gamma_t,\mathcal{P},i}^{(\pi^{-i},\mu^i)} - (1-\gamma_t)V_{\gamma_t,\mathcal{P},i}^{(\pi_t^{-i},\mu^i)} \leq \epsilon, \tag{140}$$

which is due to the $\frac{1}{1-\gamma}$-Lipschitz of the robust discounted value function in $\pi$, and the closeness of $\pi_t$ and $\pi$ in (134). Finally,

$$-(1-\gamma_t)V_{\gamma_t,\mathcal{P},i}^{\pi_t} + (1-\gamma_t)V_{\gamma_t,\mathcal{P},i}^{(\pi_t^{-i},\mu^i)} \leq 0 \tag{141}$$

due to (135). Combining all results hence implies that

$$g_{\mathcal{P},i}^{(\pi^{-i},\mu^i)} - g_{\mathcal{P},i}^{\pi} \leq \epsilon, \tag{142}$$

for any $i$ and $\mu^i$ uniformly. Thus let $\epsilon \to 0$ we complete the proof.

(2). Set $\gamma$ to be any factor that is larger than $\gamma_e$. Then

$$g_{\mathcal{P},i}^{(\pi^{-i},\mu^i)} - g_{\mathcal{P},i}^{\pi} = g_{\mathcal{P},i}^{(\pi^{-i},\mu^i)} - (1-\gamma)V_{\gamma,\mathcal{P},i}^{(\pi^{-i},\mu^i)} + (1-\gamma)V_{\gamma,\mathcal{P},i}^{(\pi^{-i},\mu^i)} - (1-\gamma)V_{\gamma,\mathcal{P},i}^{\pi} + (1-\gamma)V_{\gamma,\mathcal{P},i}^{\pi} - g_{\mathcal{P},i}^{\pi}. \tag{143}$$

Note that for $\gamma > \gamma_e$, it holds that

$$g_{\mathcal{P},i}^{(\pi^{-i},\mu^i)} - g_{\mathcal{P},i}^{\pi} \leq 2\epsilon + (1-\gamma)V_{\gamma,\mathcal{P},i}^{(\pi^{-i},\mu^i)} - (1-\gamma)V_{\gamma,\mathcal{P},i}^{\pi} \leq 3\epsilon, \tag{144}$$

due to $\pi$ is an $\epsilon$-NE for the discounted robust MG, which hence completes the proof.

(3). The proof is a direct corollary of Theorem 7.1 and results from (Roch et al., 2025a). Specifically, it is shown in Theorem 3.4 in (Roch et al., 2025a) that, for any $\gamma > 1 - \frac{\epsilon}{D}$[4], it holds that

$$\|(1-\gamma)V_{\gamma,\mathcal{P},i}^{\pi} - g_{\mathcal{P},i}^{\pi}\| \leq \epsilon, \forall \pi, \forall i. \tag{145}$$

Combining with Theorem 7.1 further completes the proof. □

---

[4]The results in (Roch et al., 2025a) is derived for $\mathcal{H}$ which is the robust optimal span. However, their results also hold for any constant that is larger than $\mathcal{H}$. Our claim thus holds by noting that $D \geq \mathcal{H}$ (Roch et al., 2025a).

## J. Experiments

### J.1. Experiments for Robust TD Algorithm

In this section, we provide numerical evidence that the proposed robust TD-descent algorithm (Algorithm 1) behaves in accordance with the theoretical analysis.

For a joint policy $\pi$ and gains/biases $(g, V)$ as in Definition H.2, we track the following quantities: (1). the global robust TD gap $\Delta_{\mathrm{R}}(\pi, g, V)$ as in Definition H.2. In the experiments, $(g, V)$ are always taken as the robust average-reward optimal gain-bias pairs for the current induced DR-MDPs, as we specified in Algorithm 1; (2). the robust Nash gap for the policy $\pi$: $\mathrm{NashGap}(\pi) := \max_{k \in \mathcal{N}} \mathrm{gap}_k(\pi)$ where $\mathrm{gap}_k(\pi) := \max_{\nu_k} g_{\mathcal{P},k}^{\nu_k, \pi_{-k}} - g_{\mathcal{P},k}^{\pi}$. We aim to show that, our Algorithm 1 ensures both $\Delta_{\mathrm{R}}(\pi, g, V)$ and robust Nash gap converge to 0 (note that $\mathrm{NashGap}(\pi) = 0$ implies $\pi$ is a NE), validating our theoretical results.

We consider a two-player, two-state, two-action DR-MG. The state space is $\mathcal{S} = \{0, 1\}$ and each player has two actions $\mathcal{A}_k = \{0, 1\}$ at each state. The uncertain set over transitions is two-state rectangular: for each $(s, a_1, a_2)$ the probability $p(s' = 1 \mid s, a_1, a_2)$ lies in a fixed interval $[p_{\min}(s, a_1, a_2), p_{\max}(s, a_1, a_2)]$, with the adversary choosing $p$ to minimize the player's expected next-state bias. The intervals are chosen so that the Markov chain under any stationary joint policy is irreducible, satisfying Assumption 4.1. The reward tensors $r_k(s, a_1, a_2)$ are randomly generated between $[0, 1]$. All experiments are conducted in the planning setting: the reward functions and rectangular uncertainty sets are known, and robust Bellman equations are solved through Relative Value Iteration (RVI) in (Wang et al., 2023c).

We then implement our Algorithm 1. In the experiments, we use 600 outer iterations. Figure 1 aggregates the results over $N_{\mathrm{games}} = 30$ independent random games. The left panel shows the global robust TD gap $\Delta_{\mathrm{R}}(\pi^n, g^n, V^n)$ as a function of iteration, and the right panel shows the robust Nash gap $\mathrm{NashGap}(\pi^n)$. The envelopes represent the standard deviations of these 30 runs.

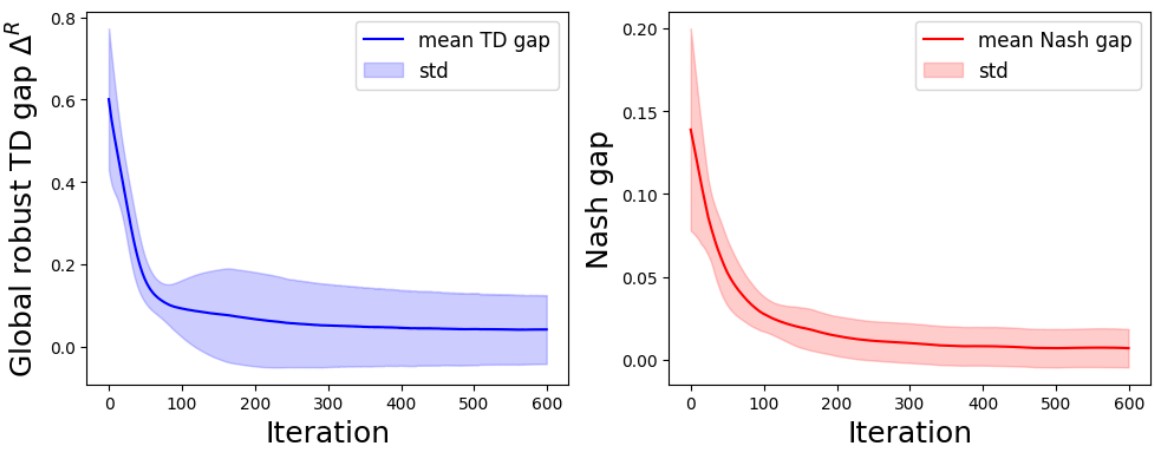

*Figure 1.* Robust TD

Across all random instances, we note that both $\Delta_{\mathrm{R}}$ and $\mathrm{NashGap}$ decrease steadily with the number of iterations, with the mean curves decreasing and approaching to zero. These results are then consistent with our theoretical results in Theorem 6.3, validating our TD-based algorithm and verifying our results.

### J.2. Additional Experiments on Average Reward DR-MGs

In this section, we develop numerical experiments to validate the advantages of our formulation and methods. We mainly consider two aspects: the long-term performance and the robustness to environment uncertainty.

#### J.2.1. LONG-TERM PERFORMANCE

To empirically validate our approach, we perform a series of experiments designed to compare the performance of our robust Nash-Iteration algorithm with average reward against the standard discounted-reward robust Nash-Iteration (Zhang et al.,

2020; Shi et al., 2024) under DR-MGs. We aim to show that our algorithm will yield a joint policy that has better long-term performance than those derived from the discounted criterion.

We design a 'Structured Random Environment' to create a complex strategic challenge with meaningful trade-offs between short-term gains and long-term consequences. The environment is a DR-MG with 20 states and 5 actions. The state space $S$ is partitioned into two clusters:

- **Prosperous States ($|S_p| = 5$):** A small set of states characterized by a higher mean reward (drawn from $\mathsf{N}(2, 1)$).

- **Deprived States ($|S_d| = 15$):** A larger set of states with a lower mean reward (drawn from $\mathsf{N}(-2, 1)$),

where $\mathsf{N}(\cdot, \cdot)$ is the *Gaussian* distribution. The nominal transition kernel $\mathsf{P}_0$ is also structured to create persistence within these clusters. Transitions originating from a state in a given cluster are five times more likely to terminate in a state within the same cluster. This design creates a strategic trap: an agent in a prosperous state might be tempted by an action with a high immediate reward that carries a significant risk of transitioning it to the deprived cluster, from which escape is difficult.

We further set the uncertainty set using a Kullback-Leibler (KL) divergence ball around the nominal kernel $\mathsf{P}_0$:

$$\mathcal{P}_s^a = \{q \in \Delta(\mathcal{S}) \mid D_{KL}(q||(\mathsf{P}_0)_s^a) \leq \theta\}.$$

Based on the scale of our environment, the uncertainty budget $\theta$ is set to a modest value of 0.01.

We compare our average-reward Robust Nash-Iteration algorithm (Algorithm 2) and the Discounted-Reward Robust Nash-Iteration. We implement the two algorithms and evaluate their output policies under the robust average-reward criterion. The experimental procedure is as follows:

(1). The optimal average-reward policy, $\pi_{avg}$, is computed using the average-reward Robust Nash-Iteration algorithm. We evaluate the worst-case average reward of Player 1 under the policy, using the robust average reward single-agent RL method in (Wang et al., 2023b).

(2). For a range of discount factors $\gamma$ from 0.5 to 0.99, the corresponding optimal discounted policy, $\pi_{disc}(\gamma)$, is computed using the Discounted-Reward Robust Nash-Iteration algorithm. And for each $\pi_{disc}(\gamma)$, we similarly evaluate its worst-case long-term performance.

For both algorithms, we plot the performance of Player 1 as a performance metric. Results are presented in Figure 2 (under different random environments). Policies learned under the discounted criterion consistently underperform the dedicated average-reward policy. This gap is most pronounced for smaller values of $\gamma$, where the myopic nature of the discounted algorithm leads it to fall into the strategic trap of the environment by choosing actions that yield high immediate rewards but lead to the persistent, low-reward "deprived" cluster of states. As $\gamma$ approaches 1, the discounted policy becomes more far-sighted, and its performance converges toward the optimal baseline.

Our results hence provide strong evidence that, for DRMGs under the average reward criterion with long-term performance optimization goals, our robust Nash-Iteration algorithm achieves a better performance, compared to the common practice of approximating this objective with a discounted formulation using $\gamma \approx 1$.

### J.2.2. ROBUSTNESS

We then test the robustness of our method by comparing it with the non-robust average-reward Nash-Iteration algorithm (Li, 2003), to test their ability to handle model uncertainty. Using the same environment, we evaluate both learners against a worst-case adversary throughout training to measure their inherent robustness. The results are presented in Figure 3. The policy learned by the robust algorithm consistently and monotonically improves its worst-case performance, rapidly converging to a high, stable value. This is because the algorithm directly optimizes for resilience against the adversary at each step. In contrast, the policy learned by the non-robust algorithm exhibits lower performance when evaluated against a worst-case adversary. Ultimately, it does not have the ability to reliably account for perturbations in the underlying environment, as opposed to its robust counterpart, which aligns with our theoretical results.

These numerical results therefore verify that our method achieves **superior long-term performance** with **increased stability** in the presence of model uncertainty in multi-agent systems, validating our theoretical results.

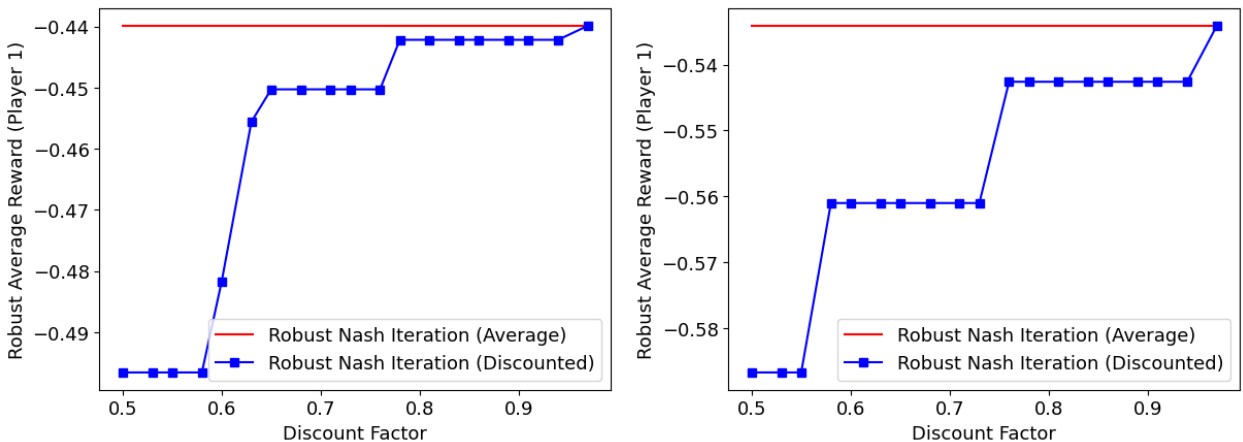

*Figure 2.* Performance Comparisons of Average and Discounted Robust Nash-Iteration

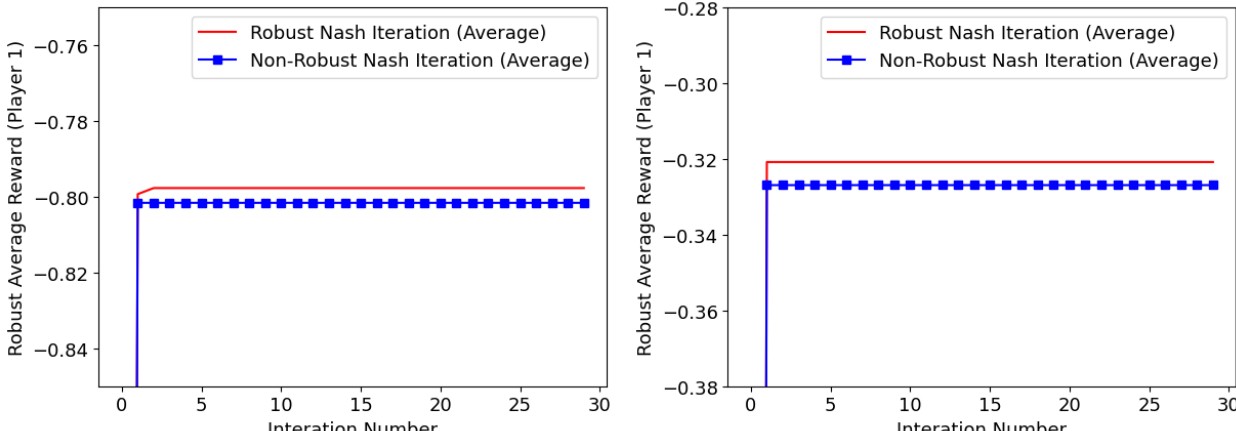

*Figure 3.* Performance Comparisons of Robust/Non-Robust Nash-Iteration

