# OpenReview forum: "Distributionally Robust Markov Games with Average Reward"
_ICML.cc/2026/Conference — ICML 2026 regular_

### Official Review · Reviewer_XAhP · 2026-03-01

**Soundness:** 4
**Presentation:** 4
**Significance:** 4
**Originality:** 4
**Overall Recommendation:** 6
**Confidence:** 5

**Summary:**

The paper explores Distributionally Robust Markov Games (DR-MGs) under the average-reward criterion, a framework designed for multi-agent systems operating over long horizons with potential model mismatches. The authors address the challenge of finding a Nash Equilibrium (NE) when transition kernels are uncertain and belong to an ambiguity set. The work examines a notable aspect of these games by establishing the correspondence between optimal policies and the solutions to robust Bellman equations in both irreducible and weakly communicating settings. Overall, the authors address a major challenge by proving the existence of stationary robust NE and proposing two convergent algorithms: Robust Nash-Iteration and Robust TD Descent.

**Compliance With Llm Reviewing Policy:**

Affirmed.

**Final Justification:**

I will keep my positive score. The paper is high quality, and the rebuttal addressed all the concerns.

**Key Questions For Authors:**

1. How does the Robust TD Descent algorithm scale as the number of agents $\mathcal{N}$ increases, particularly regarding the coordination needed for a Nash Equilibrium?

2. Can the framework be extended to cases where the uncertainty set $\mathcal{P}$ is not $(s, a)$-rectangular

3. What are the implications of your results for zero-sum games specifically, given that the standard min-max duality is often compromised in robust settings?

**Limitations:**

Yes

**Strengths And Weaknesses:**

Strengths:
1. The paper provides a solid mathematical foundation for average-reward DR-MGs, a domain significantly more complex than the discounted-reward counterpart due to the sensitivity of long-term chain structures.

2The authors present both a high-level iteration scheme (Nash-Iteration) and a practical, two-time-scale reinforcement learning algorithm (Robust TD Descent) that operates without requiring a Nash oracle.

3. By extending the analysis from irreducible to weakly communicating settings, the work ensures that its theoretical guarantees apply to a broader and more realistic class of Markov games.

4. The established connection between average-reward and discounted-reward robust NE (as $\gamma \to 1$) provides a clear path for leveraging existing discounted-reward solvers.

Weakness:
1. Similar to many robust frameworks, the inner-loop minimization over the ambiguity set $\mathcal{P}$ can be computationally expensive, particularly if the uncertainty set has a complex geometry

---

> ### Author Rebuttal · Authors · 2026-03-26
>
> We thank Reviewer XAhP for the thorough and strongly positive review, for the careful reading (confidence 5), and for the excellent scores across all criteria. Below, we response to each point of the review.
>
>
> **Q1: How does the Robust TD Descent algorithm scale as the number of agents increases?**
> The per-iteration computational complexity of Algorithm 1 is $O(N \cdot T_{\mathrm{RB}}(S, A_{\max}) + S \cdot N \cdot |\mathcal{A}| \cdot C_\sigma)$, as detailed in Section H.1. Here $|\mathcal{A}| = \prod_i |A_i|$ is the joint action space size, which grows exponentially in the number of agents $N$. This exponential dependence on $N$ is inherent to all Nash-equilibrium-based algorithms for general-sum games, as computing the joint TD errors requires summing over all joint actions. However:
>
> (1) for games with factored action spaces or local interactions, the effective joint action space can be much smaller. Our framework can exploit such structure.
>
> (2) The algorithm does not require a NE-computing oracle (unlike Algorithm 2), avoiding the PPAD-hardness bottleneck that plagues many MARL algorithms.
>
>
>
> **Q2: Can the framework be extended to cases where the uncertainty set is not $(s,a)$-rectangular?**
> This is an important and challenging question. Our current results fundamentally rely on $(s,a)$-rectangularity for two reasons:
> (1). It enables the decomposition of the multi-agent robust Bellman equation into per-agent induced DR-MDPs (Definition 4.5), which is essential for our best-response analysis. (2). It ensures that the support function $\sigma_{\mathcal{P}^a_s}(V)$ decomposes across state-action pairs, enabling tractable computation.
>
>
> Other uncertainty sets (e.g., $s$-rectangular or factor-model sets) would introduce coupling across actions and potentially across states. However, our analysis framework for the convexity and continuity could be adopted. Extending our multi-agent framework to such settings is a valuable future direction that we will highlight in the revision.
>
>
> **Q3: What are the implications for zero-sum games, given that min-max duality is often compromised in robust settings?** We first highlight that, in robust settings, there won't be any zero-sum structures in agents' payoff/value functions. Specifically, even in a two-player games with opposite rewards $r_1=-r_2$, their robust value functions will not be opposite: $V_{\mathcal{P},r_1} \neq -V_{\mathcal{P},r_2}$. Thus, such games will be treated as a general-sum game.
>
> Our results are for general robust games, which cover this special reward-zero-sum game, and our results have direct implications for them: Our Theorems 4.8 and 5.4 apply to reward-zero-sum games as a special case, confirming that robust NE exists under the assumptions. This is noteworthy because, as we discuss in Section 3, the standard min-max duality fails here, and the non-robust analysis based on the Shapley's saddle point argument cannot be applied.
>
>
>
> **Weakness: The inner-loop minimization over the ambiguity set can be computationally expensive if the uncertainty set has complex geometry.**
>
>
> We agree that the inner-loop optimization $\sigma_{\mathcal{P}^a_s}(V) = \min_{p \in \mathcal{P}^a_s} p^\top V$ is a key computational subroutine. However, in most robust reinforcement learning or distributionally robust optimization studies, many commonly used uncertainty sets are defined as distribution ambiguity sets: $\mathcal{P}^a_s=\{q\in\Delta(\mathcal{S}): D(q||p^a_s)\leq R^a_s \}$, for some nominal kernel $p$, some divergence function $D$, and some uncertainty radius $R$. Specifically, for these sets, it admits efficient closed-form or convex-optimization solutions:
>
> (1). KL-divergence balls: The solution involves a water-filling algorithm with $O(|S| \log |S|)$ complexity (Iyengar, 2005).
>
> (2). Total-variation balls: Closed-form solution in $O(|S|)$ (Nilim and El Ghaoui, 2003).
>
> (3). For more general $f$-divergence uncertainty set, the worst-case $ \min_{p \in \mathcal{P}^a_s} p^\top V$ has a specific convex duality form, which can be easily solved through gradient descent (Ghosh et al., 2025)
>
> Our complexity analysis in Section H.1 also accounts for this through the generic cost $C_\sigma$, which is polynomial for all standard uncertainty sets.

---

> > ### Author Rebuttal · Reviewer_XAhP · 2026-04-01
> >
> > I am happy with the answers and keeping my scores.

---

> > > ### Author Response · Authors · 2026-04-01
> > >
> > > We sincerely appreciate your quick response and remained strongly positive support!

---

### Official Review · Reviewer_uo2n · 2026-03-08

**Soundness:** 4
**Presentation:** 4
**Significance:** 4
**Originality:** 4
**Overall Recommendation:** 6
**Confidence:** 3

**Summary:**

This paper focuses on games played on Markov chains where the players are assumed to be robust in the sense that they are not just averse to the actions of other players, but also to the uncertainty behind the transition kernel of the game. Because the worst case transition kernel could be different for different players, the end result is a game whose the value can be lower than its certain counterpart, and lead to a potential loss of Nash equilibria (Lemma 3.1). This paper focuses on the conditions for establishing the existence of Nash equilibria. In particular, the authors show that irreducibility is sufficient for this purpose (Theorem 4.8), and then relax this to weakly communicating Markov chains (Theorem 5.4). They then propose a temporal difference descent scheme (Algorithm 1) to compute the Nash equilibrium, which also returns the optimal policies for each player.

**Compliance With Llm Reviewing Policy:**

Affirmed.

**Final Justification:**

I was originally already pleased with the paper (recommendation = 6). As such, there is no need for me to adjust my scores.

**Key Questions For Authors:**

-

**Limitations:**

No, but this paper is most primarily theoretical, so it is clear where the assumptions are made and where the results end.

**Strengths And Weaknesses:**

Thank you for the chance to review this paper. I have learnt a lot from it.

Soundness: I did not spot any specific problems, but I am not an expert in Markov games. The robust aspects look fine to me.

Presentation: The paper is easy to read.

Significance: The paper has done enough work to warrant not just one, but multiple conference papers, in my opinion. I am amply satisfied with it. An interesting direction that the authors can consider is this: At present, all of the players can have different worst-case kernels; but supposing we know that the “environment player” is “fair” in some sense, can we solve the problem where the worst-case distribution is constrained to be the same distribution for all players?

Originality: While this problem has been discussed, I suppose the solution methods are quite interesting.

---

> ### Author Rebuttal · Authors · 2026-03-26
>
> We sincerely thank Reviewer uo2n for the enthusiastic review, for noting that ``the paper has done enough work to warrant not just one, but multiple conference papers,'' and for the strongly positive assessment across all criteria.
>
> **Interesting direction: can the worst-case distribution be constrained to be the same for all players?**
> This is a fascinating suggestion. In our current formulation, each player only consider its own reward, and thus has their own worst-case kernel. Constraining all players to face the same worst-case kernel $P$ would requires all agents share the same or related rewards. There are two settings that this is the case.
>
> (1). In cooperative robust games, all agents aim to maximize the worst case w.r.t. some unified reward (e.g., all local reward could be identical, or the averaged team reward $\bar{r}=\frac{\sum r_i}{N}$). Then, all players will share the same worst-case kernel.
>
> (2). In an attack-defense robust game (can be viewed as a robust Stackelberg game), there is a single leader player aims to maximize its own reward, and all other players aim to minimize the leader's reward. Specifically, the leader is a defense player, whereas all other agents are attacker, and they can utilize the environment to attack the leader's performance. In this case, all agents only care about the leader's reward, and thus share the same worst-case kernel.
>
> In these specific games, we believe our proof techniques, particularly the Kakutani fixed-point approach and the Bellman equation analysis, can be adapted to this setting, potentially yielding stronger results (e.g., the convexity of the worst-case kernel set may be easier to exploit). We will add this as a concrete future direction in our revision and acknowledge this suggestion.

---

> > ### Author Rebuttal · Reviewer_uo2n · 2026-04-01
> >
> > Thank you for replying to my question. I will keep my score as it is. My thoughts regarding the case of a common worst-case distribution is this: Often in robust optimization, worst-case evaluation is taken constraint-wise. While it may be possible to write a formulation that compels the worst-case parameter / distribution to be the same across all constraints, this usually results in drastic degradations of tractability. In relation to the case on robust games, it causes the optimal response of each player to be coupled in some sense; thus one needs to simultaneously solve the worst-case responses for every player. I am not sure what is the implication of this, hence the question. In the second setting you suggested, this problem seems to go away, but I am not sure if it is the case in general. I would, nonetheless, argue that the common worst-case distribution case is most reflective of "reality", in that nature is somehow not really a player, but an environment of the game. Thus, I do question the "third player" interpretation of nature as an adversary. But of course, altogether my question is beyond the scope of an already rather broad examination. Thank you for entertaining my antics.

---

> > > ### Author Response · Authors · 2026-04-01
> > >
> > > We sincerely appreciate your quick response and remained strongly positive support.
> > >
> > > We fully agree with your key observation: in our case, the nature is not a player. For each player, it chooses a player-specific kernel for it, and only 'executes' this kernel only when interacting with that player. This hence makes the nature not really a player. We will adjust the discussions in our paper by changing the term 'additional environment player' to 'player-specific environment', to avoid this misleading part.
> > >
> > >
> > > On the other hand, in our per-player worst-case formulation, the rectangularity structure allows each player's worst-case kernel to be computed independently (which is what makes the induced DR-MDP decomposition in Definition 4.5 work). Under a common kernel constraint $P_1^w=P_2^w=⋯=P_N^w$, this decomposition breaks — the players' robust evaluations become coupled through the shared kernel, and the nature then becomes a player that solves a joint min problem:
> > > $$P\in \min_{P\in\mathcal{P}}​(\text{ some aggregation of all players’ values under } P).$$
> > > However, the issue here is that, the aggregation itself is unclear in the general-sum case (since players have conflicting objectives and can disagree on what the worst-case is). And if some aggregation is selected (like the team reward or the leader's reward in our examples), then the nature becomes a true player and the problem becomes a $N+1$-player game.
> > >
> > > However, this game will not be a standard Markov Game or a DR-MG (but it could be modeled as a Stackelberg game between nature and the players),  and will require additional studies.
> > >
> > > We believe the two extremes, fully independent worst-case kernels (our paper) vs. fully shared worst-case kernel, represent two ends of a modeling spectrum, and intermediate formulations (e.g., partially coupled uncertainty) are also interesting. We will incorporate this discussion in our revision. We again thank you for pointing this out and helping us to further strengthen our work.

---

### Official Review · Reviewer_knuu · 2026-03-10

**Soundness:** 3
**Presentation:** 3
**Significance:** 2
**Originality:** 2
**Overall Recommendation:** 4
**Confidence:** 2

**Summary:**

This paper studies distributionally robust Markov games (DR-MGs) with the average-reward criterion. Under irreducibility and weakly communicating assumptions, the paper establishes the existence of a stationary Nash equilibrium (NE). It also proposes two algorithms, Robust Nash Iteration and Robust TD Descent, both with provable convergence guarantees. In addition, under the irreducibility assumption, the paper shows that the average-reward NE can be approximated by the discounted DR-MGs.

**Compliance With Llm Reviewing Policy:**

Affirmed.

**Final Justification:**

This paper proves the existence of a stationary NE for DR-MGs under both irreducible and weakly communicating settings by constructing a set-valued map based on the constant-gain optimal robust Bellman operator. The paper also proposes algorithms with provable convergence guarantees, and shows that the average-reward NE can be approximated by the equilibria of discounted DR-MGs. The paper is well presented, and I believe these results are valuable to the game theory community. I was concerned about the novelty of the results and algorithms in this work. The rebuttal kindly addresses this point.

**Key Questions For Authors:**

The authors state that “although a large body of existing work develops convergent algorithms, their convergence guarantees either rely on an incorrect result...”. Could you clarify this point? Does this mean that there are prior results claiming the existence of a Nash equilibrium in distributionally robust Markov games, but with errors in their proofs? If so, is this paper the first to provide a correct proof of the existence of a Nash equilibrium in DR-MGs, or do prior correct results already exist and this paper simply offers a different proof?

I also have a question regarding the novelty of the proposed algorithms. Compared with prior algorithms, how novel are these methods? Does this paper provide the first convergence guarantee for DR-MG setting?

**Limitations:**

Yes

**Strengths And Weaknesses:**

First, this paper proves the existence of a stationary NE for DR-MGs under both irreducible and weakly communicating settings by constructing a set-valued map based on the constant-gain optimal robust Bellman operator, and it explains how this approach differs from prior techniques. The paper also proposes algorithms with provable convergence guarantees, and shows that the average-reward NE can be approximated by the equilibria of discounted DR-MGs. The paper is well presented, and I believe these results are valuable to the game theory community.

However, I am not fully convinced about the novelty of the results. Please refer to the “Key Questions For Authors” section. In addition, the convergence guarantees are asymptotic and therefore do not indicate how many iterations are needed to achieve an $\epsilon$-accurate solution. Also, those convergence results only hold for irreducible setting.

---

> ### Author Rebuttal · Authors · 2026-03-26
>
> We thank Reviewer knuu for the constructive feedback and for recognizing the value of our results to the game theory community. We believe some concerns are from some misunderstandings, and we clarify them to address the reviewer's concerns as below.
>
> **Q1: The authors state that ``although a large body of existing work develops convergent algorithms, their convergence guarantees either rely on an incorrect result...''. Could you clarify? Does this mean prior proofs have errors? Is this paper the first correct proof of NE existence?**
>
> There **does not** exist any study/proof for average reward DR-MGs before.
>
> We apologize for the misunderstandings here. This sentence is for **single-agent** robust average reward (as we mentioned in Line 257). The algorithms we discussed therein, are all for single-agent robust average reward RL. In our studies, we introduce the induced robust MDP framework, which enables us to utilize single-agent robust MDP to study multi-agent games. Thus, in this section, we mainly aim to highlight the hardness of robust average reward (even in single-agent), and develop our characterizations of the single-agent robust Bellman equation (Thm 4.6).  These prior single-agent studies do not imply our results.
>
> We also want to clarify that, the prior work on robust Markov games is extremely limited: all of them are developed for discounted/finite-horizon settings, and average-reward robust Markov games are never studied before our work (Please see the reference list of prior works in Appendix Sec A). More importantly, as we discussed in Sec 3, these existing works cannot be directly adapted and applied to average reward. Our study hence stands for the first study of average-reward robust Markov games, providing the most fundamental and first results.
>
> **Q2: How novel are the proposed algorithms compared with prior algorithms? Does this paper provide the first convergence guarantee for the DR-MG setting?**
>
> As mentioned above, our algorithms are the first with provable convergence guarantees for average-reward robust Markov games. We highlight our novelty as follows:
>
> (1). Robust Nash Iteration (Algorithm 2): While Nash value iteration is classical for non-robust Markov games (Li, 2003; Hu and Wellman, 2003), extending it to the robust setting is non-trivial. The key challenge is that the robust Bellman operator is non-linear (due to the $\min$ over the uncertainty set), and the usual contraction arguments in prior discounted settings must be replaced by a span-seminorm contraction analysis (Theorem G.2). No prior work provides such a guarantee for DR-MGs.
>
> (2). Robust TD Descent (Algorithm 1): This algorithm is entirely new. It is based on our novel TD-loss characterization of NE (Theorem 6.2), which has no analogue in the prior robust MARL literature. The algorithm bypasses the NE oracle requirement and runs in polynomial time per iteration (Section H.1). The two-time-scale convergence analysis (Theorem 6.3) leverages the Moreau-envelope smoothing technique, which is new in the context of robust games.
>
> (3). Ours are the first provably convergent algorithm for average-reward DR-MGs. Existing algorithms are all developed for discounted/finite-horizon DR-MGs (Zhang et al., 2020; Shi et al., 2024), and they cannot be applied here because they rely on the contraction property of the discounted Bellman operator or backward induction, which fail in the average-reward setting.
>
>
> **Weakness. The convergence guarantees are asymptotic and do not indicate how many iterations are needed for an $\epsilon$-accurate solution. Also, convergence results only hold for the irreducible setting.**
>
> (1). Asymptotic vs. finite-time: Our work shall be viewed as the first fundamental study of the average reward DR-MGs, which is itself valuable: the solvability or even asymptotic convergence are unclear before us. Moreover, even in the non-robust average-reward Markov game literature, finite-time convergence rates are largely open for general-sum games and will be out of this paper's scope. The asymptotic guarantees we provide are the first of their kind for DR-MGs and lay the groundwork for future finite-time analysis. We note that for more structured games (e.g., fully cooperative games), our Bellman equation results (Theorem 4.6) can potentially be combined with existing techniques to derive finite-time bounds; we plan to pursue this in follow-up work.
>
> (2). Irreducible setting for algorithms: Our existence results cover both irreducible (Theorem 4.8) and weakly communicating (Theorem 5.4) settings. We focus on the irreducible case because even single-agent average-reward robust RL under more generally setting are not yet well-developed and itself is significant challenging. We also highlight that the irreducible is the most standard and widely studied setting in robust average reward. Extending algorithms to the weakly communicating case is an important direction that our theoretical framework enables.

---

> > ### Author Rebuttal · Reviewer_knuu · 2026-04-03
> >
> > Thank you for the clarifications. I have improved my score.

---

> > > ### Author Response · Authors · 2026-04-04
> > >
> > > Dear reviewer,
> > >
> > > We sincerely appreciate your quick response and positive evaluation of our work! We are happy to know that we addressed your concern!

---

### Official Review · Reviewer_zg9a · 2026-03-15

**Soundness:** 3
**Presentation:** 3
**Significance:** 3
**Originality:** 2
**Overall Recommendation:** 4
**Confidence:** 3

**Summary:**

This paper studies distributionally robust Markov games (DR-MGs) with average reward. The main technical contributions are as follows:

1. Theorem 4.8 shows existence of a robust Nash equilibrium in DR-MGs under an irreducibility assumption (Assumption 4.1). The proof relies on existence and uniqueness of solutions to the (single-agent) robust Bellman optimality equation (Theorem 4.6)

2. Theorem 5.4 extends the existence result to weakly communicating DR-MGs (Assumption 5.1) using Kakutani’s fixed point theorem using a joint map of subsets of the best response map and recent work on robust MDPs (Wang and Si, 2025).

3. Theorem 6.1 shows convergence of a robust Nash iteration algorithm (inspired from the existing non-robust algorithm) to a robust Nash equilibrium.

4. Theorem 6.2 establishes a characterization of robust Nash equilibria in terms of a new TD loss which plays the role of a Lyapunov function and Theorem 6.3 shows that a robust TD descent algorithm inspired by the aforementioned characterization converts to stationary points of the regularized TD loss.

5. Theorem 7.1 derives connections between Nash equilibria in discounted DR-MGs and average reward DR-MGs.

**Compliance With Llm Reviewing Policy:**

Affirmed.

**Final Justification:**

I will keep my positive score, the rebuttal addressed my questions in details.

**Additional comments following up on the last authors' response:** (as the interface does not seem to allow to respond directly to the last authors' comments).

Thank you for your detailed responses, I have no further questions and I will keep my positive score.

Q1, Q2, Q3. I think it would help to add some comments to the paper for clarification.

Q4. I think this uniqueness assumption is not so natural and can be acknowledged as a limitation of the current analysis given the discussion. Noted for the reference to Wan et al. 2021 but Wan and Sutton (2022) do not require it for non-robust average-reward RL, so that setting cannot be used as a justification for the assumption being standard as in the initial rebuttal.

**Key Questions For Authors:**

Q1. Theorem 6.3 shows convergence to a stationary point to the regularized TD-loss. Is this stationary point guaranteed to be an approximate Nash equilibrium of the original DRM-MG?

Q2. In Markov games, it can be shown that first-order stationary points (not in the sense of the TD loss of Theorem 6.2) and Nash equilibria are equivalent thanks to gradient domination (see e.g. Theorem 1 in Zhang et al. 2024 or Lemma 3 in Giannou et al. 2022 for discounted reward Markov games). Any Nash equilibrium is a stationary point in general. Is the reverse implication true for DRM-MGs? I suspect it is not the case as agentwise gradient domination may not hold. Is there an implication of this fact that Nash equilibria may not coincide with first-order stationary points regarding Theorem 6.3 (which considers a different TD loss though)?

Giannou, A., Lotidis, K., Mertikopoulos, P., & Vlatakis-Gkaragkounis, E. V. On the convergence of policy gradient methods to Nash equilibria in general stochastic games. Neurips 2022.   Zhang, R., Ren, Z., & Li, N. Gradient play in stochastic games: stationary points, convergence, and sample complexity. IEEE Transactions on Automatic Control, 69(10), 6499-6514, 2024.

Q3. Theorem 6.2 shows a characterization of robust NE in terms of zeros of the TD loss you introduce. Then can you clarify why it is interesting to show convergence to stationary points of the (regularized) TD loss? Are zeros of the TD loss also stationary points of the TD loss, i.e. are they actually (at least local) minimizers?

Q4. Can you comment more on the need to add an existence and uniqueness requirement of the Bellman equation (7) in Assumption 5.1 and on how reasonable is this assumption?

**Limitations:**

See weaknesses, I do not see major ones otherwise.

**Strengths And Weaknesses:**

**STRENGTHS:**

1. The paper is nicely written and easy to follow. The exposition is gradual and culminates in the discussion of computational methods.
2. Compared to the discounted reward setting, the average reward setting is less studied and more challenging, especially for Markov games. The average reward setting in RL, the study of robust MDPs to model uncertainty and Markov games to capture strategic interaction are active research topics. The paper sits at the intersection of these.
3. The work establishes some fundamental properties of DRM-MGs around the existence of equilibria and their computation. I believe this paper and its results will motivate future work as it provides some basis to build on.

**SOUNDNESS and TECHNICAL NOVELTY:**

 The results are sound and build on some fundamental tools like Kakutani’s fixed point theorem and properties of robust Bellman operators. Skimming through the proofs, they look fairly simple and build on the robust MDP literature but they look complete and detailed.  Nevertheless, I am not an expert on robust MDPs and I am not very familiar with the closely related literature in that space, which looks important to evaluate technical novelty. Proofs were not carefully checked.


**WEAKNESSES:**

The results are mainly about existence and asymptotic convergence of some algorithms that either require strong oracles (NE computation) that are notoriously difficult to access (especially in general sum cases without structure and existing PPAD complexity results) or some procedures that do not seem to guarantee convergence to (approximate) robust NE. The paper does not further investigate finite-time guarantees in more structured DRM-MGs. This is not a major concern though as fundamental questions such as existence and asymptotic convergence are not yet settled.

**Additional minor comments:**

1. l. 139: Existence in the $n$-player general-sum setting has been proved by Fink 1964, Shapley 1953 is restricted to two-player zero-sum Markov games.
 2. l. 168: ‘and it becomes even more pronounced in general-sum or multi-player settings’. I am not sure to understand this comment here, what would be duality in an $N$-player setting (beyond the zero-sum Markov game setting which relies on it as exploited by Shapley 1953)?
3. p. 7, l. 383: ‘Notably, these assumptions’, which assumptions?
4. p. 8, Theorem 7.1: ‘the Nash equilibrium’, Nash equilibria are not guarantees to be unique in general.
5. Typos: l. 218 ‘reslut’, l. 252: $\sigma$ used before its definition in l. 226 2nd column, l.313 ‘unieuq’.

---

> ### Author Rebuttal · Authors · 2026-03-26
>
> We sincerely thank Reviewer zg9a for the detailed review and overall positive evaluations. Below we response to each of your questions.
>
> We first want to clarify our proofs, although rely on single-agent MDPs, are not directly achievable. For instance, the uniqueness/convexity of Bellman equation solutions, optimality of proxy map in weakly communicating settings, and convergence analysis are new and absent from prior works.
>
> **Q1. Stationary point and approximate NE.**
> We clarify the connection in two steps:
> (1). Smoothed game: By Corollary H.9, any global minimizer of $L\_\lambda$ (with value $0$) is a NE of the $\lambda$-smoothed robust game. Thus stationary points with $L\_\lambda \approx 0$ are therefore approximate NE of the smoothed game.
>
> (2). Original game: As $\lambda \to 0$, $\sigma^{\lambda}\_{\mathcal{P}}(v) \to \sigma\_{\mathcal{P}}(v)$ uniformly (Remark H.10), so the smoothed TD errors converge to the original ones. Thus, for sufficiently small $\lambda$, any approximate NE of the smoothed game is an $O(\lambda + \epsilon)$-approximate NE of the original DR-MG.
>
> We acknowledge that a stationary point may not be a robust NE, but every NE is a stationary point. Hence our robust TD algorithm converges to a weaker solution to the DR-MGs.
>
> **Q2. Gradient domination.**
> The gradient domination property established in Zhang et al. (2024) and Giannou et al. (2022) crucially relies on the smoothness of the value function with respect to policies. In DR-MGs, the robust value function involves a $\min$ over the uncertainty set, making it generally non-smooth. As a result, agentwise gradient domination does not hold in general for DR-MGs. We could bypass this by our Moreau-envelope smoothing $L_\lambda$ to recover differentiability while approximating the original objective as $\lambda \to 0$.
>
> On the other hand, we expect that under some additional assumptions, a form of gradient domination may be recoverable for the smoothed problem. For instance, as in Giannou et al. (2022), if the robust NE is a strict equilibrium, we could utilize the connection of our TD loss and policy gradient to similarly derive a local gradient domination for the smoothed game, and show local convergence to NE with initializations close to this NE. We will leave this as an important future direction.
>
> **Q3. Convergence to stationary points.** As we mentioned above, being a robust NE is a sufficient condition for being a stationary point of $L(\pi)$. With some additional conditions, we also expect to derive a local gradient domination, which implies our algorithm will locally converge to a robust NE. Thus, our stationary convergence can be viewed as the first step for average reward DR-MGs.
>
> Our experiments (Appendix J.1) also show that the algorithm consistently drives both $L$ and the Nash gap to near zero, suggesting that our stationary convergence could also ensure good performance empirically.
>
> Finally, we highlight that, the convergence to exact NE is generally unachievable for general-sum games, even in non-robust settings. Existing convergence results either require the NE computing oracle (as our Nash iteration algorithm), or guarantee only local convergence (Zhang et al. 2024, Giannou et al. 2022)/global convergence under stronger structures (like potential games (Zhang et al. 2024). Moreover, with robustness, the problem becomes much more challenging. Our robust TD algorithm thus stands for the first implementable algorithm for average reward DR-MGs, and we will explore stronger convergence with some additional structures.
>
>
> **Q4. Assumption.**  The uniqueness is a standard assumption in both robust and non-robust average-reward settings. It is adopted, for instance, in Wan and Sutton (2022) for non-robust average-reward RL; Moreover, as we show in Lemma E.1, the commonly used unichain assumption (Wang et al., 2023b,c) automatically implies this uniqueness.
>
> Without uniqueness, the proxy map $G_i(\pi_{-i})$ in Section 5 could become ill-defined or set-valued in ways that prevent application of Kakutani's fixed-point theorem. Uniqueness ensures that the Bellman-greedy response is always well-defined and varies continuously with the opponents' policies, and it is a necessary condition for uniform boundedness of the solutions for the induced Bellman equation, which is important for hemi-continuity.
>
> We highlight that this assumption is due to the current very limited understanding of weakly communicating robust MDP. It will be our next step to consider more generally settings and extend our results.
>
> **Minor comment and typos.**
> Thank you for the correction. We will revise line 139 to properly attribute the $n$-player general-sum existence result to Fink (1964). For Line 168, we agree this sentence is imprecise. Our intended meaning is that in general-sum games, there is no saddle-point structure to exploit at all, making equilibrium existence harder to establish. We will revise this passage for clarity.

---

> > ### Author Rebuttal · Reviewer_zg9a · 2026-04-04
> >
> > I thank the authors for the detailed rebuttal, I would like to maintain my current positive score.
> >
> > I have a few follow-up clarification questions:
> >
> > Q1. Stationary point and approximate NE. Why do we have that the relaxation (1) from exact to approximate? Does it follow simply from the definition of Nash to get this robustness? I guess that the method does not guarantee that the loss $L_{\lambda}$ is close to zero (globally) because of non-convexity, so stationary points are not necessarily robust NE (unless the loss is small). I think it is worth highlighting this limitation.
> >
> > Q2. I see, thank you for the clarification. This is interesting. I expect the idea of considering smooth losses to pose some challenges when taking $\lambda$ to zero but it may help establishing some connection.
> >
> > Q3. I agree regarding convergence to robust NE. I think my main point is is to highlight the limitation regarding the fact that the set of stationary points does not necessarily coincide with the set of Nash in DR-MG in contrast to Markov games where it does.
> >
> > Q4. Regarding uniqueness, it is typically not required to apply Kakutani's fixed point theorem which is designed to handle set valued mappings (e.g. best response like in the proof of existence of stationary Markov Nash equilibria in Markov games). So I am not sure this has to be the reason to make such an assumption. Continuity is not a reason neither as hemicontinuity is enough. Maybe uniqueness simplifies some part of the proof for the application of Kakutani's fixed point theorem. Can you clarify the need of such an assumption? The rebuttal mentions that the commonly used unichain assumption guarantees uniqueness. But then, why should uniqueness be meaningful under the weaker weakly communicating assumption? Is there a way to guarantee that this assumption is at least possible? Also in Wan and Sutton (2022) for average-reward RL, I was not able to locate a similar additional uniqueness assumption among their 5 assumptions.

---

> > > ### Author Response · Authors · 2026-04-04
> > >
> > > We appreciate the reviewer's reply and positive evaluation of our work. We further clarify and answer your questions as follows.
> > >
> > > Q1. We would like to clarify that we did not claim every stationary point is a Nash equilibrium. As stated in our response, only a stationary point with $L_\lambda \approx 0$ (which is a global minimizer, since $L_\lambda \geq 0$) approximates a smoothed NE. On the other hand, we showed that any smoothed NE will be zero points of $L_\lambda$ and hence is a stationary point.
> > >
> > > Our algorithm guarantees convergence to a stationary point of $L_\lambda$, which is a necessary condition for being a NE of the smoothed game, but this stationary point may not be a global minimizer due to non-convexity.
> > >
> > > Thus, our convergence result should be interpreted as convergence to a stationary point of a proxy function $L_\lambda$, whose global minimizer is approximately a Nash equilibrium of the original DR-MG as $\lambda \to 0$. That is, the returned stationary point satisfies a necessary condition for being an approximate Nash equilibrium of the original DR-MG. We noted this distinction in our paper (Line 419) that this convergence is weaker than our Nash iteration, and will further clearly state this limitation in the revision.
> > >
> > >
> > > Q4. We apologize for our unclear discussion in the rebuttal due to space limitations. You are correct that Kakutani's theorem itself does not require uniqueness, and this assumption may not be necessary but can simplify proofs.
> > >
> > >
> > > The uniqueness in Assumption 5.1 is not for Kakutani directly, but for ensuring the proxy map $G_i(\pi_{-i})$ is well-defined and upper hemicontinuous. Specifically, $G_i(\pi_{-i})$ is constructed from the solution $(h^\star_{\pi_{-i}}, \alpha^\star_{\pi_{-i}})$ of the optimal robust Bellman equation for the induced DR-MDP $M_i(\pi_{-i})$. Without uniqueness (up to constant shifts), multiple solutions could exist with different relative values $h^\star(s) - h^\star(s')$, leading to different greedy sets $M_s(\pi_{-i})$ and making $G_i$ depend on the choice of solution. Uniqueness ensures $G_i$ is unambiguously defined. Furthermore, in proving upper hemicontinuity (Prop E.7, Theorem E.13, Lemma E.11), we need that as $\pi^n_{-i} \to \pi_{-i}$, the normalized solutions converge to a **unique** limit  (otherwise the maximizer sets $M_s$ could jump discontinuously depending on which solution branch is selected), and this can be guaranteed by the uniqueness.
> > >
> > > However, we totally agree with the reviewer and acknowledge that this assumption may be removable with a more refined argument — for instance, by defining $G_i$ as a union over all possible solutions and verifying the Kakutani conditions for this enlarged map. However, this could be challenging and beyond the paper's scope, as the current understanding of weakly communicating robust Bellman equations (Wang and Si, 2025) is still limited and additional studies are needed for such an extension.
> > >
> > >
> > >
> > > We cited unichain to show that our assumption is implied by a commonly used condition, not to suggest it requires unichain. The uniqueness in Assumption 5.1 concerns only the optimal Bellman equation, not the Bellman equation for every policy. A weakly communicating robust MDP can fail to be unichain (some policies induce multiple recurrent classes) while still having a unique optimal bias function — this occurs when the optimal policies do not create competing recurrent classes with identical gain but distinct bias structures. In this sense, uniqueness is a strictly weaker requirement than unichain, making it a meaningful assumption within the weakly communicating setting.
> > >
> > >
> > > We also apologize for the incorrect reference provided. The reference should be (Wan et al., 2021: Learning and Planning in Average-Reward Markov Decision Processes).  In their Theorem 1, assumption 2 requires the uniqueness of the solution. We will correct this reference mistake in our paper.
> > >
> > >
> > > Q2 \& Q3. Thank you for the clarification. We fully agree with Q3: the set of stationary points does not necessarily coincide with the set of NE in DR-MGs, and this is an important distinction from non-robust Markov games where gradient domination can close this gap. We will state this explicitly in the revision. For Q2, we agree that the behavior as $\lambda \to 0$ deserves further study, and we plan to investigate the connection between the smoothed and original loss landscapes in future work. We thank the reviewer for the constructive discussion, and we will include discussions on them in our paper.

---

### Decision · Program_Chairs · 2026-04-30

**Decision:**

Accept (regular)

**Comment:**

The paper developed new theoretical foundation for distributionally robust Markov games with the average reward. The discussion reached a consensus about the contribution of the paper. A nitpicky comment is it will still be beneficial to include some toy numerical examples to demonstrate the theoretical and algorithmic results in the paper.